# ODE Discovery for Longitudinal Heterogeneous Treatment Effects Inference

**Krzysztof Kacprzyk**[*]
University of Cambridge

**Samuel Holt**[*]
University of Cambridge

**Jeroen Berrevoets**[*]
University of Cambridge

**Zhaozhi Qian**
University of Cambridge

**Mihaela van der Schaar**
University of Cambridge

## Abstract

Inferring unbiased treatment effects has received widespread attention in the machine learning community. In recent years, our community has proposed numerous solutions in standard settings, high-dimensional treatment settings, and even longitudinal settings. While very diverse, the solution has mostly relied on neural networks for inference and simultaneous correction of assignment bias. New approaches typically build on top of previous approaches by proposing new (or refined) architectures and learning algorithms. However, the end result—a neural-network-based inference machine—remains unchallenged. In this paper, we introduce a different type of solution in the longitudinal setting: a closed-form ordinary differential equation (ODE). While we still rely on continuous optimization to learn an ODE, the resulting inference machine is no longer a neural network. Doing so yields several advantages such as interpretability, irregular sampling, and a different set of identification assumptions. Above all, we consider the introduction of a completely new *type* of solution to be our most important contribution as it may spark entirely new innovations in treatment effects in general. We facilitate this by formulating our contribution as a framework that can transform any ODE discovery method into a treatment effects method.

## 1 Introduction

Inferring treatment effects over time has received a lot of attention from the machine-learning community (Lim, 2018; Schulam & Saria, 2017; Gwak et al., 2020; Bica et al., 2020b). A major reason for this is the wide range of applications one can apply such a longitudinal treatment effects model. Consider for example the important problem of constructing a treatment plan for cancer patients or even a training schedule to combat unemployment.

The increased attention from the machine learning community resulted in many methods relying on novel neural network architectures and learning algorithms. Once trained, these neural nets are used as inference machines, with innovation focusing on new architectures and learning strategies. This type of approach has indeed yielded many successes, but we believe that for some situations, one may want to consider an entirely different type of model: the *ordinary differential equation* (ODE).

This point is illustrated in fig. 1, where we show a standard treatment effects (TE) model on the left, and our new approach on the right. Essentially, a standard TE model will first build a representation to adjust the covariate shift—e.g. through propensity weighting (Robins et al., 2000) or adversarial learning (Bica et al., 2020b)—used to model the outcome. Conversely, our new approach does *not* adjust the dataspace in any way, and instead discovers a global ODE which we refine *per patient*, as the goal of ODE discovery is finding underlying closed-form concise ODEs from observed trajectories.

Doing so yields advantages ranging from interpretability to irregular sampling (Lok, 2008; Saarela & Liu, 2016; Ryalen et al., 2019), and even modifying certain assumptions relied upon by contemporary techniques (Pearl, 2009; Bollen & Pearl, 2013). Most of all, we believe that proposing this new strategy may spark a radically different approach to inferring treatment effects over time. Moreover, given that the resulting model is an equation, one may use the discovered solution to engage in further

---

[*]Equal contribution; authors listed in reverse alphabetic order.

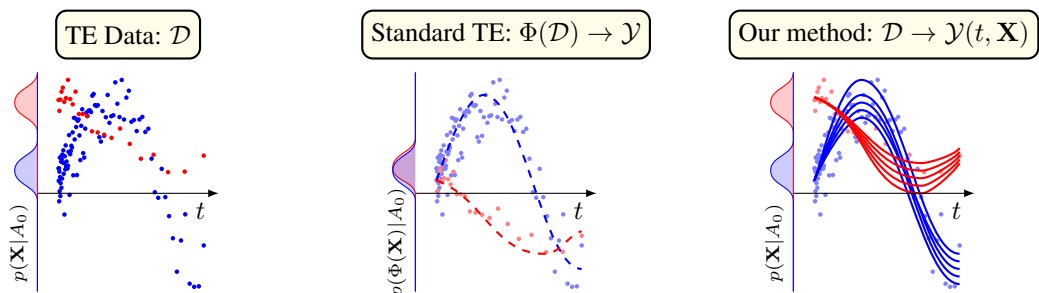

Figure 1: **Conceptual overview.** *Left:* A longitudinal treatment effects dataset with biased samples. *Middle:* A standard treatment effects (TE) approach will learn a representation of the data, $\mathcal{D}$, and will use the representation for inference. *Right:* Our approach, which learns an ODE, refined for each specific patient in the dataset.

research and data collection, given that we can now *understand* certain behaviours of the treatment in the environment. The latter is, of course, not possible when using black-box neural network models (Rudin, 2019; Angrist, 1991; Kraemer et al., 2002; Bica et al., 2020a; 2021).

**Our contribution** is a usable framework that allows us to translate any ODE discovery method (Brunton et al., 2016) into the treatment effects problem formulation. While other TE inference methods have *used* ideas from ODE literature, none have focused on ODE *discovery*, we have devoted appendix C.1 to explain this subtle but important difference. Hence, in this paper, we first explain the differences between ODE discovery (and ODEs in general), and treatment effects inference, before connecting them through our proposal. Using our framework, we build an example method (called INSITE)[1], tested in accepted benchmark settings used throughout the literature. Transforming an ODE discovery method into a TE method results in a new set of identification assumptions. This need not be a limitation, instead, it can be seen as an *extension* as the typical TE identification assumptions may not always hold in ODE discovery settings.

## 2 HETEROGENEOUS TREATMENT EFFECTS OVER TIME

To reformulate the longitudinal heterogeneous treatment effects problem as an ODE discovery problem, in this section, we first introduce longitudinal treatment effects with its assumptions and then introduce ODE discovery in section 3. Let random variables $\boldsymbol{X}_t^{(i)} \in \mathbb{R}^d$, $\boldsymbol{A}_t^{(i)} \in \mathcal{A}$ and $Y_t^{(i)} \in \mathcal{Y}$ be the $i^{\text{th}} \in [N]$ individual's features, treatment, and outcome at time $t \in [0, T]$, with $T$ the time horizon. We also denote the static covariates of the $i^{\text{th}}$ individual as $\boldsymbol{V}^{(i)} \in \mathbb{R}^m$. In the treatment effects literature, typically $\mathcal{A} = \{0, 1\}$ and $\mathcal{Y} = \mathbb{R}$. Unless required for clarity, we will drop the individual indicator $i$. We use the following notation $\boldsymbol{X}_{t_1:t_2}$ to denote the observed features in the time window $[t_1, t_2]$. The observed dataset contains the realizations of the random variables above, $\mathcal{D} = \{(\boldsymbol{v}, \langle \boldsymbol{x}_t, y_t, \boldsymbol{a}_t \rangle)^{(i)} : i \in [N], t \in [0, T^{(i)}]\}$.

Of interest is estimating the expected potential outcome $Y_{t:t+\tau}(\bar{\boldsymbol{a}}_{t:t+\tau})$, for some $\tau > 0$ under *hypothetical* future treatments $\bar{\boldsymbol{a}}_{t:t+\tau}$ given the *historical* features $\boldsymbol{X}_{0:t}$ and the previous treatments $\boldsymbol{A}_{0:t}$ (Neyman, 1923; Rubin, 1980):

$$\mathbb{E}[Y_{t:t+\tau}(\bar{\boldsymbol{a}}_{t:t+\tau})|\boldsymbol{V}, \boldsymbol{X}_{0:t}, \boldsymbol{A}_{0:t}] \tag{1}$$

By definition, the potential outcome defined above includes multiple hypothetical scenarios that cannot be simultaneously observed in the real world. This problem is often referred to as the fundamental challenge to treatment effects and causal inference (Holland, 1986).

As such, treatment effects (over time) literature typically introduces a set of assumptions to link the (unobservable) potential outcomes to the observable quantities $Y$, $\boldsymbol{X}$, and $\boldsymbol{A}$, to correctly estimate $Y_{t:t+\tau}(\bar{\boldsymbol{a}}_{t:t+\tau})$ in eq. (1). These assumptions are:

**Assumption 2.1 (Consistency)** *For an observed treatment process $\boldsymbol{A}_{0:T^{(i)}} = \boldsymbol{a}$, the potential outcome is the same as the factual outcome $\boldsymbol{Y}(\boldsymbol{a}) = \boldsymbol{Y}_{0:T^{(i)}}$.*

---

[1]We provide code at https://github.com/samholt/ODE-Discovery-for-Longitudinal-Heterogeneous-Treatment-Effects-Inference.

**Assumption 2.2 (Overlap)** *The treatment intensity process $\lambda(t|\mathfrak{F}_t)$ is not deterministic given any filtration $\mathfrak{F}_t$[2] (Klenke, 2008) and time point $t \in [0, T]$, i.e.,*

$$\gamma < \lambda(t|\mathfrak{F}_t) = \lim_{\delta t \to 0} \frac{p(A_{t+\delta t} - A_t \neq 0|\mathfrak{F}_t)}{\delta t} < 1 - \gamma, \quad with \quad \gamma \in (0, 1)$$

**Assumption 2.3 (Ignorability)** *The intensity process $\lambda(t|\mathfrak{F}_t)$ given the filtration $\mathfrak{F}_t$ is equal to the intensity process that is generated by the filtration $\mathfrak{F} \cup \{\sigma(Y_s) : s > t\}$ that includes the $\sigma$-algebras generated by future outcomes $\{\sigma(Y_s) : s > t\}$.*

The above generalizes the standard identification assumptions made in static treatment effect literature (Rosenbaum & Rubin, 1983; Seedat et al., 2022). This generalization is largely based on previous extensions to continuous-time causal effects (Lok, 2008; Saarela & Liu, 2016; Ryalen et al., 2019).

Assum. 2.2 and assum. 2.3 rely on a *treatment intensity process*, $\lambda(t|\mathfrak{F}_t)$ which can be considered a generalization of the propensity score in continuous time (Robins, 1999). Essentially, assum. 2.2 allows that any treatment can be chosen at time $t$, given the past observations in the filtering $\mathfrak{F}_t$. Furthermore, assum. 2.3 ensures it is sufficient to condition on a patient's past observed trajectory to block any backdoor paths to future potential outcomes (eq. (1)).

Given the above, we are allowed the following equality, which identifies the potential outcome (LHS) to be estimated using the observed variables (RHS):

$$\mathbb{E}[Y_{t:t+\tau}(\bar{a}_{t:t+\tau})|V, X_{0:t}, A_{0:t}] = \mathbb{E}[Y_{t:t+\tau}|V, X_{0:t}, A_{0:t}, A_{t:t+\tau} = \bar{a}_{t:t+\tau}], \qquad (2)$$

which is exploited by most works in the treatment effects literature (albeit by first regularizing models such that they respect assum. 2.1 to 2.3) (Bica et al., 2020b; Lim, 2018; Melnychuk et al., 2022). A thorough overview of related works is presented in appendix C.

## 3 UNDERPINNINGS OF ORDINARY DIFFERENTIAL EQUATION DISCOVERY

We now propose to model this treatment effect over time problem as a dynamical system, where their temporal evolution can often be well represented by ODEs (Hamilton, 2020). That means we assume the time-varying features and outcomes of the $i^{\text{th}}$ individual ($x_t^{(i)} \, \forall t \in [0, T^{(i)}]$) are discrete (and possibly noisy) measurements of underlying continuous trajectories of observed features $x^{(i)} : [0, T] \to \mathbb{R}^d$ and potential outcomes $y^{(i)} : [0, T] \to \mathbb{R}$, where $T \in \mathbb{R}$ is called the *time horizon* (Birkhoff, 1927). To make our formalism consistent, we also assume there is a *treatment trajectory* $a : [0, T] \to \mathbb{R}^k$ such that $a_t^{(i)}$'s either constitute snapshots of the underlying continuous treatment $a$ or if the treatments are administered at discrete times then $a$ is a step function whose values corresponds to the currently administered treatment.

There are many fields which already use ODEs as a formal language to express time dynamics. One such example is pharmacology, where a large portion of the literature is dedicated to recovering an ODE from observational data. The found ODE is then used to reason about possible treatments and disease progression (Geng et al., 2017; Butner et al., 2020). We further assume that this system is modelled by a system of ODEs which describes the time derivative of $x$ as a function of $x$, $v$, and $a$. The outcome trajectory, $y$, depends on the features $x$. In particular, we assume

$$\frac{dx(t)}{dt} = \dot{x}(t) = F(v, x(t), a(t)) \quad \text{and} \quad y(t) = g(x(t)), \qquad (3)$$

where $\dot{x}(t)$ is the differential of $x$. $g$ is prespecified by the user and often models the outcome as one of the features ($g(x) = x_j$), e.g., the tumour volume. This is a general formulation that may also include direct effects of $a$ on $y$. We further discuss eq. (3) in appendix D. The goal of ODE discovery is to recover the underlying system of ODEs $F$ based on the observed dataset $\mathcal{D}$ (Brunel, 2008; Brunel et al., 2014). Of course, the reliance on such a dataset implies that, while the trajectories are *defined* in continuous time, they are *observed* in discrete time. Moreover, methods for ODE discovery make the following assumptions:

**Assumption 3.1 (Existence and Uniqueness)** *The underlying process can be modelled by a system of ODEs $\dot{x}(t) = F(v, x(t), a(t))$,[3] and for every initial condition $x_0$, $v$ and treatment plan $a$*

---

[2]We further explain treatment intensity processes and filtrations in the assumptions in appendix B.

[3]Current ODE discovery methods are not designed to work with static features but they can be adapted to this setting by considering them as time-varying features that are constant throughout the trajectory.

*at $t_0$, there exists a unique continuous solution $\boldsymbol{x} : [t_0, T] \rightarrow \mathbb{R}^d$ satisfying the ODEs for all $t \in (t_0, T)$(Lindelöf, 1894; Ince, 1956).*

**Assumption 3.2 (Observability)** *All dimensions of all variables in $\boldsymbol{F}$ are observed for all individuals, ensuring sufficient data to identify the system's dynamics and infer the ODE's structure and parameters (Kailath, 1980).*

**Assumption 3.3 (Functional Space)** *Each ODE in $\boldsymbol{F}$ belongs to some subspace of closed-form ODEs. These are equations that can be represented as mathematical expressions consisting of binary operations $\{+, -, \times, \div\}$, input variables, some well-known functions (e.g., $\{\log, \exp, \sin\}$), and numeric constants (e.g., $\{-0.2, \ldots, 5.2\} \in \mathbb{R}$) (Schmidt & Lipson, 2009).*

**Identification.** Assum. 3.1 and 3.2 play a crucial role in ODE discovery. Essentially, we require making such assumptions in order to correctly *identify* the underlying equation. The assumptions made in the treatment effects literature (assum. 2.1 to 2.3) serve a similar purpose as they allow us to interpret the estimand as a *causal* effect, i.e., they are necessary for identification (Imbens & Rubin, 2015; Rosenbaum & Rubin, 1983; Imbens, 2004).

Assum. 3.1 ensures that the discovered ODE has a unique solution, which is essential for making reliable predictions, assum. 3.2 is necessary such that the observed data can be used to accurately identify the underlying ODE. Finally, assum. 3.3 defines the space of equations for the optimization algorithm to consider. We review methods for ODE discovery in appendix C.

## 4    CONNECTING TREATMENT EFFECTS INFERENCE AND ODE DISCOVERY

### —THE FRAMEWORK

From eqs. (2) and (3), one can see that both treatment effects and ODE discovery involve learning a function that can issue predictions about future states. Although the connection is natural, there remain some discrepancies between these two fields. To apply ODE discovery methods for treatment effects inference, we have to resolve these discrepancies. Essentially, we establish a framework one can use to apply *any* ODE discovery technique in treatment effects.

We identify three discrepancies between the treatment effects literature and the ODE discovery literature: (1) different assumptions, (2) discrete (not continuous) treatment plans, and (3) variability across subjects. Each discrepancy is explained and reconciled in a dedicated subsection below with actionable steps.

Figure 2 shows the areas ODE discovery methods can be expanded to, using our framework. Ranging from simple adaption, to proposing a completely new method (in section 5), our framework significantly increases the reach of existing ODE discovery methods. Our new method—INSITE—should be considered the result after *implementing* our practical framework we present in the remainder of this section.

Figure 2: **Dimensions of our framework.** The x-axis shows different treatment types (cfr. section 4.2) and the y-axis shows between-subject variability in increasing difficulty (cfr. section 4.3). ✓indicates "no adapting needed", and ✿ shows our framework. In green shows what ODE discovery methods can do out of the box. Blue shows INSITE's possibilities, encompassing all settings, including complex BSV.

### 4.1    DECIDING ON ASSUMPTIONS (DISCR. 1)

**Discrepancy 1 (Different assumptions.)** *In sections 2 and 3 we listed the most common assumptions made in treatment effects and ODE discovery literature, respectively. While solving similar problems, these assumptions do not correspond one-to-one (table 1). However, the fact that they do not can be seen as a major advantage—in some scenarios it may be more appropriate to assume assum. 2.1 to 2.3 versus assum. 3.1 to 3.3 or vice versa, which expands the application domain.*

Increasingly, the treatment effects literature is considering settings that violate the overlap assumption assum. 2.2 (D'Amour et al., 2021). It has been shown that correct model specification can weaken or even replace the overlap assumption (Gelman & Hill, 2006, Chapter 10). The same holds true for

Table 1: **Comparing assumptions.** This table lists the typical assumptions made in ODE discovery and in treatment effects. While some seem to correspond (or are at least similar), we have shaded in green where we could relax some assumptions with others. This is a powerful idea stemming directly from connecting these two fields. We show the robustness of these assumptions in violating settings in section 6. We note that the correspondence of similarity between these assumptions is not formal but indicates the discrepancy between the formalisms of treatment effect inference and ODE discovery that needs to be considered as part of the framework.

| ODE discovery | | Treatment effects | | Explanation |
|---|---|---|---|---|
| *ref* | *assumption* | *assumption* | *ref* | |
| 3.1 | *existence & uniqueness* | *consistency* | 2.1 | 2.1 is *implicit* through 3.2. |
| 3.2 | *observability* | *overlap* | 2.2 | 2.2 can be relaxed by 3.1 and 3.3 |
| 3.3 | *functional spaces* | *ignorability* | 2.3 | 2.3 is *similar* as 3.2. |

the ODE discovery methods, where overlap in assum. 2.2 can be relaxed with assum. 3.1 and 3.3. Consider an example where the true and specified models are both linear. Here, overlap can be violated as we can safely extrapolate outside the support of either treatment distribution. In contrast, the recent neural network-based treatment effects models can rarely satisfy correct model specification (Lim, 2018; Bica et al., 2020b; Berrevoets et al., 2021; Melnychuk et al., 2022). As a result, the overlap assumption plays a key role for these methods. We show this empirically in section 6.

> **Framework step 1:** *Accept ODE discovery assumptions (assum. 3.1 to 3.3).*

### 4.2 INCORPORATING DIVERSE TREATMENT TYPES (DISCR. 2)

**Discrepancy 2 (Discrete treatment plans.)** *In section 3 we assume that $a$ is a trajectory like $x$ and $y$, with $a_t$ a snapshot observed similarly to observing covariates and outcomes. This implies that, like covariates and outcomes, the treatment plan is defined in continuous time, and the treatment is itself a continuous value. While there exist scenarios where this could be possible (e.g., in settings where treatment is constantly administered), most settings violate this. Hence, we need to reconcile modelling treatments in continuous time and values to settings violating these basic setups.*

To connect ODE discovery with the treatment effects literature, we require incorporating different types of treatment plans. Continuous valued treatment that is administered over time is adopted quite naturally in a differential equation. However, the treatment effects literature is typically focused on other types of treatment (Bica et al., 2021): binary treatments, categorical treatments, multiple simultaneous treatments, and dosage. Each of these treatments can be static (i.e., constant throughout the trajectory) or dynamic. With such diverse treatments, it might be impossible to express $F$ as a closed-form expression. For instance, where the treatment is a categorical variable (i.e., $\mathcal{A} = \{1, \ldots, k\}$). This violates assum. 3.3 of continuous solutions made by ODE discovery methods.

To reconcile it, we need to decide how to incorporate treatment $a$ into $F$. This can be done in two ways: either we discover different (piecewise) closed-form ODEs for different treatments (Trefethen et al., 2017; Jianwang et al., 2021), or we incorporate the action variable $a$ into the closed-form ODE. Depending on the type of treatment one or both of these approaches can be chosen. In appendix E we outline the ways of incorporating the treatment plan $a$ in $F$ according to the treatment types listed above. In summary, there are 4 different treatment types: binary, categorical, multiple, and continuous treatments (Bica et al., 2020b). Each of them can be a static or dynamic treatment, which is either constant or changes during a trajectory, respectively. This results in 8 scenarios, which we model in two ways: either the ODE changes for each treatment option; or the treatment is part of the starting condition. These are all outlined in table 6 in appendix E.

In section 6 we demonstrate the effectiveness of discovering ODEs per categorical treatment.

> **Framework step 2:** *Incorporate treatment plans as dynamical systems using appendix E.*

### 4.3 MODELLING BETWEEN-SUBJECT VARIABILITY (DISCR. 3)

**Discrepancy 3 (Between-subject variability.)** *In a classical ODE discovery setting, we usually have only one kind of between-subject variability corresponding to a noisy measurement - residual unexplained variability (RUV). However, there are a few other sources of variability that need to be accounted for before considering it as an ODE discovery problem.*

Table 2: **Layers of Between-Subject Variability (BSV).** We show four levels of between-subject variability and explain the different sources of where variability is coming from. In the table, we have: ODE (i), RUV (ii), Covariates (iii), Parameter distributions (iv) and $q(a) = aw_0 + w_1$, detailed in appendix F

| | (i) | (ii) | (iii) | (iv) | Causal graph | Example $y(t)$ | Parameters (eq. (5)) |
|---|---|---|---|---|---|---|---|
| | ODE. | +RUV | +Cov. | +Dist | | | |
| A | ✓ | ✗ | ✗ | ✗ | $\overset{\curvearrowright}{X} \rightarrow Y$ | $x(t)$ | $C_0 = c_0, C_1 = c_1$ |
| B | ✓ | ✓ | ✗ | ✗ | $\overset{\curvearrowright}{X} \rightarrow Y \leftarrow \epsilon$ | $x(t) + \epsilon$ | $C_0 = c_0, C_1 = c_1$ |
| C | ✓ | ✓ | ✓ | ✗ | $V \rightarrow \overset{\curvearrowright}{X} \rightarrow Y \leftarrow \epsilon$ | $x(t) + \epsilon$ | $C_0 = q(c_0), C_1 = q(c_1)$ |
| D | ✓ | ✓ | ✓ | ✓ | $V \overset{\epsilon}{\rightarrow} P \overset{\epsilon}{\rightarrow} \overset{\curvearrowright}{X} \rightarrow Y$ | $x(t) + \epsilon$ | $C_0 \sim \mathcal{N}(q(c_0), \sigma_0), C_1 \sim \mathcal{N}(q(c_1), \sigma_1)$ |

As pharmacological models play a prominent role in treatment effect literature (Geng et al., 2017; Bica et al., 2020b; Berrevoets et al., 2021; Lim, 2018; Seedat et al., 2022) we employ formalism from pharmacology literature to discuss between-subject variability (BSV).

There are two types of BSV (Mould & Upton, 2012): *(i) Unexplained variability* which includes RUV and parameter distributions; and *(ii) Explained variability* includes covariate models where we model the impact of static covariates on the equation's parameters.

We base the model of our causal dynamical system on the formalism introduced in Peters et al. (2022). In particular, they use the term *Deterministic* Casual Kinetic Model (DCKM), which can be summarised as a system of first-order autonomous ODEs. This is the simplest pharmacological model with no BSV (setting **A** in Table 2). In light of realism, however, we can add different layers of BSV, making the model more complex and thus more difficult to discover by current methods.

Table 2 outlines different types of BSV as graphical models, each with increasing complexity. In section 6, we relate the parameter columns in table 2 to the underlying ground-truth equations used in our experiments. We now detail these layers in increasing complexity.

**B: RUV - noisy measurements.** Noisy measurements are one of the biggest challenges of ODE discovery (Brunton et al., 2016). This is because noisy measurements cause derivative estimation to be very challenging. Recently, methods based on the weak formulation of ODEs (Qian et al., 2022; Messenger & Bortz, 2021) have managed to circumvent the derivative estimation step, making them more robust to noisy measurements. We represent DCKMs with noisy measurements graphically similar to the standard DCKM but will now explicitly add noise-related nodes (table 2 row B).

**C: Covariate models.** One way we can incorporate explainable variability into the model is by modelling the impact of the covariates on the model parameters. The covariates are incorporated into the model by closed-form expressions. Usually simple ones such as linear, exponential, or power (Mould & Upton, 2012). Sometimes other equations are used, such as complex closed-form expressions or piece-wise functions (Chung et al., 2021). We add a node corresponding to static covariates to represent covariate models graphically (table 2 row C).

**D: Parameter distributions.** Although the covariate models let us calculate the *group parameters*—parameters for a specific group of people based, e.g., on their age or weight, the actual parameter for an individual might deviate from this value in some random way. We can model this by assuming a distribution of parameter values with its mean depending on the covariates. We can depict this graphically by adding the nodes corresponding to the parameters and the associated noise nodes coming into them. Similarly, ODE discovery methods are not designed to discover such models (table 2 row D).

> **Framework step 3:** *Decide on the BSV using table 2 and, if type D, adjust as in section 5.*

## 5 INSITE —THE FRAMEWORK IN PRACTICE

Having introduced our general framework of applying ODE discovery techniques to treatment effects inference (in section 4), we now turn to apply it. Following the framework steps (FS) as described in section 4, we have to: (FS1) accept a new set of assumptions (discr. 1), (FS2) incorporate treatment plans (discr. 2), and (FS3) decide on the BSV type (discr. 3). For our purposes, we use SINDy (Brunton et al., 2016), one of the most widely adopted and used methods in ODE discovery. One can use any underlying ODE discovery methods as long as they: (1) model ODEs that include a set of numeric constants ($\beta \in \mathbb{R}^m$, with $m$ numeric constants), and (2) we accept that BSV is modelled

---

**Algorithm 1** Individualized Nonlinear Sparse Identification Treatment Effect (INSITE)

---

 1: **Input:** Patient data $\mathcal{D}$ cfr. section 2; deterministic DE discovery method, `DE`.
 2: **Output:** Patient-specific DEs $\boldsymbol{F}^{(i)}$; population DEs $\bar{\boldsymbol{F}}$.
 3: $\bar{\boldsymbol{F}} \leftarrow$`DE`$(\mathcal{D})$            ▷ **Step 1:** *Discover Population Differential Equation*
 4: ***for*** *patient* $(i)$ *in* $\mathcal{D}$ ***do***        ▷ **Step 2:** *Discover Patient-Specific Differential Equations*
 5:      *Fine-tune numeric constants in* $\bar{\boldsymbol{F}}$ *using patient* $(i)$ *eq.* (4)
 6:      *Obtain patient-specific equation* $\boldsymbol{F}^{(i)}$
 7: ***return*** *Patient-specific* $\boldsymbol{F}^{(i)}$ *and* $\bar{\boldsymbol{F}}$

---

through these numeric constants alone, i.e., two varying subjects can only differ by having different model numeric constants.[4]

*Individualized Nonlinear Sparse Identification Treatment Effect* (INSITE) consists of two main steps, reminiscent of the remaining discr. 2 and 3: (1) discover the *population* (global) differential equation $\bar{\boldsymbol{F}}$ for all patients, and (2) discover the *patient-specific* differential equation $\boldsymbol{F}^{(i)}$ by fine-tuning the population equation's numeric constants—as outlined in algorithm 1. Having a set of numeric constants (one for each patient) we derive a population differential equation where each individual's numeric constants are represented by a sample from this population differential equation distribution.

◆ **Step 1 (cfr. framework step 2):** *Discovering the Population Differential Equation.* Given an observed dataset $\mathcal{D}$ of patients, we aim to discover the population differential equation governing the patient covariates' interaction with their potential outcomes. We can use any deterministic differential equation discovery method such as SINDy (Brunton et al., 2016) to recover the population ODE $\bar{\boldsymbol{F}}$. We adapt it to handle treatments as outlined in section 4.2 and appendix E (as per discr. 2), where for each separate $k$ categorical (or binary) treatment, we discover a separate ODE—which is only active for that categorical treatment, i.e., $\bar{\boldsymbol{F}} = \{\bar{\boldsymbol{F}}_1, \ldots, \bar{\boldsymbol{F}}_k\}$.

◆ **Step 2 (cfr. framework step 3):** *Discovering Patient-Specific Differential Equations.* Once the population differential equation is discovered, we then fine-tune the numeric constants in the population equation to obtain patient-specific differential equations $\boldsymbol{F}^{(i)}$. By keeping the same functional form but allowing unique numeric constants for each patient to be refined, we can model the *individualized* treatment effect and account for patient heterogeneity (between-subject variability). Crucially, to avoid overfitting and allow only small deviations away from the initial population numeric constants $\bar{\boldsymbol{\beta}}$, we fit the observed patient trajectory up to the current time by minimizing,

$$\mathcal{L}(\boldsymbol{\beta}^{(i)}) = \frac{1}{T^{(i)}} \|\boldsymbol{Y}^{(i)} - \hat{\boldsymbol{Y}}(\boldsymbol{X}^{(i)}, \boldsymbol{A}^{(i)}, \boldsymbol{\beta}^{(i)})\|_2^2 + \lambda \|\bar{\boldsymbol{\beta}} - \boldsymbol{\beta}^{(i)}\|_2^2, \tag{4}$$

a Mean Squared Error (MSE) of the predicted trajectory evolution $\hat{\boldsymbol{Y}}$ for the observed trajectory. Additionally, we also require a regularization term in eq. (4). Specifically, for each individual trajectory, $(i)$, we refine the population set of parameters $(\bar{\boldsymbol{\beta}})$ to obtain $\boldsymbol{\beta}^{(i)}$ while still using them to regularize to prevent overfitting. The latter is an insight borrowed from transfer learning (Tommasi et al., 2010; Tommasi & Caputo, 2009; Takada & Fujisawa, 2020). Interestingly, without such regularization, we find INSITE to underperform significantly (cfr. appendix K.1). We use Broyden–Fletcher–Goldfarb–Shanno (BFGS) algorithm (Fletcher, 2013) to minimize eq. (4) as is standard in symbolic regression (Petersen et al., 2020), and provide full inference details in appendix G.1.

**Parameter distributions.** Unique to our method is that after obtaining the patient-specific differential equations, we can derive a population differential equation where each numeric constant is represented by a distribution, such as a normal distribution or a mixture of distributions—recovering a probabilistic interpretation of the underlying data generating process. This is explored in appendix K.2.

## 6   EXPERIMENTS AND EVALUATION

To allow for a robust and systematic comparison of longitudinal treatment effect (LTE) methods and ODE discovery techniques, we create a synthetic testbed designed for treatment effect prediction

---

[4]Essentially, we cannot have a mathematical expression changed across subjects (such as an $\exp$ changed to a $\log$. In fact, a scenario such as this would violate assum. 3.3).

across different synthetic datasets generated from different classes of pharmacological models. We note that using synthetic datasets is common in benchmarking LTE methods, as for a real dataset the counterfactual outcomes are unknown (Lim, 2018; Bica et al., 2020b). Our testbed consists of:

**Diverse underlying pharmacological models $F$.** The synthetic datasets are generated by sampling a pharmacological model $F$ with a given treatment assignment policy. To ensure robustness across diverse pharmacological scenarios, we include the model classes outlined in section 4.3, from A to D—which differ between noise, static covariates, and parametric distributions of parameters. We analyze two standard Pharmacokinetic-Pharmacodynamic (PKPD) models for each model class (section 4.3). First, a one-compartmental PKPD model (eq. (5)) with a binary static action. Second, a state-of-the-art biomedical PKPD model of tumor growth, used to simulate the combined effects of chemotherapy and radiotherapy in lung cancer (Geng et al., 2017) (eq. (6))—this has been extensively used by other works (Seedat et al., 2022; Bica et al., 2020b; Melnychuk et al., 2022). Here, to explore continuous types of treatments, we use a continuous chemotherapy treatment $c(t)$ and a binary radiotherapy treatment $d(t)$, both changing over time. For both models $y = x$, is the volume of the tumor $t$ days after diagnosis, modeled separately as:

$$\frac{dx(t)}{dt} = \begin{cases} -\frac{C_0}{V}x(t), & \text{if } a = 0 \\ -\frac{C_1}{V}x(t), & \text{if } a = 1 \end{cases} \tag{5}$$

$$\frac{dx(t)}{dt} = \Big( \underbrace{\rho \log\left(\frac{K}{x(t)}\right)}_{\text{Tumorgrowth}} - \underbrace{\beta_c C(t)}_{\text{Chemotherapy}} - \underbrace{(\alpha_r d(t) + \beta_r d(t)^2)}_{\text{Radiotherapy}} + \underbrace{e_t}_{\text{Noise}} \Big) x(t) \tag{6}$$

where the parameters $C_0, C_1, V, \rho, K, \gamma, \alpha, \beta, e_t$ are sampled according to the different layers of between-subject variability (table 2) forming variations of A-D, with parameter distributions following that as described in Geng et al. (2017) or otherwise detailed in appendix F. We also compare to the standard implementation of eq. (6) (labelled as **Cancer PKPD**) to ensure standard comparison to existing state-of-the-art methods, where the treatments are both binary. We further detail dataset generation for each in appendix F.

**Action assignment policy.** We introduce time-dependent confounding by making the treatment assignment vary from a purely random treatment assignment to purely deterministic based on a threshold of the outcome predictor value, controlled by a scalar $\gamma \in \mathbb{R}_+$—such that $\gamma = 0$ corresponds to no time-dependent confounding and larger values correspond to increasing time-dependent confounding. Further details of this treatment policy are in appendix F.

**Benchmark methods**. We seek to compare against the existing state-of-the-art (SOTA) methods from (1) longitudinal treatment effect models and (2) ODE discovery. First, longitudinal treatment effect models often consist of black-box neural network-based approaches, i.e., not closed-form or easily interpretable. A significant theme in these works is the development of methods to mitigate time-dependent confounding. Many mitigation methods have been proposed that we use as benchmarks, such as propensity networks in Marginal Structural Models (**MSM**) (Robins et al., 2000) and Recurrent Marginal Structural Networks (**RMSN**) (Lim, 2018), Gradient Reversal in Counterfactual Recurrent Networks (**CRN**) (Bica et al., 2020b), Confusion in Causal Transformers (**CT**) (Melnychuk et al., 2022) and g-computation in G-Net (**G-Net**) (Li et al., 2021). Second, ODE discovery methods aim to discover an underlying closed-form mathematical equation $F$ that best fits the underlying controlled ODE of the observed state and action trajectories dataset, $\mathcal{D}$. In these methods, the state derivative is unobserved; therefore, they have to approximate the derivative using finite differences, as in Sparse Identification of Nonlinear Dynamics (Brunton et al., 2016), or employ a variational loss as the objective, as in Weak Sparse Identification of Nonlinear Dynamics (Messenger & Bortz, 2021)[5]. To make these two ODE discovery methods more competitive, we adapt them to model individual ODEs per categorical treatment (section 4.2), (termed **A-SINDy**, **A-WSINDy** respectively). We further discuss why we chose these SOTA methods and their implementation details in appendix G.

**Evaluation.** We evaluate against the standard longitudinal treatment effect metrics of test counterfactual $\tau$-step ahead prediction normalized root mean squared error (RMSE), where $\tau = \{1, 2, 3, 4, 5, 6\}$, for a sliding treatment plan (Melnychuk et al., 2022). Unless otherwise specified, each result is run for five random seed runs with time-dependent confounding of $\gamma = 2$—see appendix H for details.

---

[5]We only include results for WSINDy when it is possible to use it, as it does not support sparse short trajectories. Detailed further in appendix G.

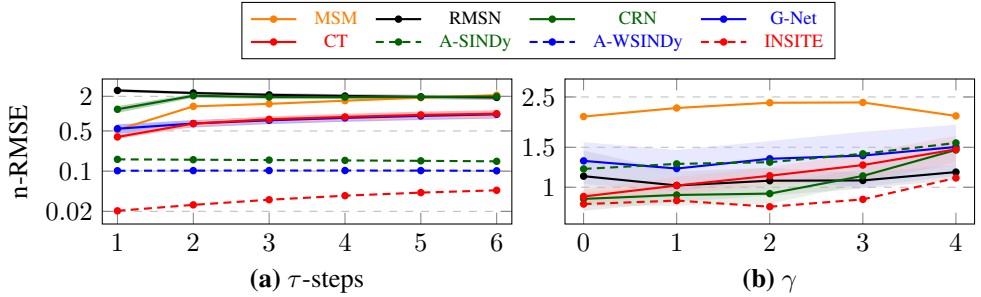

Figure 3: **(a) Counterfactual $\tau$-step ahead prediction error ($\gamma = 2$)**, on eq. (5).D from table 3. **(b) Counterfactual 6-step ahead prediction error (for increasing time-dependent confounding, $\gamma$)**, on the standard Cancer PKPD dataset. INSITE maintains a low normalized RMSE (high performance) across long time horizons and increasing time-dependent confounding. Further results are in appendix K.

Table 3: **Counterfactual normalized RMSE,** for 6-step ahead prediction of the benchmarks on each synthetic dataset detailed in appendix F. We quote 95% confidence intervals with each value, and all metrics are averaged over ten random seed runs. Each PKPD underlying pharmacological model is simulated with different layers of BSV (table 2) for A-D. Other $\tau$-step results are in the appendix K.3. Our contribution is shaded below.

| | Method | eq. (5).A | eq. (5).B | eq. (5).C | eq. (5).D | eq. (6).A | eq. (6).B | eq. (6).C | eq. (6).D | Cancer PKPD |
|---|---|---|---|---|---|---|---|---|---|---|
| LTE | MSM | 0.99±8.37e-17 | 0.99±0.00 | 0.97±8.37e-17 | 2.09±0.00 | 2.55±0.13 | 2.55±0.13 | 2.06±0.16 | 2.11±0.04 | 2.30±0.12 |
| | RMSN | 1.92±0.24 | 1.94±0.23 | 1.68±0.19 | 1.91±0.18 | 1.23±0.15 | 1.25±0.15 | 1.10±0.18 | 1.10±0.11 | 1.04±0.17 |
| | CRN | 1.05±0.10 | 1.05±0.10 | 0.82±0.09 | 1.98±0.14 | 1.05±0.03 | 1.05±0.03 | 1.03±0.08 | 1.03±0.10 | 0.92±0.08 |
| | G-Net | 0.91±0.20 | 0.91±0.20 | 0.72±0.14 | 0.97±0.15 | 1.33±0.27 | 1.34±0.27 | 1.02±0.11 | 1.25±0.15 | 1.22±0.14 |
| | CT | 0.90±0.18 | 0.90±0.18 | 0.75±0.14 | 1.00±0.14 | 1.29±0.07 | 1.29±0.10 | 1.03±0.11 | 1.14±0.10 | 1.07±0.07 |
| ODE-D | A-SINDy | 0.11±0.00 | 0.11±0.00 | 0.13±2.09e-17 | 0.15±0.00 | 1.45±0.03 | 1.45±0.03 | 1.40±0.01 | 1.51±0.09 | 1.23±0.13 |
| | A-WSINDy | 0.11±7.24e-05 | 0.11±2.49e-04 | 0.12±1.47e-03 | 0.10±7.61e-04 | NA | NA | NA | NA | NA |
| | INSITE | **0.02** ±2.62e-18 | **0.03** ±0.00 | **0.04** ±0.00 | **0.05** ±5.23e-18 | **0.94** ±0.05 | **0.94** ±0.05 | **0.84** ±0.04 | **0.87** ±0.08 | **0.79** ±8.37e-17 |

**Main results.** The test counterfactual normalized RMSE for 6-step ahead prediction for each benchmark dataset is tabulated in table 3. Our method, INSITE, achieves the lowest test counterfactual normalized RMSE across all methods. We provide additional experimental results in appendix K.

**Interpretable equations.** Unique to the proposed framework and INSITE, is that the discovered differential equation is fully interpretable; however, it relies on strong assumptions. Some of these discovered equations are shown in appendix J. It is clear that even with binary actions, INSITE is able to discover a useful equation that performs well and is similar in form to the true underlying equation, even discovering it exactly in some simple settings of eq. (5). The method can even do this in the presence of noise (BSV layers of B-D) and extrapolate well for future $\tau$-step predictions, fig. 3 (a).

**Model misspecification.** INSITE and the adapted ODE discovery methods (A-SINDy, A-WSINDy) are only correctly specified for eq. (5).A-D datasets, as they use the feature library of $\mathcal{L}_{\text{INSITE}} = \{1, x_0, x_1, x_0 x_1\}$. Crucially, the ODE discovery methods are misspecified for eq. (6).A-D, and the Cancer PKPD datasets, as their underlying equation would require the feature library of $\mathcal{L}_{\text{Cancer}} = \{1, x_0, x_1, x_0 x_1, x_0^2 x_1, x_0 x_1^2, \log(x_0), \log(x_1)\}$ to be correctly specified. Although this misspecification persists, as noticeable from the increased order of magnitude increase in error, we still observe INSITE achieves a lower error than the longitudinal treatment effect models (app. K.6).

**Relaxing the overlap assumption.** We can relax the overlap assumption (assum. 2.2) by increasing the time-dependent confounding of the treatment given, increasing $\gamma$. We observe, in fig. 3 (b), that INSITE can still discover a good approximating underlying equation even in the presence of high time-dependent confounding—hence where there is low overlap.

**INSITE Ablation.** INSITE's gain in performance derives from our framework's discr. 2, to discover individual ODEs per categorical treatment, as well as the ability to discover individualized (fine-tuned) ODEs. Notably, the improvement arising from the ability to discover individual ODEs per categorical treatment can be widely applied to existing ODE discovery methods using our framework for

Table 4: **INSITE Ablation**

| ODE per Cat. (section 4.2) | Fine Tuning | eq. (5).D 6-step n-RMSE |
|---|---|---|
| ✓ | ✓ | **0.05** ±5.23e-18 |
| ✗ | ✓ | 0.43 ±7.71e-17 |
| ✓ | ✗ | 0.15 ±0.00 |
| ✗ | ✗ | 0.87 ±0.00 |

treatment effects over time. Furthermore, we conduct additional insight experiments to evaluate the methods against datasets of smaller sample sizes and increasing observation noise in appendix K.

## 7 CONCLUSION AND FUTURE WORK

In conclusion, we presented a first framework that connects longitudinal heterogeneous treatment effects with ODE discovery methods, enabling improved interpretability and performance. Naturally, this is just the beginning. We hope that building on our framework (appendix A), the connection between treatment effects and ODE discovery is further solidified and perhaps extended to other types of differential equations and dynamical systems in general (see e.g. appendix C).

**Ethics statement**. This paper's novel approach of integrating ODE discovery methods into treatment effects inference can revolutionize precision medicine by enabling personalized, effective treatment strategies. However, misuse or application in inappropriate contexts could lead to incorrect treatment decisions. Moreover, despite the potential for individualized treatment, privacy concerns arise. Thus, proper data governance, clear communication of model limitations, and rigorous validation using diverse datasets are vital for responsible application.

Regarding limitations of our method, INSITE, and our framework in general, we list 3 major limitations one should take into account before applying our work in practice:

1. A correct set of candidate functions (tokens) is necessary for correct model recovery. To show the importance of this, we include experiments where we have wrongly specified this token library (see appendix K.6) and observe degrading performance as a result.

2. ODE discovery works best in sparse settings. The reason is two-fold: from a technical point of view, sparse equations are much less complex and simply easier to recover; from a usability point of view, the usefulness of non-sparse equations is limited as interpretability is negatively affected by non-sparse (or non-parsimonious) equations (Crabbe et al., 2020).

3. ODEs are noise free. Since we recover ODEs, the found equations do not model a source of noise as is typically the case in structural equation modelling. To model noise terms explicitly, our framework should be extended into stochastic DEs, as is stated in our future works paragraph above.

**Reproducibility statement**. We provide all code at `https://github.com/samholt/ODE -Discovery-for-Longitudinal-Heterogeneous-Treatment-Effects-Inf erence`. To ensure this paper is fully reproducible, we include an extensive appendix with all implementation and experimental details to recreate the method and experiments. These are outlined as the following: for benchmark dataset details, see appendix F; benchmark method implementation details, including those of the treatment effects baselines and adapted ODE discovery methods which include the proposed INSITE method see G; evaluation metric details see appendix H; dataset generation and model training see appendix I.

**Acknowledgements.** The authors would like to acknowledge and thank their corresponding funders, where SH is funded by AstraZeneca, JB is funded by W.D. Armstrong Trust, KK is funded by Roche, ZQ is funded by the Office of Naval Research (ONR). Moreover, we would like to warmly thank all the anonymous reviewers, alongside research group members of the van der Schaar lab (`www.vanderschaar-lab.com`), for their valuable input, comments, and suggestions as the paper was developed.

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

# Appendix

## Table of Contents

**Code.** All code is available at `https://github.com/samholt/ODE-Discovery-for-Longitudinal-Heterogeneous-Treatment-Effects-Inference`. We have a broader research group codebase at `https://github.com/vanderschaarlab/ODE-Discovery-for-Longitudinal-Heterogeneous-Treatment-Effects-Inference`.

## Author Contributions

**All authors** provided valuable contributions to the paper from the idea, writing, and editing, and managing the project's progress.

**KK** wrote the formalism in section 3, proposed (and described) how to adapt ODE discovery methods to include discrete treatment plans (section 4.2) and different layers of between-subject variability (section 4.3), constructed the initial experimental settings in table 8, and provided the idea for fig. 2.

**SH** wrote parts of the introduction, all the ODE assumptions in section 3; proposed and wrote the method in section 5, developed and wrote all the experiments in section 6, and the conclusion in section 7; designed and came up with the method INSITE; solely coded up the method and all baseline implementations, initial exploratory experiments for INSITE; all coding and experiment design, implementation and running for all nine datasets, seven baselines, evaluation; led all results within the paper in section 6, and eight additional new experimental settings and ablations of all the baselines across all datasets in appendix K; solely wrote 20 appendix sections, of additional experiments, ablations, explanations of key parts of the paper, additional related work, baseline descriptions, and synthetic dataset descriptions.

**JB** connected causal treatment effect estimation with ordinary differential equation discovery leading to the framework presented in this paper.

## A    Future Work

In our paper we have been very explicit about the settings in which we can use ODE discovery settings (see section 3 in particular). As these settings do not correspond one-to-one with standard TE settings, our paper can be viewed as an expansion of typical TE application areas. However, there is still more to be done! In particular, future endeavours could include: discovering stochastic differential equations, leveraging latent variable models for handling unobserved variables, learning piece-wise continuous ODE systems (Wang et al., 2023), and developing strategies to transfer population-level ODEs to other populations where there is limited data. Furthermore, INSITE could be considered a first link in connecting equation discovery over time with causality (Berrevoets et al., 2023), we hope many more works of this type will follow.

## B    Explanation of Treatment Intensity Processes and Filtrations in the Assumptions

The concept of treatment intensity processes has its roots in the generalization for treatment effects in continuous time, introduced in seminal works such as Lok (2008) and later embraced by Saarela & Liu (2016). Intensity processes serve as an extension of propensity scores, traditionally denoted as $p(T|X)$, to a continuous time framework.

In treatment effects literature, propensity scores provide the probability of treatment assignment conditioned on observed covariates. However, when dealing with continuous-time data, the traditional propensity scores pose a challenge. Specifically, propensities in this setting would govern entire treatment trajectories, considering not just the treatment at a specific time point $T$, but all potential future treatments. Given the vast space of potential trajectories in discrete time, this complexity becomes unmanageable in continuous time.

To address this challenge, filtrations are employed to restrict when a new treatment can be sampled. Filtrations essentially offer a controlled space for the random processes at play. Using the terminology from probability theory:

> For a stochastic process $(X_n)_{n \in \mathbb{N}}$, the filtration $\mathcal{F}_n := \sigma(X_k | k \leq n)$ is a $\sigma$-algebra. This means the filtration $\mathbb{F} = (\mathcal{F}_n)_{n \in \mathbb{N}}$ limits the stochastic process to only those random variables $X_i$ for which $i \leq n$, even though the original probability space might encompass variables with $i > n$.

Such a structure becomes increasingly important in continuous time applications, as it enables us to understand and manage the dependencies among observations over time. Furthermore, intensity

processes in this framework can be seen as analogous to well-known processes in other contexts, such as the Hawkes point processes (that model excitation) or the Poisson point processes (which assume no influence from past observations).

Now, relating back to assum. 2.2 (Overlap) and assum. 2.3 (Ignorability): The intensity process, represented by $\lambda(t|\mathfrak{F}_t)$, captures the propensity of an individual receiving treatment at time $t$ given all the information up to that point, encapsulated in the filtration $\mathfrak{F}_t$. The overlap assumption ensures that the intensity process is never deterministic, implying that there's always some randomness in treatment assignment irrespective of past information. On the other hand, the ignorability assumption implies that the intensity process conditioned on past information (up to time $t$) is the same as the intensity process that considers even the future outcomes beyond time $t$. This ensures that treatment assignment is independent of potential outcomes, thereby satisfying a key requirement for unbiased causal inference.

## C    EXTENDED RELATED WORK

### C.1    RELATED WORK APPROACHES

This research brings together two previously separated fields, namely temporal treatment effect estimation and ODE discovery, and there exists a plethora of works within each domain. Here, we will focus on the most relevant and recent ones.

**Treatment Effects over Time.** There has been substantial research on the problem of estimating treatment effects. One of the pioneering works in this area is the potential outcomes framework introduced by Neyman (1923) and further developed by Rubin (1980).

Traditional methods for estimating treatment effects include propensity score matching (Rosenbaum & Rubin, 1983), instrumental variables (Angrist et al., 1996; Angrist, 1991), and difference-in-differences (Athey & Imbens, 2006). However, these methods are primarily suited for static treatment settings and may have limitations when extended to handle treatment effects over (continuous) time.

The treatment effects literature is well represented in discrete time settings. With more traditional work stemming from epidemiology, based on g-computation, Structural Nested Models, Gaussian Processes, and Marginal Structural Models (MSMs) (Robins, 1994; Robins & Hernán, 2009; Xu et al., 2016; Robins et al., 2000); and more recent work relying on advances in deep learning architectures (Bica et al., 2020b; Melnychuk et al., 2022; Berrevoets et al., 2021; Lim, 2018). Some works have expanded the static treatment effect literature to the continuous time setting (Lok, 2008; Saarela & Liu, 2016; Ryalen et al., 2019), introducing new assumptions to help identify potential outcomes. Nevertheless, these works still face some critical challenges. For example, they often assume constant treatment effects and do not naturally handle irregular sampling and continuous treatments.

**ODE discovery.** The field of ODE discovery has also seen significant progress over the years. One of the primary goals of ODE discovery is to learn a model of a system's behaviour from observational data. This involves discovering the underlying system of ODEs that governs the system.

The methods that solve this problem range from linear ODE discovery techniques (Ljung, 1998) to more recent machine learning approaches like Recurrent Neural Networks (RNN) and Long Short-Term Memory (LSTM) (Hochreiter & Schmidhuber, 1997; Sherstinsky, 2020). However, these later methods often produce black-box models that lack the interpretability required for many applications, including those in regulated domains.

Recent efforts in the ML community have focused on finding interpretable models using techniques such as Sparse Identification of Nonlinear Dynamics (SINDy) (Brunton et al., 2016), which discovers sparse dynamical systems from data. However, traditional ODE discovery methods largely focus on homogeneous systems where the system dynamics are the same across different systems or individuals. They do not readily handle the heterogeneity of treatment effects (between-subject variability), which is a major concern in real-world applications like personalized medicine.

We consider ODE discovery to be a sub-field of the wider Symbolic Regression field, which is concerned with discovering a more general class of equation (beyond ODEs) (Camps-Valls et al., 2023; Mouli et al., 2023; Cranmer et al., 2020).

**Dynamical systems, ODEs, and treatment effects.** We want to explicitly state that, ODE discovery is a *subfield* of the broader area of dynamical systems identification. We believe this to be necessary as often work is presented as an identification method for *dynamical systems*, while they are only applicable to ODEs which is only one particular type of dynamical system. That means that works such as Schulam & Saria (2017); Hızlı et al. (2022); Noorbakhsh & Rodriguez (2022); Ryalen et al. (2020); Alaa & van der Schaar (2019); Qian et al. (2020); Soleimani et al. (2017) may appear related as they model treatment effects in *a type of* dynamical system. However, it is crucial to note that none of these works model treatment effects as an ODE, nor are they concerned with actually uncovering the underlying ground truth dynamical system in general. Instead, they model treatment effects as a parameterized dynamical system (such as, for example, a point process (Noorbakhsh & Rodriguez, 2022; Alaa & van der Schaar, 2019)), which is learned purely for accurate inference. In contrast, while we are also interested in accurate inference, the resulting model (the ODE) is additionally important.

**Fusing Treatment Effects and ODE discovery.** Our work is the first, to our knowledge, to combine these two fields for the problem of treatment effect estimation. We seek to harness the strengths of both fields to address their respective challenges, such as the lack of interpretability and robustness in the treatment effects literature and the assumptions about system behaviours in the ODE discovery community. In particular, our INSITE approach is unique in its ability to address the challenges from both communities and thus provides a promising new direction for these intertwined fields.

There are some ideas that *seem* to correspond, but only at first glance. For example Florens et al. (2008) connects control (a field which is very much related to ODE discovery), yet differs one some crucial points: (i) time is not considered; (ii) no equations are discovered; (iii) the assumption sets differ widely.

## C.2 RELATED WORK METHODS

**Longitudinal Treatment Effect Methods.** Longitudinal treatment effect methods focus on estimating the outcome of treatment at a given time $t$, often using black-box neural network-based approaches, i.e., not closed-form or easily interpretable. A significant theme in these works is the development of methods to mitigate time-dependent confounding. Many mitigation methods have been proposed that we use as benchmarks, such as propensity networks in Marginal Structural Models (**MSM**) (Robins et al., 2000) and Recurrent Marginal Structural Networks (**RMSN**) (Lim, 2018), Gradient Reversal in Counterfactual Recurrent Networks (**CRN**) (Bica et al., 2020b), Confusion in Causal Transformers (**CT**) (Melnychuk et al., 2022) and g-computation in G-Net (**G-Net**) (Li et al., 2021).

**ODE Discovery Methods.** ODE discovery methods aim to discover an underlying closed-form mathematical equation $F$ that best fits the underlying controlled ODE of the observed state and action trajectories dataset, $\mathcal{D}$. In these methods, the state derivative is unobserved; therefore, they have to approximate the derivative using finite differences, as in Sparse Identification of Nonlinear Dynamics (Brunton et al., 2016), or employ a variational loss as the objective, as in Weak Sparse Identification of Nonlinear Dynamics (Messenger & Bortz, 2021). Underlying these methods is the broader class of methods for symbolic regression (Holt et al., 2023b). To make these two ODE discovery methods more competitive, we adapt them to model individual ODEs per categorical treatment (section 4.2), (termed **A-SINDy**, **A-WSINDy** respectively).

**Other Related Approaches.** In addition to the aforementioned categories, several other methods have been proposed to model longitudinal data and estimate treatment effects. For instance, recurrent neural networks (RNNs) (Cho et al., 2014) and long short-term memory (LSTM) networks (Hochreiter & Schmidhuber, 1997) are commonly used for handling sequential data, as well as other sequential time series models (Holt et al., 2022), and they have been applied to estimate treatment effects in longitudinal settings (Lim, 2018). Gaussian processes (Rasmussen et al., 2006) and continuous-time Gaussian processes (Alvarez & Lawrence, 2009) have also been employed for modelling and learning from longitudinal data. Furthermore, counterfactual regression (Johansson et al., 2016) and doubly robust estimation (Funk et al., 2011) are statistical methods used to estimate causal effects in observational studies, and they can be adapted to handle time-dependent confounding in longitudinal settings. Additionally, once a model is learned, it can be used for optimal planning (Holt et al., 2023a; 2024), and such time-series models can also applied to other related domains such as in finance (Jha et al., 2020).

A more detailed comparison of the related approaches discussed above can be seen in table 5. This table compares various aspects of these methods, such as interpretability, between-subject variability, continuous-time modelling, robustness to observation noise and the ability to handle categorical and continuous treatments, and data size requirements.

Table 5: **Comparison of related works.** Columns: Interpretable?—can it provide a closed-form equation? BSV?—can between-subject variability be modeled? Continuous-time?—is it a continuous-time model (i.e., able to learn from irregularly sampled observation and treatment trajectories naturally)? Noise?—is it robust to observation noise? Categorical $a \in \mathbb{Z}$?—can it model categorical treatments (inclusive of binary treatments)? Continuous $a \in \mathbb{R}$?—can it model continuous treatments?

| Approach | Ref | Interpretable? | BSV? | Continuous-time? | Noise? | Categorical $a \in \mathbb{Z}$? | Continuous $a \in \mathbb{R}$? |
|---|---|---|---|---|---|---|---|
| **Longitudinal Treatment Effect Methods** | | | | | | | |
| MSM | (Robins et al., 2000) | ✗ | ✓ | ✗ | ✓ | ✓ | ✗ |
| RMSN | (Lim, 2018) | ✗ | ✓ | ✗ | ✓ | ✓ | ✗ |
| CRN | (Bica et al., 2020b) | ✗ | ✓ | ✗ | ✓ | ✓ | ✗ |
| G-Net | (Li et al., 2021) | ✗ | ✓ | ✗ | ✓ | ✓ | ✗ |
| CT | (Melnychuk et al., 2022) | ✗ | ✓ | ✗ | ✓ | ✓ | ✗ |
| **ODE discovery Methods** | | | | | | | |
| A-SINDy | (Brunton et al., 2016) | ✓ | ✗-population only | ✓ | ✗ | ✓ | ✓ |
| A-WSINDy | (Messenger & Bortz, 2021) | ✓ | ✗-population only | ✓ | ✓ | ✓ | ✓ |
| INSITE | (Ours) | ✓ | ✓ | ✓ | ✓ | ✓ | ✓ |

# D  ODE FORMALISM FOR TREATMENT EFFECT ESTIMATION

In this section, we want to justify our ODE formalism for treatment effect estimation, eq. (3), and explain its generality.

We assume that the treatment effect is a deterministic function of the modelled covariates (often the identity). This is inspired by examples considered in treatment effect papers, where we equate the treatment effect to the tumour volume or the response to a drug given a particular concentration (through a dose-response curve) (Bica et al., 2020b; Berrevoets et al., 2021; Lim, 2018; Seedat et al., 2022).

We want to emphasize that a direct link between $a$ and $y$ can be included through extended covariates $x'(t) = (x(t), y(t))$ and the function that links covariates to the outcome as $g(x') = g(x, y) = y$. Then the new treatment effect variable is defined as $y'(t) = g(x'(t)) = y(t)$. This is the same as treating one of the covariates as the outcome. Please note that our formalism extends this setting and allows for a more complicated treatment outcome that depends on several covariates.

# E  TREATMENTS

In the following, we summarize the ways of incorporating the treatment plan $a$ in $F$ according to the treatment types of binary treatments, categorical treatments, multiple simultaneous treatments, and continuous treatments. This is necessary so that we can simplify the complex and possibly not closed-form $F$ into simpler closed-form $f$'s that we can discover using current methods. In this subsection, every function denoted by the letter $f$ or $f$ is assumed to be closed-form.

As described in Bica et al. (2021) there are four main types of treatments considered in the treatment effect estimation literature

- Binary treatment (treatment or no treatment)
- Categorical treatment (one treatment out of multiple possible treatment options)
- Multiple treatments assigned simultaneously
- Single treatment or multiple treatments with associated dosage

Each of these treatments can be either static (constant throughout the trajectory) or dynamic. As we are interested in the discovery of closed-form ODEs, the treatment can be incorporated into $F$ in two ways.

- We learn different closed-form ODEs for different treatments

- We incorporate the action variable $\boldsymbol{a}$ into the closed-form ODE.

Depending on the type of treatment one or both of these approaches can be chosen. We summarize the ways of treating $\boldsymbol{a}$ in $\boldsymbol{F}$ based on the type of treatment in Table 6.

**Binary treatment** considers only two kinds of treatments (usually "treatment" and "no treatment"). The action trajectory is one-dimensional with values in $\{0, 1\}$. In a static setting, $a$ is constant, in a dynamic setting it is piece-wise constant—that means that we are allowed to change the treatment throughout the trajectory. This kind of action can be incorporated in $\boldsymbol{F}$ in two ways. We can either learn two different closed-form systems of ODEs ($\boldsymbol{f}_0$ and $\boldsymbol{f}_1$) or incorporate $a$ directly in the equation, e.g., $\boldsymbol{f}_0 + a(t)\boldsymbol{f}_1$.

**Categorical treatment** considers one treatment out of multiple possible treatments. The action trajectory is one-dimensional with values in $[1, K]$. As previously, $a$ is constant in a static setting and piece-wise constant in a dynamic setting. We no longer can easily incorporate $a$ into the closed-form expression (as it attains $K$ discrete values), so the only option is to learn $K$ separate closed-form systems of ODEs $\boldsymbol{f}_1, \ldots, \boldsymbol{f}_K$.

**Multiple treatments** considers scenarios where multiple treatments can be assigned simultaneously. We represent $\boldsymbol{a}$ as a vector $\{0, 1\}^K$, where $a_i(t) = 1$ if the $i^{\text{th}}$ treatment was assigned at time $t$. As previously, $a$ is constant in a static setting and piece-wise constant in a dynamic setting. We can consider this case as just having $2^K$ treatments (all possible subsets) and reduce to a categorical treatment where we learn $2^K$ separate equations. An alternative approach would be to learn separate terms for each treatment and combine them using a principle of superposition (Thron, 1974). We can then represent the system of ODEs at time $t$ as $\sum_{i=1}^{K} a_i(t)\boldsymbol{f}_i(\boldsymbol{x}(t), \boldsymbol{v})$. However, no current ODE discovery algorithm leverages this representation.

**Continuous treatment** is usually considered when we want to model the dosage or the strength of the treatment. $\boldsymbol{a}(t)$ is represented as a $K$-dimensional real vector where $a_i(t)$ is the dose/strength of the $i^{\text{th}}$ treatment. In a static setting, we consider $a$ to be constant. In a dynamic setting, we consider it to be continuous. As $a_i(t)$ can have any real value, the only way to incorporate it into the equations is to have it directly in the closed-form ODE $\boldsymbol{f}(\boldsymbol{x}(t), \boldsymbol{v}, \boldsymbol{a}(t))$.

Table 6: **How different treatment types can be represented in an ODE formalism.** Here S/D corresponds to a static or dynamic treatment.

| Treatment | S/D | Domain of $\boldsymbol{a}$ | Constant | $\boldsymbol{F}(\boldsymbol{x}(t), \boldsymbol{v}, \boldsymbol{a}(t))$ |
|---|---|---|---|---|
| Binary | S | $a(t) \in \{0, 1\}$ | Yes | $\boldsymbol{f}_{a(t)}(\boldsymbol{x}(t), \boldsymbol{v})$ or |
| | D | | Piece-wise | $\boldsymbol{f}_0(\boldsymbol{x}(t), \boldsymbol{v}) + a(t)\boldsymbol{f}_1(\boldsymbol{x}(t), \boldsymbol{v})$ |
| Categorical | S | $a(t) \in [1, K]$ | Yes | $\boldsymbol{f}_{a(t)}(\boldsymbol{x}(t), \boldsymbol{v})$ |
| | D | | Piece-wise | |
| Multiple | S | $\boldsymbol{a}(t) \in \{0, 1\}^K$ | Yes | $\boldsymbol{f}_{\boldsymbol{a}(t)}(\boldsymbol{x}(t), \boldsymbol{v})$ or |
| | D | | Piece-wise | $\sum_{i=1}^{K} a_i(t)\boldsymbol{f}_i(\boldsymbol{x}(t), \boldsymbol{v})$ |
| Continuous | S | $\boldsymbol{a}(t) \in \mathbb{R}^K$ | Yes | $\boldsymbol{f}(\boldsymbol{x}(t), \boldsymbol{v}, \boldsymbol{a}(t))$ |
| | D | | No | |

## F BENCHMARK DATASET DETAILS

In the following we outline the standard Cancer PKPD equation, then outline the between-subject variability layers that we use to generate the different equation classes from the two base PKPD equation models. We also provide the parameter distributions used to generate the different equation classes. For dataset generation and model training see appendix I.

### F.1 CANCER PKPD

This is a state-of-the-art biomedical Pharmacokinetic-Pharmacodynamic (PKPD) model of tumour growth, used to simulate the combined effects of chemotherapy and radiotherapy in lung cancer (Geng et al., 2017) (eq. (7))—this has been extensively used by other works (Seedat et al., 2022; Bica et al., 2020b; Melnychuk et al., 2022). Specifically, this models the volume of the tumour $x(t)$ for days $t$ after the cancer diagnosis—where the outcome is one-dimensional. The model has two binary treatments: (1) radiotherapy $a_t^r$ and (2) chemotherapy $a_t^c$.

$$\frac{dx(t)}{dt} = \Big( \underbrace{\rho \log \Big( \frac{K}{x(t)} \Big)}_{\text{Tumor growth}} - \underbrace{\beta_c C(t)}_{\text{Chemotherapy}} - \underbrace{(\alpha_r d(t) + \beta_r d(t)^2)}_{\text{Radiotherapy}} + \underbrace{e_t}_{\text{Noise}} \Big) x(t) \tag{7}$$

Where the parameters $K, \rho, \beta_c, \alpha_r, \beta_r$ for each simulated patient are sampled distributions detailed in Geng et al. (2017), which are also described in table 7. Here $e_t \sim \mathcal{N}(0, 0.01^2)$ is random noise, modelling randomness in the tumour growth.

Table 7: **Cancer PKPD parameter distributions.**

| Model | Variable | Parameter | Distribution | Parameter Value ($\mu, \sigma$)) |
|---|---|---|---|---|
| Tumor growth | Growth parameter | $\rho$ | Normal | $7.00 \times 10^{-5}, 7.23 \times 10^{-3}$ |
| | Carrying capacity | $K$ | Constant | 30 |
| Radiotherapy | Radio cell kill ($\alpha$) | $\alpha_r$ | Normal | 0.0398, 0.168 |
| | Radio cell kill ($\beta$) | $\beta_r$ | - | Set s.t. $\alpha/\beta$=10 |
| Chemotherapy | Chemo cell kill | $\beta_c$ | Normal | 0.028, 0.0007 |

Furthermore, we incorporate heterogeneous responses, following Bica et al. (2020b); Lim (2018); Melnychuk et al. (2022), where the means are modified for $\beta_c$ and $\alpha_r$ by creating three groups of patients (i.e., to represent three types of patients with heterogeneity in treatment response).

For patient group 1, we modify the mean of $\alpha_r$ so that $\mu(\alpha_r) = 1.1 \times \mu(\alpha_r)$ and for patient group 3, we modify the mean of $\alpha_c$ so that $\mu(\alpha_c) = 1.1 \times \mu(\alpha_c)$.

Additionally, the chemotherapy drug concentration $c(t)$ follows an exponential decay relationship with a half-life of one day:

$$\frac{dc(t)}{dt} = -0.5c(t) \tag{8}$$

where the chemotherapy binary action represents increasing the $c(t)$ concentration by 5.0mg/m$^3$ of Vinblastine given at time $t$.

Whereas the radiotherapy concentration $d(t)$ represents $2.0Gy$ fractions of radiotherapy given at timestep $t$, where Gy is the Gray ionizing radiation dose.

**Time-dependent confounding.** We introduce time-varying confounding into the data generation process. This is accomplished by characterizing the allocation of chemotherapy and radiotherapy as Bernoulli random variables. The associated probabilities, $p_c$ and $p_r$, are determined by the tumor diameter as follows:

$$p_c(t) = \sigma \left( \frac{\gamma_c}{D_{\max}} (\bar{D}(t) - \delta_c) \right) \qquad p_r(t) = \sigma \left( \frac{\gamma_r}{D_{\max}} (\bar{D}(t) - \delta_r) \right), \tag{9}$$

where $D_{\max} = 13$cm represents the largest tumor diameter, $\theta_c = \theta_r = D_{\max}/2$ and $\bar{D}(t)$ signifies the mean tumor diameter. The parameters $\gamma_c$ and $\gamma_r$ manage the extent of time-varying confounding. Higher values of $\gamma_{c,r}$ amplify the influence of this confounding factor over time.

### F.2 SYNTHETIC EQUATION CLASSES

We now outline the two synthetic equations, which we then use in the four layers of between-subject variability settings A-D table 2—which differ between noise, static covariates, and parametric distributions of parameters. The two synthetic equations we study are two standard PKPD models,

the first being a one-compartmental PKPD model with a binary static action. The second is the same state-of-the-art biomedical PKPD model of tumor growth from the Cancer PKPD dataset (Geng et al., 2017), where the model has a continuous chemotherapy treatment $c(t)$ and a binary radiotherapy treatment $d(t)$. Therefore the volume of the tumor $t$ days after diagnosis for each model is given by:

$$\frac{dx(t)}{dt} = \begin{cases} -\frac{C_0}{V}x(t), & \text{if } a = 0 \\ -\frac{C_1}{V}x(t), & \text{if } a = 1 \end{cases} \tag{10}$$

$$\frac{dx(t)}{dt} = \Big( \underbrace{\rho \log\left(\frac{K}{x(t)}\right)}_{\text{Tumorgrowth}} - \underbrace{\beta_c C(t)}_{\text{Chemotherapy}} - \underbrace{(\alpha_r d(t) + \beta_r d(t)^2)}_{\text{Radiotherapy}} + \underbrace{e_t}_{\text{Noise}} \Big) x(t) \tag{11}$$

where the parameters $C_0, C_1, V, \rho, K, \gamma, \alpha, \beta, e_t$ are sampled according to the different layers of between-subject variability (table 2) forming variations of A-D, following the setup in table 8.

Specifically, for the one-compartmental PKPD model of eq. (10), $c_0 \sim \mathcal{N}(0.5, 0.05)$, $c_1 \sim \mathcal{N}(0.5, 0.05)$, $V = 1$ and has additive observation noise of $\epsilon \sim \mathcal{N}(0, 0.01)$. Moreover, $w_0 = 1.0, w_1 = 0.05, w_2 = 1.0, w_3 = 0.15$.

For the tumour growth PKPD model, eq. (11), we ablate the full model to create a homogeneous version of one patient type ($v_{\text{Patient Type}} = 1$) for the no noise BSV layer (A) and noise BSV layer (B), then reintroduce patient heterogeneity ($v_{\text{Patient Type}} \in \{1, 2, 3\}$) and restrict $\beta_c$ to only be the mean value for the BSV layer of covariates (C). Finally, the full BSV layer (D) of noise, covariates and parametric distribution of parameters is the full Cancer PKPD model. Here unless otherwise noted the other parameters are still sampled from their respective defined distributions, as outlined in table 7. Here we follow the same treatment assignment policy as the Cancer PKPD model above, and as the LTE methods are only designed to handle binary treatments, we provide the LTE methods with the binary treatment assignments, and the corresponding ODE discovery methods with the continuous value of $c(t)$ and the binary value of $d(t)$.

**Time-dependent confounding** Similarly to the Cancer PKPD model, we also introduce time-dependent confounding in the data-generating process, following a similar setup. We achieve this by characterizing each treatment assignment as Bernoulli random variables. Each associated probability $p$ depends on the outcome normalized value. Here, we can vary the scalar value $\gamma \in [0, \inf)$ to increase time-dependent confounding.

$$\begin{aligned} p(t) &= \sigma\left(\gamma\left(\bar{y}(t) - 0.5\right)\right) \\ a(t) &\sim \text{Bern}(p(t)) \end{aligned} \tag{12}$$

Where $\bar{y}(t)$ is a rolling window average of the previous outcomes, with a window size of $\omega = 15$.

## G  BENCHMARK METHOD IMPLEMENTATION DETAILS

We seek to compare against the existing state-of-the-art (SOTA) methods from (1) longitudinal treatment effect models and (2) ODE discovery. First, longitudinal treatment effect models often consist of black-box neural network-based approaches, i.e., not closed-form or easily interpretable. A significant theme in these works is the development of methods to mitigate time-dependent confounding. Many mitigation methods have been proposed that we use as benchmarks, such as propensity networks in Marginal Structural Models (**MSM**) (Robins et al., 2000) and Recurrent Marginal Structural Networks (**RMSN**) (Lim, 2018), Gradient Reversal in Counterfactual Recurrent Networks (**CRN**) (Bica et al., 2020b), Confusion in Causal Transformers (**CT**) (Melnychuk et al., 2022) and g-computation in G-Net (**G-Net**) (Li et al., 2021). Second, ODE discovery methods aim to discover an underlying closed-form mathematical equation $\boldsymbol{F}$ that best fits the underlying controlled ODE of the observed state and action trajectories dataset, $\mathcal{D}$. In these methods, the state derivative is unobserved; therefore, they have to approximate the derivative using finite differences, as in Sparse Identification of Nonlinear Dynamics (Brunton et al., 2016), or employ a variational loss as the objective, as in Weak Sparse Identification of Nonlinear Dynamics (Messenger & Bortz, 2021). [6] To

---

[6]We only include results for WSINDy when it is possible to use it, as it does not support sparse short trajectories.

Table 8: **Parameter values for the different layers of between-subject variability (A-D) for the two PKPD models.**

| Dataset | Outcome ($y$) | Covariates | Parameters | Treatment |
|---|---|---|---|---|
| eq. (5).A | $y(t) = x(t)$ | $c_0, c_1$ | $C_0 = c_0$ 
 $C_1 = c_1$ | $a(t) = a(0) \in \{0, 1\}$ |
| eq. (5).B | $y(t) = x(t) + \epsilon$ | $c_0, c_1$ | $C_0 = c_0$ 
 $C_1 = c_1$ | $a(t) = a(0) \in \{0, 1\}$ |
| eq. (5).C | $y(t) = x(t) + \epsilon$ | $c_0, c_1$ | $C_0 = c_0 w_0 + w_1$ 
 $C_1 = c_1 w_2 + w_3$ | $a(t) = a(0) \in \{0, 1\}$ |
| eq. (5).D | $y(t) = x(t) + \epsilon$ | $c_0, c_1$ | $C_0 \sim \mathcal{N}(c_0 w_0 + w_1, 0.25)$ 
 $C_1 \sim \mathcal{N}(c_1 w_2 + w_3, 0.25)$ | $a(t) = a(0) \in \{0, 1\}$ |
| eq. (6).A | $y(t) = x(t)$ | $v_{\text{Patient Type}} = 1$ | $\mu(\alpha_r) = 1.1 \times \mu(\alpha_r)$ 
 $\beta_c = \mu(\beta_c)$ | $\boldsymbol{a} = (c(t), d(t))$ |
| eq. (6).B | $y(t) = x(t) + \epsilon$ | $v_{\text{Patient Type}} = 1$ | $\mu(\alpha_r) = 1.1 \times \mu(\alpha_r)$ 
 $\beta_c = \mu(\beta_c)$ | $\boldsymbol{a} = (c(t), d(t))$ |
| eq. (6).C | $y(t) = x(t) + \epsilon$ | $v_{\text{Patient Type}} \in \{1, 2, 3\}$ | $v_{\text{Patient Type}} = \begin{cases} 1: & \mu(\alpha_r) = 1.1 \times \mu(\alpha_r) \\ 3: & \mu(\alpha_c) = 1.1 \times \mu(\alpha_c) \\ 2: & \text{No modification} \end{cases}$ 
 $\beta_c = \mu(\beta_c)$ | $\boldsymbol{a} = (c(t), d(t))$ |
| eq. (6).D | $y(t) = x(t) + \epsilon$ | $v_{\text{Patient Type}} \in \{1, 2, 3\}$ | $v_{\text{Patient Type}} = \begin{cases} 1: & \mu(\alpha_r) = 1.1 \times \mu(\alpha_r) \\ 3: & \mu(\alpha_c) = 1.1 \times \mu(\alpha_c) \\ 2: & \text{No modification} \end{cases}$ 
 $\beta_c \sim \mathcal{N}(\mu(\beta_c), \sigma(\beta_c))$ | $\boldsymbol{a} = (c(t), d(t))$ |

make these two ODE discovery methods more competitive, we adapt them to model individual ODEs per categorical treatment (section 4.2), (termed **A-SINDy**, **A-WSINDy** respectively). We further discuss why we chose these state-of-the-art methods and their associated implementation details in the following.

To make the LTE methods competitive, the hyperparameters are tuned following Melnychuk et al. (2022), to the Cancer PKPD dataset—and then kept constant for all experiments. Specifically, we use Melnychuk et al. (2022)'s tuned hyperparameters, which are detailed below for each LTE method. We also detail hyperparameter tuning in appendix I.

Longitudinal Treatment Effect Models

**Marginal Structural Models (MSMs)** Robins et al. (2000); Hernán et al. (2001) are popular methods in epidemiology designed for the estimation of counterfactual outcomes with inverse probability of treatment weights (IPTW) through linear modelling. This approach utilizes stabilized weights to eliminate time-varying confounding bias.

Upon estimation of the stabilized weights, they are normalized and truncated at their 1st and 99th percentiles to align with Lim (2018). Outcome regressions are applied separately for each prediction horizon. For any given horizon $\tau$, the dataset is split into smaller segments using a rolling origin, and stabilized weights are computed for each segment.

No hyperparameters are required for MSMs; hence, in all experiments, training and validation subsets are combined.

**Recurrent Marginal Structural Networks (RMSNs)** Lim (2018) use a sequence-to-sequence architecture composed of four LSTM subnetworks. It manages multiple binary treatments by re-weighting the objective with the IPTW during training, creating a pseudo-population that emulates a randomized controlled trial.

The propensity networks are initially trained to estimate the stabilized weights. Subsequently, the encoder is trained using a mean squared error weighted with the stabilized weights for one-step-ahead predictions. Lastly, the decoder is trained by minimizing the loss using the fully stabilized weights. The dataset is processed into smaller chunks using rolling origins for this stage of training.

Here the hyperparameters are: the propensity treatment model has 8 sequential hidden units, a dropout rate of 0.1, one layer, uses a batch size of 64, with a max grad norm of 2.0, and is optimized with the Adam optimizer (Kingma & Ba, 2014) with a learning rate of 0.001. The propensity history model has 16 sequential units, with a dropout rate of 0.3, one layer, uses a batch size of 256, with a max grad norm of 1.0, and an Adam optimizer with a learning rate of 0.01. The encoder model has 12 sequential hidden units, a dropout rate of 0.1, one layer, a batch size of 64, a max grad norm of 2.0, and an Adam optimizer with a learning rate of 0.001. Moreover, the decoder model uses 64 sequential hidden units, with a dropout rate of 0.2, one layer, a batch size of 256, a max grad norm of 1.0, and an Adam optimizer with a learning rate of 0.001.

**Counterfactual Recurrent Network (CRN)** Bica et al. (2020b) utilizes an encoder-decoder architecture, applying an adversarial learning technique, gradient reversal Ganin & Lempitsky (2015), to create balanced representations non-predictive of the treatment assignment.

CRN's encoder and decoder are composed of a single LSTM-layer each. Training involves minimizing a loss function that applies gradient reversal to minimize cross-entropy between the predicted and current treatment, while simultaneously maximizing the entropy of the built representations.

Here the hyperparameters are, the encoder model has a balancing representation size of 6, 18 fully connected hidden units, with a dropout rate of 0.2, 24 sequential hidden units, uses a batch size of 64 and is optimized with an Adam optimizer with a learning rate of 0.01. Moreover, the decoder model has a balancing representation size of 3, 9 fully connected hidden units, with a dropout rate of 0.2, 24 sequential hidden units, uses a batch size of 512 and is optimized with an Adam optimizer with a learning rate of 0.001. To make this baseline as competitive as possible, we use the domain confusion balancing loss from Melnychuk et al. (2022).

**G-Net** Li et al. (2021) is a recent method for estimating time-varying treatment effects, based on the G-computation formulation. It leverages a recurrent architecture that directly models the evolution of treatments and outcomes over time.

The network is designed to simultaneously model the auto-regressive dynamics of outcomes and the time-varying effects of treatments. For each time step, the model uses LSTM cells to capture the auto-regressive outcome dynamics and a separate LSTM layer to model the time-varying treatment effects.

The model is trained using a mean squared error loss for outcome prediction and a binary cross-entropy loss for treatment assignment prediction.

Here the hyperparameters are, 24 sequential hidden units, 48 fully connected hidden units, one layer, r size of 3, with a dropout rate of 0.1, uses a batch size of 128, 25 Monte Carlo samples and is optimized with an Adam optimizer with a learning rate of 0.01.

**Causal Transformer (CT)** Melnychuk et al. (2022) is a recent state-of-the-art method for treatment effects over time. It is based on the Transformer architecture Vaswani et al. (2017), which is effective in modelling long-range dependencies in sequential data. CT uses a causal attention mechanism to model the temporal dynamics of the treatment effects. It is composed of three transformer subnetworks with separate inputs for time-varying covariates, previous treatments, and previous outcomes into a joint network that has in-between cross-attentions.

To encourage balanced representations that are predictive of the next outcome but non-predictive of the current treatment assignment, CT optimizes a loss that is composed of two terms. The first term is a supervised loss, such as Mean Squared Error (MSE), which measures the difference between the predicted and actual outcomes. The second term is a domain confusion loss, which encourages the model to minimize the difference between the distributions of representations conditioned on different treatments.

Here the hyperparameters are 16 sequential hidden units, a balancing representation size of 16, 32 fully connected units, a dropout rate of 0.1, a batch size of 256, and is optimized with an Adam optimizer with a learning rate of 0.01.

ODE DISCOVERY METHODS

**Sparse Identification of Nonlinear Dynamics (SINDy)** (Brunton et al., 2016), a data-driven framework that aims to discover the governing dynamical system equations directly from time-series data. The algorithm works by iteratively performing sparse regression on a library of candidate functions to identify the sparsest yet most accurate representation of the dynamical system.

In our implementation, we use a polynomial library of order two, which is a feature library of $\mathcal{L} = \{1, x_0, x_1, x_0 x_1\}$. Finite difference approximations are used to compute time derivatives from the input time-series data, of order one. Here the alpha parameter is kept constant at 0.5 across all experiments, and the sparsity threshold is set to 0.001 for all experiments, apart from the eq. (5) datasets where it is set to 0.1.

**Weak Sparse Identification of Nonlinear Dynamics (WSINDy)** (Messenger & Bortz, 2021) is a variant of SINDy designed to handle noisy sampled time-series data. Instead of directly differentiating the time-series data to obtain derivatives, WSINDy formulates a variational optimization problem that simultaneously identifies the governing equations and estimates the derivatives.

The implementation details are largely similar to those for SINDy, with the significant difference being in the formulation of the loss function for the optimization problem. Here, we use 100 domain centres and a polynomial library of order two. Specifically, WSINDy requires multiple domain centres in the variational formulation, which precludes the application of WSINDy to sparse and short sequence trajectories—leading to the inclusion of this baseline only where it is possible to apply it (when the minimum trajectories are long enough that it can be applied).

To form the **Adapted SINDy (A-SINDy)** and **Adapted WSINDy (A-WSINDy)**, we modified the SINDy and WSINDy methods to handle categorical treatments by learning individual ODEs for each treatment group. This adaptation makes these models more competitive for the benchmark comparisons in this study.

**Individualized Nonlinear Sparse Identification Treatment Effect (INSITE)**, our proposed method, builds on top of SINDy that discovers a population (global) differential equation for all patients in the training dataset. Therefore it uses the same hyperparameters as defined above for SINDy.

Moreover, INSITE fine-tunes the discovered population ODE to existing observed patient trajectories by minimizing the MSE loss between the predicted and observed outcomes. Here the regularization parameter $\lambda$ is set to $\lambda = 10.0$ across all experiments. It was set following the same hyperparameter tuning procedure on the validation dataset on the Cancer PKPD dataset and then kept constant for all experiments.

## G.1 INSITES INFERENCE PROCEDURE

INSITE principally consists of a training step and an inference step. First, for the training step, it discovers a population (global) differential equation for all patients in the training dataset. The exact form of this global differential equation that is discovered depends on the treatment type, as following the types outlined in appendix E. Commonly, the treatment will be a *categorical treatment*, where the action trajectory is one-dimensional with values in $[1, K]$, therefore the form of the global ODEs that are learned are $K$ separate closed-form systems of ODEs $\boldsymbol{f}_1, \ldots, \boldsymbol{f}_K$.

For the inference step at run-time, a new (unseen) patient is observed with covariates and an initial treatment history following a specified treatment plan. INSITE fine-tunes the active $K$ discovered population ODEs to existing observed patient trajectories by minimizing the MSE loss between the predicted and observed outcomes, for the active $K^{\text{th}}$ separate closed-form ODE as determined by the existing observed treatments. Given this, there can exist a case where a discrete treatment type may not be observed in the existing patient observed treatment history, in that case, the separate ODE that corresponds to that discrete treatment is not fine-tuned, and instead is left as the underlying population (global) differential equation for all patients in the training dataset with that specific discrete treatment.

## H  EVALUATION METRICS

We evaluate against the standard longitudinal treatment effect metrics of test counterfactual $\tau$-step ahead prediction normalized root mean squared error (RMSE), where $\tau = \{1, 2, 3, 4, 5, 6\}$, for a sliding treatment plan (Bica et al., 2020b; Melnychuk et al., 2022). Unless otherwise specified, each result is run for five random seed runs with time-dependent confounding of $\gamma = 2$.

To evaluate each baseline, we use the same setup as Melnychuk et al. (2022). That is for each dataset, for each patient in the test set and for each time step, we simulate multiple counterfactual trajectories by setting the treatment to counterfactual possible treatment values, depending on $\tau$. First, for one-step ahead prediction $\tau = 1$, we simulate all the possible counterfactual treatment values (e.g., in eq. (5) we simulate both $a = 0$ and $a = 1$, and eq. (6) & Cancer PKPD we simulate all four combinations of $\{(a_t^r = 0, a_t^c = 0), (a_t^r = 1, a_t^c = 0), (a_t^r = 0, a_t^c = 1), (a_t^r = 1, a_t^c = 1)\}$) for the next counterfactual outcome $y(t + 1)$. This corresponds to the PKPD model under all the feasible treatment assignments. Second, for multi-step ahead prediction, the number of possible potential outcomes $y(t + 2, \ldots, t + \tau_{\max})$ grows exponentially with the projection horizon $\tau_{\max}$. Therefore we use the sliding window treatment assignment (Bica et al., 2020b; Melnychuk et al., 2022). This sliding window treatment assignment can help test that the correct timing of treatment is chosen. It is implemented by simulating trajectories with a single treatment, however, the treatments are iteratively moved over a window ranging from $t$ to $t + \tau_{\max} - 1$, this essentially results in $2(\tau_{\max} - 1)$ trajectories.

For each random seed run, we generate a new train, validation and test dataset independently. With 1000 training trajectories, 100 validation trajectories and 100 test trajectories, unless otherwise noted. We then train each baseline on the training dataset and tune the hyperparameters on the validation dataset. We then evaluate the performance of each baseline on the test dataset. We repeat this process for each random seed run, for a total of five random seed runs, unless otherwise noted. We then report the mean and 95% confidence interval of the normalized RMSE between the predicted and observed outcomes for each of these counterfactual trajectories. We report the normalized RMSE, as is standard (Bica et al., 2020b; Melnychuk et al., 2022), where we normalize by the maximum outcome value for that dataset, where for eq. (5) $y_{\max} = 50.0$ and for eq. (6) and Cancer PKPD $y_{\max} = 1150 \text{cm}^3$.

## I  DATASET GENERATION AND MODEL TRAINING

**Dataset generation $\mathcal{D}$.** We generate a dataset for each underlying pharmacological model $\boldsymbol{F}$ and a given action policy. This is achieved by sampling a set of $N$ initial conditions $x_0 \in \mathcal{X}, v \in \mathcal{V}$ from the models specified domains $\mathcal{X}, \mathcal{V}$. These initial values and the action policy simulate the covariate trajectory up to a defined end time $T$, using a numerical ODE solver. This forms a dataset as described in Section 3. We follow this process to independently sample a train, validation, and test dataset $\mathcal{D}$.

**Model training.** We train each baseline on the training dataset $\mathcal{D}_{\text{train}}$. We follow the same training setup as Melnychuk et al. (2022), where all the baselines are implemented in PyTorch lightning (Falcon, 2019) and trained with the Adam optimizer (Kingma & Ba, 2014). Following Melnychuk et al. (2022), we train all LTE baselines using the teacher forcing technique (Williams & Zipser, 1989) when training the models for multi-step ahead prediction. During the evaluation of multi-step ahead prediction, we switch off teacher forcing and autoregressively feed model predictions. Furthermore, we train each baseline for 100 epochs, for the experimental evaluation setup, see appendix H. We perform all experiments and training using a single Intel Core i9-12900K CPU @ 3.20GHz, 64GB RAM with an Nvidia RTX3090 GPU 24GB.

**Hyper parameter tuning.** We followed the hyperparameter tuning setup of Melnychuk et al. (2022) and used their tuned hyperparameters for all the baselines to the Cancer PKPD dataset across all datasets and all experiments. Therefore, we only tuned the hyperparameters of the ODE discovery methods to the validation dataset of the Cancer PKPD dataset and fixed them throughout all experiments on all other datasets.

## J   INTERPRETABLE EQUATIONS

Unique to the proposed framework and INSITE is that the discovered differential equation is fully interpretable. Some of these discovered equations are shown in table 9. It is clear that even with binary actions, INSITE is able to discover a useful equation that performs well and is similar in form to the true underlying equation, even discovering it exactly in some simple settings of eq. (5) (e.g., if we discretize the numeric constants, rounding to one decimal place for eq. (5).A and eq. (5).B we recover the true underlying ODE).

Table 9: **Interpretable discovered ODEs.** We show the discovered equations alongside the true underlying data generating ODE for the one-compartment PKPD model in eq. (5), with increasing levels of BSV complexity.

| Dataset | True ODE (Data generating process) | Discovered Population ODE ($\bar{F}$) |
|---|---|---|
| eq. (5).A | $\frac{dx(t)}{dt} = \begin{cases} -C_0 x(t), & \text{if } a = 0 \\ -C_1 x(t), & \text{if } a = 1 \end{cases}$ | $\frac{dx(t)}{dt} = \begin{cases} -1.008546 C_0 x(t), & \text{if } a = 0 \\ -1.008550 C_1 x(t), & \text{if } a = 1 \end{cases}$ |
| eq. (5).B | $\frac{dx(t)}{dt} = \begin{cases} -C_0 x(t), & \text{if } a = 0 \\ -C_1 x(t), & \text{if } a = 1 \end{cases}$ | $\frac{dx(t)}{dt} = \begin{cases} -1.008559 C_0 x(t), & \text{if } a = 0 \\ -1.008545 C_1 x(t), & \text{if } a = 1 \end{cases}$ |
| eq. (5).C | $\frac{dx(t)}{dt} = \begin{cases} -C_0 x(t) - 0.05 x(t), & \text{if } a = 0 \\ -C_1 x(t) - 0.15 x(t), & \text{if } a = 1 \end{cases}$ | $\frac{dx(t)}{dt} = \begin{cases} -1.019147 C_0 x(t) - 0.045576 x(t), & \text{if } a = 0 \\ -1.021360 C_1 x(t) - 0.146457 x(t), & \text{if } a = 1 \end{cases}$ |
| eq. (5).D | $\frac{dx(t)}{dt} = \begin{cases} -C_0 x(t) - 0.05 x(t), & \text{if } a = 0 \\ -C_1 x(t) - 0.15 x(t), & \text{if } a = 1 \end{cases}$ | $\frac{dx(t)}{dt} = \begin{cases} -1.020334 C_0 x(t) - 0.107606 x(t), & \text{if } a = 0 \\ -1.018707 C_1 x(t) - 0.047018 x(t), & \text{if } a = 1 \end{cases}$ |

## K   ADDITIONAL EXPERIMENTS

### K.1   SENSITIVITY TO $\lambda$

In the following, we explore the sensitivity of INSITE to the regularization hyperparameter constant $\lambda$. As tabulated in table 10, we see that INISTE does indeed depend on the correct choice of $\lambda$. We follow the hyperparameter tuning setup of tuning $\lambda$ on the validation test dataset, as is standard for other LTE methods, such as Causal Transformer (CT) (Melnychuk et al., 2022). We further detail this tuning strategy in appendix I; specifically, we chose $\lambda = 10$, tuned on the validation set for dataset Eq. 5.D, and kept the same value across all datasets, and all runs (unless explicitly, as noted in this sensitivity analysis).

Table 10: **Counterfactual normalised RMSE,** for 6-step ahead prediction of the benchmarks on each synthetic dataset detailed in appendix F, with INSITE varying its regularization hyperparameter $\lambda$. We quote 95% confidence intervals with each value, and all metrics are averaged over ten random seed runs. Each PKPD underlying pharmacological model is simulated with different layers of BSV (table 2) for A-D. INSITE is sensitive to its regularization hyperparameter $\lambda$.

| | Method | eq. (5).A | eq. (5).B | eq. (5).C | eq. (5).D | eq. (6).A | eq. (6).B | eq. (6).C | eq. (6).D | Cancer PKPD |
|---|---|---|---|---|---|---|---|---|---|---|
| **LTE** | MSM | 0.99±8.37e-17 | 0.99±0.00 | 0.97±8.37e-17 | 2.09±0.00 | 2.55±0.13 | 2.55±0.13 | 2.06±0.16 | 2.11±0.04 | 2.30±0.12 |
| | RMSN | 1.92±0.24 | 1.94±0.23 | 1.68±0.19 | 1.91±0.18 | 1.23±0.15 | 1.25±0.15 | 1.10±0.18 | 1.10±0.11 | 1.04±0.17 |
| | CRN | 1.05±0.10 | 1.05±0.10 | 0.82±0.09 | 1.98±0.14 | 1.05±0.03 | 1.05±0.03 | 1.03±0.08 | 1.03±0.10 | 0.92±0.08 |
| | G-Net | 0.91±0.20 | 0.91±0.20 | 0.72±0.14 | 0.97±0.15 | 1.33±0.27 | 1.34±0.27 | 1.02±0.11 | 1.25±0.15 | 1.22±0.14 |
| | CT | 0.90±0.18 | 0.90±0.18 | 0.75±0.14 | 1.00±0.14 | 1.29±0.07 | 1.29±0.10 | 1.03±0.11 | 1.14±0.10 | 1.07±0.07 |
| **ODE-D** | A-SINDy | 0.11±0.00 | 0.11±0.00 | 0.13±2.09e-17 | 0.15±0.00 | 1.45±0.03 | 1.45±0.03 | 1.40±0.01 | 1.51±0.09 | 1.23±0.13 |
| | A-WSINDy | 0.11±7.24e-05 | 0.11±2.49e-04 | 0.12±1.47e-03 | 0.10±7.61e-04 | NA | NA | NA | NA | NA |
| | INSITE $\lambda = 0.0$ | **0.02** ±0.00 | **0.03** ±0.00 | **0.04** ±0.00 | **0.05** ±0.00 | 2.97 ±0.00 | 1.17 ±1.67e-16 | 2.98 ±0.00 | 0.99 ±0.00 | 1.22 ±0.00 |
| | INSITE $\lambda = 10.0$ | **0.02** ±2.62e-18 | **0.03** ±0.00 | **0.04** ±0.00 | **0.05** ±5.23e-18 | **0.94** ±0.05 | **0.94** ±0.05 | **0.84** ±0.04 | **0.87** ±0.08 | 0.79 ±8.37e-17 |
| | INSITE $\lambda = 500.0$ | 0.03 ±0.00 | **0.03** ±0.00 | **0.04** ±0.00 | **0.05** ±0.00 | 1.32 ±0.00 | 1.32 ±0.00 | 1.20 ±0.00 | 1.11 ±0.00 | **0.67** ±0.00 |

### K.2   PARAMETRIC DISTRIBUTION OF NUMERIC CONSTANTS

Unique to our method, INSITE is that after obtaining the patient-specific differential equations, we can derive a population differential equation where each numeric constant is represented by a distribution, such as a normal distribution or a mixture of distributions—recovering a probabilistic interpretation of the underlying data generating process.

We explore a further insight experiment, where we discover the parametric distribution of the numeric constants of a further synthetic equation. We extended our existing eq. (5), to have an offset term $\beta_0$ that is sampled from a bimodal Gaussian mixture distribution. We follow our same experimental

setup and discover individualized differential equations for each patient. To explore if we can recover the underlying bimodal distribution, we plot the distribution of the numeric constants $\beta_0$ for each patient. We observe that the distribution of the numeric constants for each patient is bimodal and that the two modes are the two modes of the underlying bimodal distribution. This is shown in fig. 4, with a kernel distribution estimation plot.

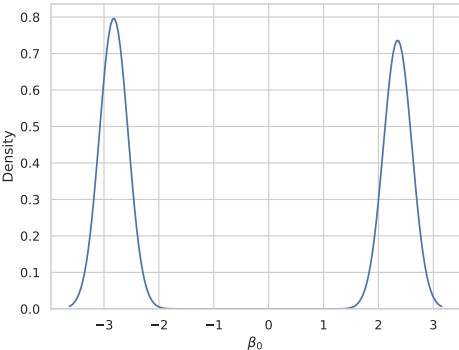

Figure 4: **Kernel distribution estimation plot** of the individualized numeric constants $\beta_0$ for each patient, for a further synthetic equation that extends eq. (5), to have an offset term $\beta_0$ that is sampled from a bimodal Gaussian mixture distribution. We observe that the distribution of the numeric constants $\beta_0$ for each patient is bimodal and that the two modes are the two modes of the underlying bimodal distribution. This verifies that INSITE can recover a probabilistic interpretation of the underlying data-generating process, with a population differential equation where each numeric constants can be represented by a distribution.

## K.3 ADDITIONAL MAIN RESULTS

In the following, we provide full additional results for all counterfactual $\tau$-step ahead prediction errors of the benchmarks on each synthetic dataset. Where we tabulate: 1-step in table 11, 2-step in table 12, 3-step in table 13, 4-step in table 14, 5-step in table 15 and 6-step in table 16.

Table 11: **Counterfactual normalized RMSE,** for 1-step ahead prediction of the benchmarks on each synthetic dataset detailed in appendix F. We quote 95% confidence intervals with each value, and all metrics are averaged over ten random seed runs. Each PKPD underlying pharmacological model is simulated with different layers of BSV (table 2) for A-D. Our contribution is  shaded  below.

| | Method | eq. (5).A | eq. (5).B | eq. (5).C | eq. (5).D | eq. (6).A | eq. (6).B | eq. (6).C | eq. (6).D | Cancer PKPD |
|---|---|---|---|---|---|---|---|---|---|---|
| **LTE** | MSM | 0.56±0.00 | 0.56±8.37e-17 | 0.67±0.00 | 0.52±8.37e-17 | 1.36±0.14 | 1.36±0.14 | 1.32±0.02 | 1.38±0.06 | 1.21±0.06 |
| | RMSN | 2.69±0.17 | 2.70±0.17 | 2.58±0.13 | 2.53±0.14 | 0.97±0.11 | 0.97±0.11 | 0.85±0.03 | 0.90±0.06 | 0.74±0.10 |
| | CRN | 1.08±0.17 | 1.10±0.18 | 1.04±0.26 | 1.19±0.16 | 0.65±0.04 | 0.65±0.04 | **0.61** ±0.01 | 0.60±0.01 | **0.60** ±0.03 |
| | G-Net | 0.46±0.13 | 0.46±0.13 | 0.39±0.10 | 0.55±0.10 | **0.61** ±0.05 | **0.61** ±0.05 | **0.61** ±0.02 | **0.57** ±0.03 | **0.59** ±0.04 |
| | CT | 0.33±0.03 | 0.33±0.03 | 0.32±0.04 | 0.39±0.03 | 0.82±0.08 | 0.82±0.08 | 0.83±0.06 | 0.81±0.07 | 0.68±0.05 |
| **ODE-D** | A-SINDy | 0.11±1.05e-17 | 0.11±2.09e-17 | 0.14±0.00 | 0.16±0.00 | 1.61±0.13 | 1.61±0.13 | 1.31±0.09 | 1.65±0.13 | 1.70±0.07 |
| | A-WSINDy | 0.11±7.92e-05 | 0.11±2.46e-04 | 0.12±1.46e-03 | 0.10±7.62e-04 | NA | NA | NA | NA | NA |
| | INSITE | **1.35e-03** ±1.03e-19 | **0.02** ±8.27e-19 | **0.02** ±2.62e-18 | **0.02** ±8.27e-19 | 0.94±0.07 | 0.94±0.07 | 0.89±0.00 | 0.80±0.01 | 0.83±7.94e-17 |

Table 12: **Counterfactual normalized RMSE,** for 2-step ahead prediction of the benchmarks on each synthetic dataset detailed in appendix F. We quote 95% confidence intervals with each value, and all metrics are averaged over ten random seed runs. Each PKPD underlying pharmacological model is simulated with different layers of BSV (table 2) for A-D. Our contribution is  shaded  below.

| | Method | eq. (5).A | eq. (5).B | eq. (5).C | eq. (5).D | eq. (6).A | eq. (6).B | eq. (6).C | eq. (6).D | Cancer PKPD |
|---|---|---|---|---|---|---|---|---|---|---|
| **LTE** | MSM | 1.31±1.67e-16 | 1.31±1.67e-16 | 1.41±1.67e-16 | 1.34±1.67e-16 | 2.00±0.15 | 2.00±0.15 | 1.62±0.04 | 1.74±0.10 | 1.78±0.11 |
| | RMSN | 2.38±0.15 | 2.39±0.14 | 2.26±0.10 | 2.28±0.13 | 1.04±0.09 | 1.04±0.09 | 1.02±0.11 | 1.14±0.16 | 0.81±0.06 |
| | CRN | 0.90±0.10 | 0.91±0.10 | 0.60±0.20 | 2.04±0.16 | **0.77** ±0.04 | **0.77** ±0.04 | **0.77** ±0.05 | **0.87** ±0.10 | **0.66** ±0.05 |
| | G-Net | 0.63±0.15 | 0.63±0.14 | 0.54±0.11 | 0.67±0.10 | 0.88±0.08 | 0.87±0.06 | 0.87±0.06 | 0.99±0.09 | 0.84±0.05 |
| | CT | 0.54±0.09 | 0.54±0.09 | 0.45±0.09 | 0.67±0.07 | 0.99±0.10 | 0.99±0.10 | 1.00±0.10 | 1.09±0.17 | 0.82±0.07 |
| **ODE-D** | A-SINDy | 0.11±1.05e-17 | 0.11±0.00 | 0.14±0.00 | 0.16±0.00 | 1.47±0.05 | 1.47±0.05 | 1.51±0.06 | 1.63±0.13 | 1.27±0.14 |
| | A-WSINDy | 0.11±7.81e-05 | 0.11±2.49e-04 | 0.12±1.49e-03 | 0.10±7.64e-04 | NA | NA | NA | NA | NA |
| | INSITE | **8.44e-03** ±1.31e-18 | **0.02** ±0.00 | **0.03** ±0.00 | **0.03** ±0.00 | **0.82** ±0.02 | **0.82** ±0.02 | 0.91±0.02 | **0.92** ±0.12 | 0.80±0.00 |

## K.4 ADDITIONAL RESULTS WITH VARYING SAMPLE SIZES

In the following, we provide additional results where we vary the number of training sample sizes, in the range of $n = \{10, 100, 1,000, 10,000\}$. For each sample size, we provide the counterfactual

Table 13: **Counterfactual normalized RMSE,** for 3-step ahead prediction of the benchmarks on each synthetic dataset detailed in appendix F. We quote 95% confidence intervals with each value, and all metrics are averaged over ten random seed runs. Each PKPD underlying pharmacological model is simulated with different layers of BSV (table 2) for A-D. Our contribution is shaded below.

| | Method | eq. (5).A | eq. (5).B | eq. (5).C | eq. (5).D | eq. (6).A | eq. (6).B | eq. (6).C | eq. (6).D | Cancer PKPD |
|---|---|---|---|---|---|---|---|---|---|---|
| LTE | MSM | 1.21±1.67e-16 | 1.21±1.67e-16 | 1.27±1.67e-16 | 1.48±0.00 | 2.36±0.17 | 2.36±0.17 | 1.91±0.08 | 2.01±0.09 | 2.09±0.12 |
| | RMSN | 2.21±0.14 | 2.22±0.13 | 2.05±0.09 | 2.12±0.14 | 1.06±0.08 | 1.06±0.08 | 1.04±0.11 | 1.14±0.15 | 0.86±0.07 |
| | CRN | 0.88±0.08 | 0.90±0.07 | 0.58±0.14 | 1.94±0.14 | **0.86**±0.03 | **0.86**±0.03 | **0.82**±0.04 | **0.96**±0.11 | **0.73**±0.05 |
| | G-Net | 0.73±0.16 | 0.73±0.16 | 0.61±0.12 | 0.77±0.11 | 1.03±0.11 | 1.02±0.11 | 0.95±0.05 | 1.16±0.10 | 1.02±0.07 |
| | CT | 0.69±0.11 | 0.70±0.12 | 0.58±0.10 | 0.80±0.09 | 1.12±0.09 | 1.13±0.09 | 1.04±0.07 | 1.13±0.15 | 0.88±0.05 |
| ODE-D | A-SINDy | 0.11±1.05e-17 | 0.11±1.05e-17 | 0.13±0.00 | 0.16±2.09e-17 | 1.45±0.05 | 1.46±0.05 | 1.48±0.04 | 1.61±0.13 | 1.25±0.13 |
| | A-WSINDy | 0.11±7.68e-05 | 0.11±2.50e-04 | 0.12±1.50e-03 | 0.10±7.66e-04 | NA | NA | NA | NA | NA |
| | INSITE | **0.01**±0.00 | **0.03**±0.00 | **0.03**±0.00 | **0.03**±0.00 | **0.83**±0.03 | **0.83**±0.03 | 0.89±8.00e-03 | **0.90**±0.11 | **0.78**±0.00 |

Table 14: **Counterfactual normalized RMSE,** for 4-step ahead prediction of the benchmarks on each synthetic dataset detailed in appendix F. We quote 95% confidence intervals with each value, and all metrics are averaged over ten random seed runs. Each PKPD underlying pharmacological model is simulated with different layers of BSV (table 2) for A-D. Our contribution is shaded below.

| | Method | eq. (5).A | eq. (5).B | eq. (5).C | eq. (5).D | eq. (6).A | eq. (6).B | eq. (6).C | eq. (6).D | Cancer PKPD |
|---|---|---|---|---|---|---|---|---|---|---|
| LTE | MSM | 1.12±0.00 | 1.12±0.00 | 1.12±1.67e-16 | 1.68±1.67e-16 | 2.53±0.16 | 2.53±0.16 | 2.06±0.10 | 2.13±0.08 | 2.24±0.12 |
| | RMSN | 2.09±0.16 | 2.10±0.16 | 1.90±0.12 | 2.03±0.16 | 1.12±0.09 | 1.12±0.09 | 1.06±0.12 | 1.12±0.13 | 0.92±0.10 |
| | CRN | 0.91±0.07 | 0.92±0.08 | 0.65±0.11 | 1.93±0.13 | 0.95±0.03 | 0.95±0.03 | **0.89**±0.05 | 1.00±0.11 | **0.80**±0.06 |
| | G-Net | 0.80±0.18 | 0.80±0.18 | 0.66±0.13 | 0.84±0.12 | 1.14±0.16 | 1.14±0.16 | 0.99±0.06 | 1.21±0.11 | 1.10±0.09 |
| | CT | 0.78±0.14 | 0.78±0.14 | 0.66±0.12 | 0.88±0.11 | 1.22±0.08 | 1.22±0.09 | 1.04±0.07 | 1.14±0.13 | 0.96±0.05 |
| ODE-D | A-SINDy | 0.11±0.00 | 0.11±1.05e-17 | 0.13±0.00 | 0.15±0.00 | 1.45±0.04 | 1.45±0.04 | 1.45±0.03 | 1.58±0.11 | 1.24±0.13 |
| | A-WSINDy | 0.11±7.55e-05 | 0.11±2.51e-04 | 0.12±1.50e-03 | 0.10±7.67e-04 | NA | NA | NA | NA | NA |
| | INSITE | **0.02**±2.62e-18 | **0.03**±5.23e-18 | **0.04**±5.23e-18 | **0.04**±0.00 | **0.86**±0.03 | **0.86**±0.03 | **0.87**±9.52e-03 | **0.88**±0.10 | **0.78**±0.00 |

Table 15: **Counterfactual normalized RMSE,** for 5-step ahead prediction of the benchmarks on each synthetic dataset detailed in appendix F. We quote 95% confidence intervals with each value, and all metrics are averaged over ten random seed runs. Each PKPD underlying pharmacological model is simulated with different layers of BSV (table 2) for A-D. Our contribution is shaded below.

| | Method | eq. (5).A | eq. (5).B | eq. (5).C | eq. (5).D | eq. (6).A | eq. (6).B | eq. (6).C | eq. (6).D | Cancer PKPD |
|---|---|---|---|---|---|---|---|---|---|---|
| LTE | MSM | 1.05±0.00 | 1.05±0.00 | 1.01±1.67e-16 | 1.89±1.67e-16 | 2.60±0.15 | 2.60±0.15 | 2.09±0.13 | 2.16±0.06 | 2.30±0.12 |
| | RMSN | 1.99±0.20 | 2.01±0.19 | 1.77±0.15 | 1.96±0.17 | 1.18±0.12 | 1.19±0.12 | 1.08±0.15 | 1.11±0.12 | 0.98±0.14 |
| | CRN | 0.98±0.08 | 0.98±0.08 | 0.73±0.09 | 1.95±0.13 | 1.02±0.03 | 1.02±0.03 | 0.96±0.06 | 1.03±0.11 | 0.87±0.07 |
| | G-Net | 0.86±0.19 | 0.86±0.19 | 0.70±0.14 | 0.91±0.14 | 1.24±0.21 | 1.24±0.21 | 1.01±0.08 | 1.24±0.13 | 1.17±0.11 |
| | CT | 0.85±0.16 | 0.86±0.17 | 0.71±0.13 | 0.95±0.14 | 1.28±0.08 | 1.28±0.09 | 1.04±0.07 | 1.16±0.11 | 1.03±0.06 |
| ODE-D | A-SINDy | 0.11±1.05e-17 | 0.11±0.00 | 0.13±0.00 | 0.15±2.09e-17 | 1.45±0.03 | 1.45±0.04 | 1.43±6.51e-03 | 1.55±0.10 | 1.24±0.13 |
| | A-WSINDy | 0.11±7.40e-05 | 0.11±2.50e-04 | 0.12±1.49e-03 | 0.10±7.65e-04 | NA | NA | NA | NA | NA |
| | INSITE | **0.02**±0.00 | **0.03**±0.00 | **0.04**±5.23e-18 | **0.04**±5.23e-18 | **0.90**±0.04 | **0.90**±0.04 | **0.85**±0.03 | **0.87**±0.09 | **0.78**±0.00 |

Table 16: **Counterfactual normalized RMSE,** for 6-step ahead prediction of the benchmarks on each synthetic dataset detailed in appendix F. We quote 95% confidence intervals with each value, and all metrics are averaged over ten random seed runs. Each PKPD underlying pharmacological model is simulated with different layers of BSV (table 2) for A-D. Our contribution is shaded below.

| | Method | eq. (5).A | eq. (5).B | eq. (5).C | eq. (5).D | eq. (6).A | eq. (6).B | eq. (6).C | eq. (6).D | Cancer PKPD |
|---|---|---|---|---|---|---|---|---|---|---|
| LTE | MSM | 0.99±8.37e-17 | 0.99±0.00 | 0.97±8.37e-17 | 2.09±0.00 | 2.55±0.13 | 2.55±0.13 | 2.06±0.16 | 2.11±0.04 | 2.30±0.12 |
| | RMSN | 1.92±0.24 | 1.94±0.23 | 1.68±0.19 | 1.91±0.18 | 1.23±0.15 | 1.25±0.15 | 1.10±0.18 | 1.10±0.11 | 1.04±0.17 |
| | CRN | 1.05±0.10 | 1.05±0.10 | 0.82±0.09 | 1.98±0.14 | 1.05±0.03 | 1.05±0.03 | 1.03±0.08 | 1.03±0.10 | 0.92±0.08 |
| | G-Net | 0.91±0.20 | 0.91±0.20 | 0.72±0.14 | 0.97±0.15 | 1.33±0.27 | 1.34±0.27 | 1.02±0.11 | 1.25±0.15 | 1.22±0.14 |
| | CT | 0.90±0.18 | 0.90±0.18 | 0.75±0.14 | 1.00±0.14 | 1.29±0.07 | 1.29±0.10 | 1.03±0.11 | 1.14±0.10 | 1.07±0.07 |
| ODE-D | A-SINDy | 0.11±0.00 | 0.11±0.00 | 0.13±2.09e-17 | 0.15±0.00 | 1.45±0.03 | 1.45±0.03 | 1.40±0.01 | 1.51±0.09 | 1.23±0.13 |
| | A-WSINDy | 0.11±7.24e-05 | 0.11±2.49e-04 | 0.12±1.47e-03 | 0.10±7.61e-04 | NA | NA | NA | NA | NA |
| | INSITE | **0.02**±2.62e-18 | **0.03**±0.00 | **0.04**±0.00 | **0.05**±5.23e-18 | **0.94**±0.05 | **0.94**±0.05 | **0.84**±0.04 | **0.87**±0.08 | **0.79**±8.37e-17 |

normalized RMSE, for 6-step ahead prediction of the benchmarks on each synthetic dataset. These are tabulated in: table 17, table 18, table 19 and table 20 respectively.

Table 17: **Counterfactual normalized RMSE,** for 6-step ahead prediction of the benchmarks on each synthetic dataset detailed in appendix F, with each method trained on a training dataset with $n = 10$ samples. We quote 95% confidence intervals with each value, and all metrics are averaged over five random seed runs. Each PKPD underlying pharmacological model is simulated with different layers of BSV (table 2) for A-D. Our contribution is shaded below.

| | Method | eq. (5).A | eq. (5).B | eq. (5).C | eq. (5).D | eq. (6).A | eq. (6).B | eq. (6).C | eq. (6).D | Cancer PKPD |
|---|---|---|---|---|---|---|---|---|---|---|
| LTE | MSM | 0.98±0.00 | 1.78±0.00 | 1.23±0.00 | 2.02±0.00 | 2.83±0.00 | 2.83±0.00 | 1.93±0.00 | 2.23±0.00 | 3.24±0.00 |
| | RMSN | 6.80±1.13 | 6.83±1.07 | 5.83±1.03 | 6.32±1.38 | 2.91±0.16 | 2.91±0.16 | 2.19±0.03 | 2.30±0.18 | 3.37±0.12 |
| | CRN | 2.36±0.46 | 2.36±0.46 | 1.96±0.45 | 3.33±0.60 | 2.60±0.03 | 2.60±0.03 | 1.44±0.07 | 2.08±0.19 | 2.99±0.27 |
| | G-Net | 3.47±0.89 | 3.47±0.89 | 3.08±0.82 | 3.53±0.83 | 2.48±0.68 | 2.48±0.68 | **1.50**±0.37 | 2.08±0.63 | **2.91**±1.09 |
| | CT | 2.54±1.44 | 2.52±1.40 | 2.09±1.30 | 2.48±1.03 | 3.01±0.05 | 2.98±0.15 | 2.01±0.37 | 2.34±0.17 | 3.48±0.12 |
| ODE-D | A-SINDy | 0.11±0.00 | 0.10±0.00 | 0.14±0.00 | 0.14±0.00 | 1.23±0.00 | 1.23±0.00 | 1.80±0.00 | 2.49±0.00 | 2.81±0.00 |
| | A-WSINDy | 0.11±1.38e-04 | 0.10±7.14e-03 | 0.12±8.77e-03 | 0.41±5.47e-04 | NA | NA | NA | NA | NA |
| | INSITE | **0.02**±0.00 | **0.03**±0.00 | **0.05**±0.00 | **0.05**±0.00 | **0.92**±0.00 | **0.92**±0.00 | 1.16±0.00 | 1.27±0.00 | 2.18±0.00 |

Table 18: **Counterfactual normalized RMSE,** for 6-step ahead prediction of the benchmarks on each synthetic dataset detailed in appendix F, with each method trained on a training dataset with $n = 100$ samples. We quote 95% confidence intervals with each value, and all metrics are averaged over five random seed runs. Each PKPD underlying pharmacological model is simulated with different layers of BSV (table 2) for A-D. Our contribution is shaded below.

| | Method | eq. (5).A | eq. (5).B | eq. (5).C | eq. (5).D | eq. (6).A | eq. (6).B | eq. (6).C | eq. (6).D | Cancer PKPD |
|---|---|---|---|---|---|---|---|---|---|---|
| LTE | MSM | 1.03±0.00 | 0.90±0.00 | 1.21±0.00 | 1.84±0.00 | 2.80±0.00 | 2.80±0.00 | 2.21±0.00 | 2.23±0.00 | 3.45±0.00 |
| | RMSN | 6.03±1.12 | 5.99±1.05 | 5.15±1.06 | 5.89±0.82 | 2.79±0.35 | 2.78±0.33 | 2.21±0.24 | 2.38±0.10 | 3.64±0.14 |
| | CRN | 1.75±0.32 | 1.75±0.32 | 1.33±0.24 | 2.77±1.29 | 2.00±0.11 | 2.00±0.11 | 1.41±0.11 | 1.41±0.12 | 2.11±0.20 |
| | G-Net | 1.66±0.41 | 1.66±0.41 | 1.33±0.30 | 2.26±0.64 | 1.54±0.19 | 1.54±0.19 | 2.03±0.19 | 2.06±0.77 | 2.30±0.86 |
| | CT | 1.52±0.92 | 1.52±0.92 | 1.28±0.60 | 1.45±0.37 | 2.53±0.31 | 2.55±0.25 | 2.04±0.08 | 2.02±0.20 | 3.20±0.20 |
| ODE-D | A-SINDy | 0.11±0.00 | 0.11±0.00 | 0.13±0.00 | 0.15±0.00 | 1.27±0.00 | 1.27±0.00 | 1.42±0.00 | 1.26±0.00 | 1.42±0.00 |
| | A-WSINDy | 0.11±2.56e-04 | 0.11±7.25e-04 | 0.12±5.21e-04 | 0.10±2.58e-03 | NA | NA | NA | NA | NA |
| | INSITE | **0.02** ±0.00 | **0.03** ±0.00 | **0.04** ±0.00 | **0.05** ±0.00 | **0.96** ±0.00 | **1.13** ±0.00 | **0.87** ±0.00 | **0.99** ±0.00 | **1.19** ±0.00 |

Table 19: **Counterfactual normalized RMSE,** for 6-step ahead prediction of the benchmarks on each synthetic dataset detailed in appendix F, with each method trained on a training dataset with $n = 1,000$ samples. We quote 95% confidence intervals with each value, and all metrics are averaged over five random seed runs. Each PKPD underlying pharmacological model is simulated with different layers of BSV (table 2) for A-D. Our contribution is shaded below.

| | Method | eq. (5).A | eq. (5).B | eq. (5).C | eq. (5).D | eq. (6).A | eq. (6).B | eq. (6).C | eq. (6).D | Cancer PKPD |
|---|---|---|---|---|---|---|---|---|---|---|
| LTE | MSM | 0.99±1.54e-16 | 0.99±0.00 | 0.97±1.54e-16 | 2.09±0.00 | 2.61±0.00 | 2.61±0.00 | 1.99±3.08e-16 | 2.13±0.00 | 2.36±0.00 |
| | RMSN | 1.87±0.33 | 1.95±0.46 | 1.67±0.30 | 1.81±0.23 | 1.30±0.14 | 1.33±0.14 | **1.00** ±0.29 | 1.16±0.08 | 1.07±0.15 |
| | CRN | 1.08±0.13 | 1.09±0.12 | 0.87±0.12 | 1.95±0.32 | 1.07±0.07 | 1.07±0.07 | 1.00±0.17 | 1.06±0.08 | 0.94±0.09 |
| | G-Net | 0.90±0.45 | 0.91±0.45 | 0.72±0.27 | 1.07±0.23 | **1.43** ±0.65 | **1.43** ±0.65 | **0.96** ±0.16 | 1.39±0.26 | 1.34±0.26 |
| | CT | 0.83±0.31 | 0.83±0.32 | 0.74±0.27 | 0.94±0.19 | 1.34±0.15 | 1.36±0.19 | 1.02±0.09 | 1.21±0.08 | 1.10±0.11 |
| ODE-D | A-SINDy | 0.11±0.00 | 0.11±0.00 | 0.13±0.00 | 0.15±0.00 | 1.46±0.00 | 1.47±0.00 | 1.40±0.00 | 1.56±0.00 | 1.29±0.00 |
| | A-WSINDy | 0.11±1.77e-04 | 0.11±4.92e-04 | 0.11±2.36e-03 | 0.10±1.87e-03 | NA | NA | NA | NA | NA |
| | INSITE | **0.02** ±0.00 | **0.03** ±0.00 | **0.04** ±0.00 | **0.05** ±0.00 | **0.96** ±0.00 | **0.96** ±0.00 | **0.82** ±1.54e-16 | **0.90** ±0.00 | **0.79** ±0.00 |

Table 20: **Counterfactual normalized RMSE,** for 6-step ahead prediction of the benchmarks on each synthetic dataset detailed in appendix F, with each method trained on a training dataset with $n = 10,000$ samples. We quote 95% confidence intervals with each value, and all metrics are averaged over five random seed runs. Each PKPD underlying pharmacological model is simulated with different layers of BSV (table 2) for A-D. Our contribution is shaded below.

| | Method | eq. (5).A | eq. (5).B | eq. (5).C | eq. (5).D | eq. (6).A | eq. (6).B | eq. (6).C | eq. (6).D | Cancer PKPD |
|---|---|---|---|---|---|---|---|---|---|---|
| LTE | MSM | 1.01±0.07 | 0.99±0.00 | 1.04±0.00 | 1.61±0.00 | 2.25±0.98 | **2.17** ±1.40 | **1.69** ±0.99 | 1.80±0.86 | 2.47±0.48 |
| | RMSN | **1.01** ±2.48 | **0.70** ±1.20 | **0.41** ±0.67 | 0.54±0.28 | **0.99** ±0.27 | **0.93** ±0.23 | **0.91** ±0.30 | **1.06** ±0.51 | 1.43±0.32 |
| | CRN | **4.83** ±9.98 | **6.20** ±12.80 | **2.31** ±4.02 | **3.58** ±9.39 | **0.93** ±0.21 | **0.91** ±0.30 | **0.71** ±0.07 | **1.56** ±2.48 | **4.22** ±4.79 |
| | G-Net | 0.93±0.53 | 1.03±0.86 | 0.89±0.17 | **0.92** ±1.00 | **1.14** ±0.30 | **1.16** ±0.49 | 0.97±0.25 | 1.09±0.13 | 1.25±0.13 |
| | CT | **0.47** ±0.50 | **0.52** ±0.81 | **0.58** ±0.85 | 0.51±0.22 | **0.95** ±0.17 | **0.88** ±0.21 | **0.74** ±0.11 | 0.83±0.12 | 1.06±0.15 |
| ODE-D | A-SINDy | 0.11±3.30e-03 | 0.11±0.00 | 0.12±0.00 | 0.15±0.00 | 1.39±0.18 | 1.39±0.18 | 1.30±0.31 | 1.43±0.20 | 1.71±0.02 |
| | A-WSINDy | 0.11±4.41e-03 | 0.11±1.27e-04 | 0.12±1.52e-03 | 0.10±2.86e-04 | NA | NA | NA | NA | NA |
| | INSITE | **0.02** ±4.15e-04 | **0.03** ±0.00 | **0.04** ±2.11e-17 | **0.05** ±0.00 | **0.96** ±7.35e-03 | **0.93** ±0.06 | **0.78** ±3.49e-03 | **0.58** ±0.17 | **0.83** ±0.26 |

### K.5 ADDITIONAL RESULTS WITH INCREASING OBSERVATION NOISE

In the following, we provide additional results where we increase the observation noise of the BSV datasets, in the range of $\epsilon = \{0.01, 0.1, 1.0\}$. For each sample size, we provide the counterfactual normalized RMSE, for 6-step ahead prediction of the benchmarks on each synthetic dataset. These are tabulated in: table 21, table 22 and table 23.

We note that for high observational noise settings, although the performance of SINDy degrades, the performance of WSINDy remains competitive. In settings where WSINDy is supported (i.e., trajectory lengths are not short and are not sparse or variable length), INSITE can use WSINDy to discover the population (global) differential equation model instead of SINDy. This arises as INSITE and more broadly the proposed framework can fine-tune any differential equation discovery method that produces a population (global) closed-form differential equation that has numeric constants.

### K.6 MODEL MISSPECIFICATION

INSITE and the other ODE discovery methods (e.g., A-SINDy) are only correctly specified for eq. (5).A-D datasets, as they use the feature library of $\mathcal{L}_{\text{INSITE}} = \{1, x_0, x_1, x_0 x_1\}$. Crucially, the ODE discovery methods are misspecified for eq. (6).A-D, and the Cancer PKPD datasets, as their underlying equation would require the feature library of $\mathcal{L}_{\text{Cancer}} = \{1, x_0, x_1, x_0 x_1, x_0^2 x_1, x_0 x_1^2, \log(x_0), \log(x_1)\}$ to be correctly specified. Although this misspecification persists (in the main experimental table in table 3), as noticeable from the increased order

Table 21: **Counterfactual normalized RMSE,** for 6-step ahead prediction of the benchmarks on each synthetic dataset detailed in appendix F, with BSV observation noise of $\epsilon = 0.01$. We quote 95% confidence intervals with each value, and all metrics are averaged over five random seed runs. Each PKPD underlying pharmacological model is simulated with different layers of BSV (table 2) for A-D. Our contribution is  shaded  below.

| | Method | eq. (5).A | eq. (5).B | eq. (5).C | eq. (5).D | eq. (6).A | eq. (6).B | eq. (6).C | eq. (6).D |
|---|---|---|---|---|---|---|---|---|---|
| LTE | MSM | 0.99±1.54e-16 | 0.99±0.00 | 0.97±1.54e-16 | 2.09±0.00 | 2.61±0.00 | 2.61±0.00 | 1.99±3.08e-16 | 2.13±0.00 |
| | RMSN | 1.87±0.33 | 1.95±0.46 | 1.67±0.30 | 1.81±0.23 | 1.30±0.14 | 1.33±0.14 | **1.00** ±0.29 | 1.16±0.08 |
| | CRN | 1.08±0.13 | 1.09±0.12 | 0.87±0.12 | 1.95±0.32 | 1.07±0.07 | 1.07±0.07 | 1.00±0.17 | 1.06±0.08 |
| | G-Net | 0.90±0.45 | 0.91±0.45 | 0.72±0.27 | 1.07±0.23 | 1.43 ±0.65 | 1.43 ±0.65 | 0.96 ±0.16 | 1.39±0.26 |
| | CT | 0.83±0.31 | 0.83±0.32 | 0.74±0.27 | 0.94±0.19 | 1.34±0.15 | 1.36±0.19 | 1.02±0.09 | 1.21±0.08 |
| ODE-D | A-SINDy | 0.11±0.00 | 0.11±0.00 | 0.13±0.00 | 0.15±0.00 | 1.46±0.00 | 1.47±0.00 | 1.40±0.00 | 1.56±0.00 |
| | A-WSINDy | 0.11±1.77e-04 | 0.11±4.92e-04 | 0.11±2.36e-03 | 0.10±1.87e-03 | NA | NA | NA | NA |
| | INSITE | **0.02** ±0.00 | **0.03** ±0.00 | **0.04** ±0.00 | **0.05** ±0.00 | **0.96** ±0.00 | **0.96** ±0.00 | **0.82** ±1.54e-16 | **0.90** ±0.00 |

Table 22: **Counterfactual normalized RMSE,** for 6-step ahead prediction of the benchmarks on each synthetic dataset detailed in appendix F, with BSV observation noise of $\epsilon = 0.1$. We quote 95% confidence intervals with each value, and all metrics are averaged over five random seed runs. Each PKPD underlying pharmacological model is simulated with different layers of BSV (table 2) for A-D. Our contribution is  shaded  below.

| | Method | eq. (5).A | eq. (5).B | eq. (5).C | eq. (5).D | eq. (6).A | eq. (6).B | eq. (6).C | eq. (6).D |
|---|---|---|---|---|---|---|---|---|---|
| LTE | MSM | 0.99±1.54e-16 | 1.11±0.00 | 1.17±0.00 | 2.51±0.00 | 2.61±0.00 | 2.61±0.00 | 1.99±0.00 | 2.13±0.00 |
| | RMSN | 1.87±0.33 | 1.94±0.61 | 1.60±0.59 | 1.86±0.23 | 1.30±0.20 | 1.32±0.20 | **0.91** ±0.15 | 1.15±0.10 |
| | CRN | 1.13±0.15 | 1.16±0.15 | 0.92±0.09 | 2.15±0.12 | **1.09** ±0.09 | 1.09±0.09 | **0.96** ±0.18 | 1.07±0.06 |
| | G-Net | 0.90±0.45 | 0.96±0.43 | 0.79±0.35 | 1.16±0.26 | **1.48** ±0.94 | **1.49** ±0.94 | **0.96** ±0.23 | 1.41±0.37 |
| | CT | 0.83±0.31 | 0.91±0.38 | 0.82±0.30 | 0.99±0.22 | 1.34±0.22 | 1.36±0.26 | 1.00±0.14 | 1.24±0.06 |
| ODE-D | A-SINDy | 0.11±0.00 | **0.23** ±0.00 | 0.24±0.00 | 0.25±0.00 | 1.46±0.00 | 1.47±0.00 | 1.40±0.00 | 1.56±0.00 |
| | A-WSINDy | 0.11±1.77e-04 | **0.23** ±2.45e-03 | 0.23±2.85e-03 | **0.23** ±2.94e-03 | NA | NA | NA | NA |
| | INSITE | **0.02** ±0.00 | **0.23**±0.00 | **0.23** ±0.00 | 0.24±0.00 | 1.42±0.00 | **0.95** ±0.00 | **0.82** ±0.00 | **0.90** ±0.00 |

Table 23: **Counterfactual normalized RMSE,** for 6-step ahead prediction of the benchmarks on each synthetic dataset detailed in appendix F, with BSV observation noise of $\epsilon = 1.0$. We quote 95% confidence intervals with each value, and all metrics are averaged over five random seed runs. Each PKPD underlying pharmacological model is simulated with different layers of BSV (table 2) for A-D. Our contribution is  shaded  below.

| | Method | eq. (5).A | eq. (5).B | eq. (5).C | eq. (5).D | eq. (6).A | eq. (6).B | eq. (6).C | eq. (6).D |
|---|---|---|---|---|---|---|---|---|---|
| LTE | MSM | 0.99±1.54e-16 | 2.46±0.00 | 3.03±0.00 | 4.16±0.00 | 2.61±0.00 | 2.61±0.00 | 1.99±0.00 | 2.13±0.00 |
| | RMSN | 1.87±0.33 | 3.60±1.07 | 3.21±0.46 | 3.77±1.28 | 1.30±0.20 | 1.35±0.25 | **0.91** ±0.13 | 1.14±0.09 |
| | CRN | 1.13±0.15 | 2.44±0.15 | 2.29±0.13 | 2.90±0.47 | **1.09** ±0.09 | 1.12±0.09 | **0.98** ±0.18 | 1.09±0.07 |
| | G-Net | 0.90±0.45 | 2.51±0.11 | 2.35±0.08 | 2.56±0.23 | **1.48** ±0.94 | **1.48** ±0.98 | **0.96** ±0.23 | 1.42±0.36 |
| | CT | 0.83±0.31 | 2.17±0.10 | 2.12±0.06 | 2.20±0.06 | 1.34±0.22 | 1.34±0.23 | 1.00±0.13 | 1.24±0.06 |
| ODE-D | A-SINDy | 0.11±0.00 | 2.06±0.00 | 2.05±0.00 | 2.07±0.00 | 1.46±0.00 | 1.47±0.00 | 1.40±0.00 | 1.56±0.00 |
| | A-WSINDy | 0.11±1.77e-04 | **2.05** ±4.19e-03 | **2.04** ±4.12e-03 | **2.07** ±5.26e-03 | NA | NA | NA | NA |
| | INSITE | **0.02** ±0.00 | 2.18±0.00 | 2.12±0.00 | 2.18±0.00 | 1.42±0.00 | **0.96** ±0.00 | **0.83** ±0.00 | **0.91** ±0.00 |

of magnitude increase in error, we still observe INSITE achieves a lower error than the longitudinal treatment effect models.

To investigate how INSITE and the baseline ODE discovery methods compare under different levels of model misspecification, we performed a complete re-run of our main experimental table with varying feature libraries. Here we went from using an overly-restricted feature library of $\mathcal{L} = \{1\}$ to a still misspecified feature library (for eq. (6).A-D, and the Cancer PKPD datasets) that is overly expressive of $\mathcal{L} = \{1, x_0, x_1, x_0^2, x_0 x_1, x_1^2, x_0^3, x_0^2 x_1, x_0 x_1^2, x_1^3\}$. The results are tabulated as: $\mathcal{L} = \{1\}$ in table 24, $\mathcal{L} = \{1, x_0, x_1\}$ in table 25, $\mathcal{L} = \{1, x_0, x_1, x_0 x_1\}$ in table 26, $\mathcal{L} = \{1, x_0, x_1, x_0^2, x_0 x_1, x_1^2\}$ in table 27, $\mathcal{L} = \{1, x_0, x_1, x_0^2, x_0 x_1, x_1^2, x_0^3, x_0^2 x_1, x_0 x_1^2, x_1^3\}$ in table 28. We observe the same pattern of a low error when the feature library is correctly specified (that is the underlying dataset was generated with features within the searching or discovery library set) and an order of magnitude increase in error or higher when the feature library is misspecified (for eq. (6).A-D, and the Cancer PKPD datasets).

An actionable insight is that the ODE discovery method should have a sufficiently expressive feature library to search over. Moreover, even if the underlying data-generating equation contains unique feature library terms, such as $\log(x_0), \log(x_1)$, that are not explicitly given, employing a polynomial feature library can serve as a robust approximation to the unidentified feature library set. This approach is supported by the Stone-Weierstrass theorem in function approximation theory, which posits that any continuous function defined over a closed interval can be uniformly approximated by a polynomial function with an arbitrarily close degree of precision (Stone, 1948).

Table 24: **Counterfactual normalized RMSE,** for 6-step ahead prediction of the benchmarks on each synthetic dataset detailed in appendix F, with all ODE discovery methods using a restricted feature library of $\mathcal{L}_{\text{INSITE}} = \{1\}$. We quote 95% confidence intervals with each value, and all metrics are averaged over ten random seed runs. Each PKPD underlying pharmacological model is simulated with different layers of BSV (table 2) for A-D. Our contribution is shaded below.

| | Method | eq. (5).A | eq. (5).B | eq. (5).C | eq. (5).D | eq. (6).A | eq. (6).B | eq. (6).C | eq. (6).D | Cancer PKPD |
|---|---|---|---|---|---|---|---|---|---|---|
| LTE | MSM | 0.99±8.37e-17 | 0.99±0.00 | 0.97±8.37e-17 | 2.09±0.00 | 2.55±0.13 | 2.55±0.13 | 2.06±0.16 | 2.11±0.04 | 2.30±0.12 |
| | RMSN | 1.92±0.24 | 1.94±0.23 | 1.68±0.19 | 1.91±0.18 | 1.23±0.15 | 1.25±0.15 | 1.10±0.18 | 1.10±0.11 | 1.04±0.17 |
| | CRN | 1.05±0.10 | 1.05±0.10 | 0.82±0.09 | 1.98±0.14 | 1.05±0.03 | 1.05±0.03 | 1.03±0.08 | 1.03±0.10 | 0.92±0.08 |
| | G-Net | 0.91±0.20 | 0.91±0.20 | 0.72±0.14 | 0.97±0.15 | 1.33±0.27 | 1.34±0.27 | 1.02±0.11 | 1.25±0.15 | 1.22±0.14 |
| | CT | 0.90±0.18 | 0.90±0.18 | 0.75±0.14 | 1.00±0.14 | 1.29±0.07 | 1.29±0.10 | 1.03±0.11 | 1.14±0.10 | 1.07±0.07 |
| ODE | A-SINDy | 28.63±0.00 | 28.63±2.68e-15 | 30.26±2.68e-15 | 28.58±0.00 | 20.51±0.00 | 20.51±0.00 | 18.33±2.68e-15 | 18.28±2.68e-15 | 17.36±0.00 |
| | INSITE | **28.43** ±2.68e-15 | **28.43** ±0.00 | **30.08** ±2.68e-15 | **28.33** ±2.68e-15 | **20.51** ±0.00 | **20.51** ±0.00 | **18.33** ±0.00 | **18.28** ±0.00 | **17.36** ±2.68e-15 |

Table 25: **Counterfactual normalized RMSE,** for 6-step ahead prediction of the benchmarks on each synthetic dataset detailed in appendix F, with all ODE discovery methods using a restricted feature library of $\mathcal{L}_{\text{INSITE}} = \{1, x_0, x_1\}$. We quote 95% confidence intervals with each value, and all metrics are averaged over ten random seed runs. Each PKPD underlying pharmacological model is simulated with different layers of BSV (table 2) for A-D. Our contribution is shaded below.

| | Method | eq. (5).A | eq. (5).B | eq. (5).C | eq. (5).D | eq. (6).A | eq. (6).B | eq. (6).C | eq. (6).D | Cancer PKPD |
|---|---|---|---|---|---|---|---|---|---|---|
| LTE | MSM | 0.99±8.37e-17 | 0.99±0.00 | 0.97±8.37e-17 | 2.09±0.00 | 2.55±0.13 | 2.55±0.13 | 2.06±0.16 | 2.11±0.04 | 2.30±0.12 |
| | RMSN | 1.92±0.24 | 1.94±0.23 | 1.68±0.19 | 1.91±0.18 | 1.23±0.15 | 1.25±0.15 | 1.10±0.18 | 1.10±0.11 | 1.04±0.17 |
| | CRN | 1.05±0.10 | 1.05±0.10 | 0.82±0.09 | 1.98±0.14 | 1.05±0.03 | 1.05±0.03 | 1.03±0.08 | 1.03±0.10 | 0.92±0.08 |
| | G-Net | 0.91±0.20 | 0.91±0.20 | 0.72±0.14 | 0.97±0.15 | 1.33±0.27 | 1.34±0.27 | 1.02±0.11 | 1.25±0.15 | 1.22±0.14 |
| | CT | 0.90±0.18 | 0.90±0.18 | 0.75±0.14 | 1.00±0.14 | 1.29±0.07 | 1.29±0.10 | 1.03±0.11 | 1.14±0.10 | 1.07±0.07 |
| ODE | A-SINDy | 0.82±0.00 | 0.82±0.00 | 0.64±0.00 | 0.87±0.00 | 0.98±0.00 | 0.98±1.54e-16 | 0.73±0.00 | 0.73±0.00 | 0.93±1.54e-16 |
| | INSITE | **0.47** ±0.00 | **0.47** ±0.00 | **0.36** ±0.00 | **0.49** ±7.71e-17 | **0.80** ±1.54e-16 | **0.80** ±1.54e-16 | **0.63** ±0.00 | **0.61** ±0.00 | **0.86** ±0.00 |

Table 26: **Counterfactual normalized RMSE,** for 6-step ahead prediction of the benchmarks on each synthetic dataset detailed in appendix F, with all ODE discovery methods using a restricted feature library of $\mathcal{L}_{\text{INSITE}} = \{1, x_0, x_1, x_0x_1\}$. We quote 95% confidence intervals with each value, and all metrics are averaged over ten random seed runs. Each PKPD underlying pharmacological model is simulated with different layers of BSV (table 2) for A-D. Our contribution is shaded below. This table is identical to table 3 and is included here for completeness.

| | Method | eq. (5).A | eq. (5).B | eq. (5).C | eq. (5).D | eq. (6).A | eq. (6).B | eq. (6).C | eq. (6).D | Cancer PKPD |
|---|---|---|---|---|---|---|---|---|---|---|
| LTE | MSM | 0.99±8.37e-17 | 0.99±0.00 | 0.97±8.37e-17 | 2.09±0.00 | 2.55±0.13 | 2.55±0.13 | 2.06±0.16 | 2.11±0.04 | 2.30±0.12 |
| | RMSN | 1.92±0.24 | 1.94±0.23 | 1.68±0.19 | 1.91±0.18 | 1.23±0.15 | 1.25±0.15 | 1.10±0.18 | 1.10±0.11 | 1.04±0.17 |
| | CRN | 1.05±0.10 | 1.05±0.10 | 0.82±0.09 | 1.98±0.14 | 1.05±0.03 | 1.05±0.03 | 1.03±0.08 | 1.03±0.10 | 0.92±0.08 |
| | G-Net | 0.91±0.20 | 0.91±0.20 | 0.72±0.14 | 0.97±0.15 | 1.33±0.27 | 1.34±0.27 | 1.02±0.11 | 1.25±0.15 | 1.22±0.14 |
| | CT | 0.90±0.18 | 0.90±0.18 | 0.75±0.14 | 1.00±0.14 | 1.29±0.07 | 1.29±0.10 | 1.03±0.11 | 1.14±0.10 | 1.07±0.07 |
| ODE | A-SINDy | 0.11±0.00 | 0.11±0.00 | 0.13±2.09e-17 | 0.15±0.00 | 1.45±0.03 | 1.45±0.03 | 1.40±0.01 | 1.51±0.09 | 1.23±0.13 |
| | INSITE | **0.02** ±2.62e-18 | **0.03** ±0.00 | **0.04** ±0.00 | **0.05** ±5.23e-18 | **0.94** ±0.05 | **0.94** ±0.05 | **0.84** ±0.04 | **0.87** ±0.08 | **0.79** ±8.37e-17 |

Table 27: **Counterfactual normalized RMSE,** for 6-step ahead prediction of the benchmarks on each synthetic dataset detailed in appendix F, with all ODE discovery methods using a restricted feature library of $\mathcal{L}_{\text{INSITE}} = \{1, x_0, x_1, x_0^2, x_0x_1, x_1^2\}$. We quote 95% confidence intervals with each value, and all metrics are averaged over ten random seed runs. Each PKPD underlying pharmacological model is simulated with different layers of BSV (table 2) for A-D. Our contribution is shaded below.

| | Method | eq. (5).A | eq. (5).B | eq. (5).C | eq. (5).D | eq. (6).A | eq. (6).B | eq. (6).C | eq. (6).D | Cancer PKPD |
|---|---|---|---|---|---|---|---|---|---|---|
| LTE | MSM | 0.99±8.37e-17 | 0.99±0.00 | 0.97±8.37e-17 | 2.09±0.00 | 2.55±0.13 | 2.55±0.13 | 2.06±0.16 | 2.11±0.04 | 2.30±0.12 |
| | RMSN | 1.92±0.24 | 1.94±0.23 | 1.68±0.19 | 1.91±0.18 | 1.23±0.15 | 1.25±0.15 | 1.10±0.18 | 1.10±0.11 | 1.04±0.17 |
| | CRN | 1.05±0.10 | 1.05±0.10 | 0.82±0.09 | 1.98±0.14 | 1.05±0.03 | 1.05±0.03 | 1.03±0.08 | 1.03±0.10 | 0.92±0.08 |
| | G-Net | 0.91±0.20 | 0.91±0.20 | 0.72±0.14 | 0.97±0.15 | 1.33±0.27 | 1.34±0.27 | 1.02±0.11 | 1.25±0.15 | 1.22±0.14 |
| | CT | 0.90±0.18 | 0.90±0.18 | 0.75±0.14 | 1.00±0.14 | 1.29±0.07 | 1.29±0.10 | 1.03±0.11 | 1.14±0.10 | 1.07±0.07 |
| ODE | A-SINDy | 0.11±1.93e-17 | 0.11±1.93e-17 | 0.12±1.93e-17 | 0.13±0.00 | 0.90±0.00 | 0.90±1.54e-16 | 0.82±0.00 | 0.85±1.54e-16 | 1.71±0.00 |
| | INSITE | **0.02** ±0.00 | **0.03** ±0.00 | **0.04** ±0.00 | **0.04** ±0.00 | **0.67** ±0.00 | **0.89** ±0.00 | **0.66** ±0.00 | **0.74** ±0.00 | **1.29** ±0.00 |

Table 28: **Counterfactual normalized RMSE,** for 6-step ahead prediction of the benchmarks on each synthetic dataset detailed in appendix F, with all ODE discovery methods using a restricted feature library of $\mathcal{L}_{\text{INSITE}} = \{1, x_0, x_1, x_0^2, x_0x_1, x_1^2, x_0^3, x_0^2x_1, x_0x_1^2, x_1^3\}$. We quote 95% confidence intervals with each value, and all metrics are averaged over ten random seed runs. Each PKPD underlying pharmacological model is simulated with different layers of BSV (table 2) for A-D. Our contribution is shaded below.

| | Method | eq. (5).A | eq. (5).B | eq. (5).C | eq. (5).D | eq. (6).A | eq. (6).B | eq. (6).C | eq. (6).D | Cancer PKPD |
|---|---|---|---|---|---|---|---|---|---|---|
| LTE | MSM | 0.99±8.37e-17 | 0.99±0.00 | 0.97±8.37e-17 | 2.09±0.00 | 2.55±0.13 | 2.55±0.13 | 2.06±0.16 | 2.11±0.04 | 2.30±0.12 |
| | RMSN | 1.92±0.24 | 1.94±0.23 | 1.68±0.19 | 1.91±0.18 | 1.23±0.15 | 1.25±0.15 | 1.10±0.18 | 1.10±0.11 | 1.04±0.17 |
| | CRN | 1.05±0.10 | 1.05±0.10 | 0.82±0.09 | 1.98±0.14 | 1.05±0.03 | 1.05±0.03 | 1.03±0.08 | 1.03±0.10 | 0.92±0.08 |
| | G-Net | 0.91±0.20 | 0.91±0.20 | 0.72±0.14 | 0.97±0.15 | 1.33±0.27 | 1.34±0.27 | 1.02±0.11 | 1.25±0.15 | 1.22±0.14 |
| | CT | 0.90±0.18 | 0.90±0.18 | 0.75±0.14 | 1.00±0.14 | 1.29±0.07 | 1.29±0.10 | 1.03±0.11 | 1.14±0.10 | 1.07±0.07 |
| ODE | A-SINDy | 0.11±1.93e-17 | 0.11±1.93e-17 | 0.12±1.93e-17 | 0.13±0.00 | 0.90±1.54e-16 | 0.90±1.54e-16 | 0.80±0.00 | 0.96±1.54e-16 | 1.68±3.08e-16 |
| | INSITE | **0.02** ±0.00 | **0.03** ±0.00 | **0.04** ±0.00 | **0.04** ±0.00 | **0.85** ±1.54e-16 | **0.85** ±0.00 | **0.75** ±0.00 | **0.70** ±0.00 | **1.00** ±1.54e-16 |

## K.7 INSITE ALSO EMPIRICALLY WORKS FOR IRREGULARLY SAMPLED DATA

INSITE also supports irregularly sampled time series data, as its underlying ODE discovery method of SINDy supports discovering ODEs from irregularly sampled time series data (Brunton et al., 2016; Goyal & Benner, 2022). We further empirically verify this by re-running SINDy and INSITE across all datasets, now with the irregularly sampled datasets, where we randomly sub-sampled irregularly, excluding 10% of the original observations along the time dimension. The subsequent results, conducted across ten random seeds, are outlined in table table 29 below. We observe that INSITE and SINDy still achieve good performance (hence low prediction error).

Table 29: **Counterfactual normalized RMSE,** for 6-step ahead prediction of the benchmarks on each synthetic dataset detailed in appendix F. We quote 95% confidence intervals with each value, and all metrics are averaged over ten random seed runs. Datasets are sub-sampled irregularly, excluding 10% of the original observations along the time dimension. Each PKPD underlying pharmacological model is simulated with different layers of BSV (table 2) for A-D. Our contribution is shaded below.

| | Method | eq. (5).A | eq. (5).B | eq. (5).C | eq. (5).D | eq. (6).A | eq. (6).B | eq. (6).C | eq. (6).D | Cancer PKPD |
|---|---|---|---|---|---|---|---|---|---|---|
| ODE | A-SINDy | 0.11±1.93e-17 | 0.11±1.93e-17 | 0.12±1.93e-17 | 0.13±0.00 | 0.90±1.54e-16 | 0.90±1.54e-16 | 0.80±0.00 | 0.96±1.54e-16 | 1.68±3.08e-16 |
| | INSITE | **0.02** ±0.00 | **0.03** ±0.00 | **0.04** ±0.00 | **0.04** ±0.00 | **0.85** ±1.54e-16 | **0.85** ±0.00 | **0.75** ±0.00 | **0.70** ±0.00 | **1.00** ±1.54e-16 |

## K.8 INSITE CAN ALSO WORK WITH MORE CHALLENGING OBSERVATION FUNCTIONS $g(x)$

Although the observation function $g$ that describes the treatment outcome as a function of the observed features is often prespecified by the user as it often models the outcome as one of the features ($g(\mathbf{x}) = x_j$), e.g., the tumor volume, other more complex forms of $g$ that are not prespecified are also of interest. To investigate whether INSITE can work under these more challenging settings of having $g$ be a non-linear function, we empirically investigated this by implementing three more challenging forms of $g(x)$ representing more complex settings. These are, (1) an exponential function $g(\mathbf{x}) = \exp(x_j)$ that often models rapid changes or growth, commonly found in many biological and economic systems (Johnson et al., 2019), (2) a quadratic function $g(\mathbf{x}) = x_j^2$ that models a non-linear relationship in systems (Gasull et al., 2004), and (3) a trigonometric function $g(\mathbf{x}) = \sin(x_j)$ that often models periodic components within oscillatory dynamics, which are commonly found in natural and engineering systems (Datta & Mohan, 1995). For each of these new observation functions we re-ran SINDy and INSITE across all datasets, conducted across five random seeds, with the results tabulated in table 30 below. We observe that INSITE and SINDy can still achieve acceptable performance. To implement this experimentally, before applying the observation function after sampling the dataset initially, we normalized the dataset using a min-max scaler (normalizer) so the min-max was from 0 to 1, applied $g(x)$ then un-normalized to return the min-max back to the original range of the dataset—we did to help numerical stability, and such a normalization step is common when pre-processing datasets.

Table 30: **Counterfactual normalized RMSE,** for 6-step ahead prediction of the benchmarks on each synthetic dataset detailed in appendix F. We quote 95% confidence intervals with each value, and all metrics are averaged over five random seed runs. We modify the observation function $g(x)$, to be one of the following $g(\mathbf{x}) = \{x_j, \exp(x_j), x_j^2, \sin(x_j)\}$. Each PKPD underlying pharmacological model is simulated with different layers of BSV (table 2) for A-D. Our contribution is shaded below.

| $g(x)$ | Method | eq. (5).A | eq. (5).B | eq. (5).C | eq. (5).D | eq. (6).A | eq. (6).B | eq. (6).C | eq. (6).D | Cancer PKPD |
|---|---|---|---|---|---|---|---|---|---|---|
| $x$ | A-SINDy | 0.11±0.00 | 0.11±0.00 | 0.13±0.00 | 0.15±0.00 | 1.46±0.00 | 1.47±0.00 | 1.40±0.00 | 1.56±0.00 | 1.29±0.00 |
| | INSITE | 0.02 ±0.00 | 0.03 ±0.00 | 0.04 ±0.00 | 0.05 ±0.00 | 0.96 ±0.00 | 0.96 ±0.00 | 0.82 ±1.54e-16 | 0.90 ±0.00 | 0.79 ±0.00 |
| $\exp(x)$ | A-SINDy | 0.75±0.02 | 0.75±0.02 | 0.74±0.03 | 0.76±0.06 | 0.76 ±0.34 | 0.76 ±0.34 | 1.21 ±1.34 | 1.02 ±0.57 | 0.94 ±0.20 |
| | INSITE | 0.59 ±0.03 | 0.59 ±0.02 | 0.61 ±0.02 | 0.62 ±0.05 | 0.67 ±0.30 | 0.68 ±0.32 | 1.08 ±1.31 | 0.90 ±0.59 | 0.78 ±0.13 |
| $x^2$ | A-SINDy | 0.10±5.85e-03 | 0.12±3.50e-03 | 0.12±3.63e-03 | 0.12±0.01 | 1.16 ±1.03 | 1.15 ±1.03 | 2.11 ±3.58 | 1.69 ±2.02 | 0.57 ±0.10 |
| | INSITE | 0.03 ±1.98e-03 | 0.09 ±2.67e-03 | 0.09 ±2.40e-03 | 0.09 ±6.26e-03 | 1.12 ±1.04 | 1.11 ±1.04 | 2.04 ±3.57 | 1.63 ±2.03 | 0.52 ±0.08 |
| $\sin(x)$ | A-SINDy | 2.42±0.07 | 2.42±0.07 | 2.31±0.07 | 2.42 ±0.40 | 1.62 ±0.43 | 1.62 ±0.43 | 2.30 ±1.86 | 1.99 ±0.69 | 2.00 ±0.46 |
| | INSITE | 2.09 ±0.10 | 2.09 ±0.10 | 2.06 ±0.08 | 2.22 ±0.25 | 1.46 ±0.36 | 1.41 ±0.42 | 1.96 ±1.78 | 1.72 ±0.76 | 1.59 ±0.29 |

## L    RECONCILING ASSUMPTIONS IN TABLE 1

Let us discuss in detail how the assumptions set for treatment effects (assum. 2.1 to 2.3) and ODE discovery (assum. 3.1 to 3.3) compare as reported in table 1 in section 4.

*existence & uniqueness (assum. 3.1)* o‐ ‐ ‐ ‐ o *overlap (assum. 2.2)* and
*functional spaces (assum. 3.3)* o‐ ‐ ‐ ‐ o *overlap (assum. 2.2)*.
These assumptions do not map one-to-one. However, they do serve a similar purpose. Essentially, overlap allows us to assume that there is always a very similar sample (in terms of covariates), such that we have access to a full range of potential outcomes such that we can estimate eq. (1) for every treatment and covariates. When violated, Gelman & Hill (2006) pointed out that one has to rely much more on correct model specification, i.e., rather than assuming overlap, one may resort to the correct model specification instead. Assum. 3.1 and 3.3 tell us exactly this: the former assumes that there exists one discoverable function (i.e., one that can be correctly specified), while the latter tells us that the correct function is comprised of the token set that is provided to the ODE discovery method. Combined, these assumptions should result in a correctly specified model, which can then be used to relax an overlap assumption.

*observability (assum. 3.2)* o——o *consistency (assum. 2.1)* and
*observability (assum. 3.2)* o——o *overlap (assum. 2.3)*.
While the previous assumptions did not map one-to-one, we find that assum. 3.2 *does* map one-to-one with assum. 2.1 and 2.3. With eq. (3) we combine covariates, (potential) outcomes and treatments into one dynamical system. Given this, assum. 3.2 states that every variable of the complete ODE is observed in the dataset. These variables include the potential outcomes (which corresponds to assum. 2.1) as well as all the covariates (corresponding to assum. 2.3). One could make the argument that assum. 3.2 is even stricter than combining assum. 2.1 and 2.3 as the latter does not mention observing treatment indicators. While there is work on missingness in treatment indicators (such as (Kuzmanovic et al., 2023)) it is still an exception to the rule.

## M    SCALE OF THE OUTCOMES

In the following Table 31, we provide the scale of outcomes for each of the datasets used in all experiments.

Table 31: Scale of outcomes for each of the datasets used in all experiments.

| Dataset | mean | std | min | 25% | 50% | 75% | max |
|---|---|---|---|---|---|---|---|
| Cancer PKPD | 6.93553 | 49.099786 | 0.0 | 0.000263 | 0.011018 | 0.179058 | 1150.34651 |
| **Eq. (5).A** | 5.225302 | 7.873385 | 0.006063 | 0.48826 | 1.723761 | 6.418705 | 49.480554 |
| **Eq. (5).B** | 5.225425 | 7.873301 | -0.00044 | 0.489809 | 1.726515 | 6.415482 | 49.481121 |
| **Eq. (5).C** | 4.376515 | 7.459998 | -0.010424 | 0.232287 | 1.039811 | 4.889878 | 49.481121 |
| **Eq. (5).D** | 4.992425 | 8.127416 | -0.012978 | 0.300557 | 1.308329 | 5.801108 | 49.848292 |
| **Eq. (6).A** | 7.772205 | 53.479708 | 0.0 | 0.000157 | 0.012505 | 0.226499 | 1144.486013 |
| **Eq. (6).B** | 7.772162 | 53.479744 | -0.034542 | 0.001075 | 0.016917 | 0.22768 | 1144.490737 |
| **Eq. (6).C** | 7.31387 | 51.156823 | -0.034717 | 0.000713 | 0.015301 | 0.189101 | 1150.336353 |
| **Eq. (6).D** | 6.935527 | 49.099739 | -0.034994 | 0.001075 | 0.01645 | 0.178607 | 1150.334351 |

## N    HOW TO CHOOSE AN ODE DISCOVERY METHOD

In the following, we first explain how Sparse Identification of Nonlinear Dynamics (SINDy) (Brunton et al., 2016) works and the necessary assumptions it imposes, then second, a discussion on alternative ODE discovery methods and how to choose a suitable one for a given problem setting.

**Sparse Identification of Nonlinear Dynamics (SINDy)** (Brunton et al., 2016), a data-driven framework that aims to discover the governing dynamical system equations directly from time-series data. The algorithm works by iteratively performing sparse regression on a library of candidate functions to identify the sparsest yet most accurate representation of the dynamical system.

SINDy makes an explicit sparse assumption that the dynamics are governed by only a few important terms so that the discovered equation is sparse in the space of possible functions. Such sparse parsimonious ODEs balance accuracy with model complexity, which aids in avoiding overfitting. SINDy considers a simpler problem of modelling

$$\frac{d\boldsymbol{x}(t)}{dt} = \dot{\boldsymbol{x}}(t) = \boldsymbol{f}(\boldsymbol{x}(t)), \tag{13}$$

where the function $\boldsymbol{f}$ is an equation with a few numbers of terms, that is a sparse equation in the space of possible functions. SINDy uses sparse regression to identify the terms in $\boldsymbol{f}$, which also helps it mitigate noise. It collects trajectories and approximates their derivative $\dot{\boldsymbol{x}}(t)$ numerically, from which it then iteratively solves eq. (13) through sparse regression, where it constructs a feature library of candidate non-linear functions $\boldsymbol{\Theta}(\boldsymbol{X})$, where $\boldsymbol{X} = [\boldsymbol{x}^T(t_1), \boldsymbol{x}^T(t_2), \ldots, \boldsymbol{x}^T(t_m)]^T$. For example, the candidate non-linear functions could consist of constant, trigonometric or polynomial terms:

$$\boldsymbol{\Theta}(\boldsymbol{X}) = \begin{bmatrix} | & | & | & | & & | & | & | & | & \\ 1 & \boldsymbol{X} & \boldsymbol{X}^{P_2} & \boldsymbol{X}^{P_3} & \cdots & \sin(\boldsymbol{X}) & \cos(\boldsymbol{X}) & \sin(2\boldsymbol{X}) & \cos(2\boldsymbol{X}) & \cdots \\ | & | & | & | & & | & | & | & | & \end{bmatrix} \tag{14}$$

Given this feature library, we can then formulate sparse vectors of coefficients $\boldsymbol{\Xi} = [\boldsymbol{\xi}_1 \quad \boldsymbol{\xi}_2 \quad \cdots \quad \boldsymbol{\xi}_n]$ which determine which features are active:

$$\dot{\boldsymbol{X}} = \boldsymbol{\Theta}(\boldsymbol{X})\boldsymbol{\Xi} \tag{15}$$

Such a method is poorly suited for a large state dimension due to the factorial growth of $\boldsymbol{\Theta}$ (Brunton et al., 2016). Furthermore, such an approach relies on choosing the right coordinates and basis for the best representation of the dynamics. Moreover, this also assumes that given the non-linear candidate function library, we can recover an equation that linearly decomposes into these candidate functions.

**Alternative ODE discovery methods**. A range of alternative methods exist, including symbolic regression (Schmidt & Lipson, 2009) and other ODE discovery methods that better handle noise, such as WSINDy (Messenger & Bortz, 2021).

**Symbolic Regression**. It can similarly discover an underlying ODE, however, it performs combinatorial search (Schmidt & Lipson, 2009), which can be slower; however, it can discover more non-linear relationships amongst the candidate library terms, such as one feature divided by another.

**Feature Learning Methods in ODE Discovery**. Feature learning methods, particularly those employing neural networks like Recurrent Neural Networks (RNNs) and Long Short-Term Memory (LSTM) networks, offer a data-driven approach for ODE discovery. These methods automatically extract relevant features from time-series data, adapting to complex dynamics without predefined candidate functions (Lechner & Hasani, 2020).

Unlike Sparse Identification of Nonlinear Dynamics (SINDy), which uses a predetermined library of functions, feature learning methods learn these features directly from data, offering flexibility in modelling non-linear relationships in systems with unknown or complicated dynamics.

However, the use of neural networks in ODE discovery often leads to models that lack interpretability and require substantial data for effective training, posing challenges in extracting physical laws or governing equations.

**Considerations for Use**. The choice between feature learning methods and others like SINDy or Symbolic Regression depends on system complexity, data availability, and interpretability needs. Feature learning is advantageous in complex, data-rich environments but less so in scenarios where interpretability and simplicity are priorities. When employing methods such as SINDy, a user could start with a feature library of polynomial terms and slowly increase the polynomial order till the empirical generalization accuracy was suitable for a given treatment effect setting. Other suitable terms commonly found in PKPD models include $\exp, \log$ terms and non-linear combinations of these terms (Geng et al., 2017).

## O    NUMERICAL SOLVER AND PARAMETERS

The numerical method (solver) used for INSITE and the other ODE discovery methods (e.g., SINDy) is that of the *Euler method*, which is a first-order numerical algorithm for solving ODEs given

an initial value. This uses a time step size of $\delta_t = 0.166$ seconds, which was sufficient for our experiments. This numerical method is used with the same settings across all methods, including INSITE. We kindly note that first, the ODE discovery methods discover an ODE, and then at inference time, when we wish to determine future values, we forward simulate the initial observed value using the discovered ODE and this Euler numerical solver to estimate future values at future times. We also kindly highlight further implementation details can be found in appendices appendices F, G and I.

