# OpenReview forum: "ODE Discovery for Longitudinal Heterogeneous Treatment Effects Inference"
_ICLR.cc/2024/Conference — ICLR 2024 spotlight_

### Official Review · Reviewer_pnuM · 2023-10-18

**Soundness:** 2 fair
**Presentation:** 1 poor
**Contribution:** 2 fair
**Rating:** 5
**Confidence:** 4

**Summary:**

The authors propose to solve a task of ODE discovery in the treatment effect estimation setting. Although the task is not sufficiently explained in the main paper, I believe it is to construct an ODE model with simple expressions (e.g., log, exp, sin) from time series data at continuous time stamps. Thus, the paper seems to tackle a novel task. However, it is completely unclear because the motivation is not sufficiently described, the paper is not well structured, the problem setting is not clearly presented, and many important descriptions are moved to Appendix.

**Strengths:**

- Introducing the ODE discovery in the context of treatment effect estimation seems novel.

**Weaknesses:**

(A) Problem setting is unclear

- First of all, in Section 1, please clearly state that what the ODE discovery task is and what its goal is. Otherwise, the readers cannot understand why it yields interpretability.

- It is unclear whether time $t$ is defined as continuous or discrete. For instance, in Section 2, “at $t \in [T]$” seems to say that the value of time $t$ is discrete and it is included in index set $[T]$. However, for example in Assumption 2.2, “time point $t \in [0, T]$” seems to say that it is continuous.

- In Section 3, “we assume that time-varying features and outcomes … are discrete measurements of underlying continuous trajectories…” is unclear. I could not understand what is discrete and what is continuous. Do you imply that trajectories are defined at continuous time and their observations are measured at discrete time? Or do you mean that the values of features and outcomes are continuous in ODE models, but their observed values are discrete? Similarly, “discrete (or continuous) treatment plan” is unclear to me. This makes the paper extremely difficult to follow.

- Eq. (3) introduces function $g$, and the authors say “it is known and is often assumed to be the identity.” Why?

- Please separate the paragraph at “The goal of ODE discovery is …” in Section 3. Please do not mix the proposed problem setting and the existing one in the same paragraph, which is hard to follow.


(B) The technical soundness is unclear

- **Assumptions**: I could not understand why the authors say that “Assumption 2.2 can be relaxed with Assumptions 3.1 and 3.3.”  I could not see what the point of Discrepancy 1 is. It simply says that “the assumptions in treatment effect estimation do not correspond one-to-one to those in ODE discovery,” which seems trivial. It is hard to follow.

- **The soundness of employing ODE discovery methods**: Although the proposed method employs the existing ODE discovery method (called SINDy), it is unclear what this method does and how it works. What is the advantages and the limitations of this method? How can we choose the existing methods?

- **Ability to express the variability of each subject**: The authors seem to simply change the constant values in simple ODE models to express the variability across subjects. Can we really always express the treatment and the outcome curves with such a simple strategy, given that the continuous time series data often involve complex dynamics?

(C) The difference from existing ODE-based treatment effect estimation methods is unclear in the main paper

- Please briefly summarize the description of “Dynamical systems, ODEs, and treatment effects” in Appendix B.1 in the main paper. This will make the motivation of this paper clearer.

**Questions:**

See all questions in (A), (B), and (C), in particular, (B).

---

> ### Author Response · Authors · 2023-11-15
> **Thank you for your review (1/2)**
>
> _We thank the reviewer for their time and helpful comments. Our rebuttal below follows the structure of your review. To facilitate further discussion, we have also stated how each specific comment changed our paper_
>
>
> **(A) Problem setting.** Thank you for highlighting areas where our problem setting needed clarification. We have made the following revisions:
>
> 1. In Section 1, we now explicitly define the goal of ODE discovery as "finding underlying closed-form concise ODEs from observed trajectories," clarifying how these transparent equations lead to interpretability.
>
> 2. To clarify the nature of time in our model, Section 2 has been updated to specify time as a continuous variable, represented as $ t \in [0, T] $, to eliminate any confusion. We also clarify explicitly that $T$ is the time horizon.
>
> 3. In Section 3, we have refined our explanation: trajectories are indeed defined in continuous time, while observations are made at discrete time points (at possibly varying intervals). We also clarified the concept of "discrete (or continuous) treatment plan" in relation to the set of allowed treatments $\mathcal{A}$.
>
> 4. The role of $g$ in Eq. (3) is to describe the treatment outcome as a function of the observed features. Thus, knowing $g$ is the same as just specifying the outcome variable. It is often assumed to correspond to one of the features, e.g., the tumor volume. We have clarified this point. For a detailed discussion about Eq. (3), please take a look at Appendix C.
>
> 5. We have separated the paragraphs in Section 3 to distinctly delineate our proposed problem setting from existing ones, enhancing readability.
>
> These revisions aim to make the problem setting clearer and the paper easier to follow.
>
> **_Actions taken:_** Revisions 1-5 (outlined above) are applied to our paper.
>
>
> `rebuttal continues in next comment...`

---

> > ### Author Response · Authors · 2023-11-15
> > **Thank you for your review (2/2)**
> >
> > `... rebuttal continued`
> >
> > **(B) Technical soundness.** We respond to each item separately below:
> >
> > _Assumptions_ - Assumptions 3.1 and 3.3 are assumptions on statistical model specification. Essentially, through Assumptions 3.1 and 3.3, we limit the hypothesis class of possible models sufficiently enough that we can accept some violation of overlap.
> >
> > As discussed in Gelman and Hill (2006) [B], with stronger specification assumptions, one can extrapolate to areas of non-overlap (i.e., violations of Assumption 2.2). For an illustrative example, consider the linear setting. A linear assumption is a very strict assumption on the model specification. When respected, one can easily extrapolate to areas that are not covered by the sample, such as a non-overlapping region in the data space.
> >
> > With our framework, we adopt Gelman and Hill’s argument to an ODE setting and make it explicit through Assumptions 3.1 and 3.3. For more information about the Gelman and Hill argument, we refer the reviewer to  Chapter 10.1, "Imbalance and lack of complete overlap", in their book: _Data analysis using regression and multilevel/hierarchical models._ For examples from their book, we refer to Chapter 10.4 (where our linear example above is borrowed from).
> >
> > **_Actions taken:_** We included a paragraph in Appendix K dedicated to the above discussion.
> >
> >
> > _The soundness of employing ODE discovery methods_ - We have added an additional appendix M that describes SINDy, how it works, and how to choose other existing methods. This further expands on the original method description of SINDy in Appendix F.
> >
> > **_Actions taken:_** We include a new Appendix M to explain SINDy and other possible ODE learners.
> >
> >
> > _Ability to express the variability of each subject_ – We wish to clarify that INSITE (the presented method) serves only as an example of transforming SINDy to a TE method using our general purpose framework. INSITE is a relatively simple but effective method that recovers underlying ODEs in a TE setting.
> >
> > In Table 2, we give an overview of possible between-subject-variability (BSV) manifestations. We note that the most complex setting D (last row) is very general. It assumes that there are latent factors ($P$) that depend on the baseline covariates ($V$) and influence the dynamics of $X$ (as opposed to a deterministic relationship between $V$ and $X$ as in setting C). Our classification does not specify the exact form of the relationship between these variables.
> >
> > INSITE tackles this setting by assuming that the latent factors correspond to the numerical parameters of the discovered ODE and whose distributions depend on the static covariates (this is a standard assumption in pharmacology [C]). It then fits them using the initial part of the observed trajectory. Standard ODE methods (without adaptations) can directly tackle only BSV levels A and B (please take a look at Figure 2).
> >
> > While effective, we make no claim that INSITE is applicable in every scenario. Variability across subjects can be expressed as parameter discrepancies (which is what INSITE assumes), but there can be other manifestations. In those alternative scenarios, one may need to resort to different ODE learners. Like in other treatment effects scenarios, choosing a correct learner is a consideration best performed by the acting domain expert.
> >
> > **_Actions taken:_** We now provide a summary of Table 2 in the main text.
> >
> >
> > **\(C\) ODE-based TE inference and ODE-discovery for TE inference.** Thank you for your suggestion. We have now included a summary of the relevant descriptions from Appendix B.1 in the main paper, outlining the distinctions between our method and existing ODE-based treatment effect estimation methods. This addition aims to clarify the motivation and unique contributions of our work.
> >
> > **_Actions taken:_** A summary of B.1 is included in the main text.
> >
> >
> > **_We thank the reviewer once more for their insightful review. We believe our paper is improved as a direct result of your comments. Should there be any more questions, please ask._**
> >
> > ## References
> >
> > [A] Edward L Ince. Ordinary differential equations. Courier Corporation, 1956.
> >
> > [B] Andrew Gelman and Jennifer Hill. _Data analysis using regression and multilevel/hierarchical models._ Cambridge University Press, 2006.
> >
> > [C] Diane R. Mould and Richard Upton. Basic Concepts in Population Modeling, Simulation, and Model-Based Drug Development. CPT: Pharmacometrics & Systems Pharmacology, 2012.

---

> ### Author Response · Authors · 2023-11-21
> **Final Call for Reviewer Engagement: Clarifications and Updates Submitted**
>
> Thank you once again for your invaluable insights and time dedicated to the review process!
>
> If there were any leftover concerns, we would sincerely appreciate the opportunity to clarify them—before the discussion period for the authors ends in one day. We believe our responses (Nov 15) have addressed in detail the full set of questions you raised:
>
> * Improving the clarity of the problem setting; see **Response (A)**
> * Exposition on technical soundness and clarifications; see **Response (B)**
> * Clarifications between existing ODE-based TE methods; see **Response (C)**
>
> These are paired with corresponding updates to the submission (Nov 15), including extensive new additional experimental results, discussions, and clarifications—which include two **new** experimental appendices **appendix J.7 and J.8**, entitled _"INSITE also empirically works for irregularly sampled data"_ and _"INSITE can also work with more challenging observation functions $g(x)$"_ respectively.
>
> We would appreciate it if the reviewer kindly let us know if there were any further questions in the very limited time remaining of one day. We are eager to do our utmost to address them!
>
> Thank you,
>
> Paper 5372 authors

---

> > ### Comment · Reviewer_pnuM · 2023-11-23
> > **Thank you for your response**
> >
> > I appreciate the authors' efforts for addressing the reviewers' concerns.
> >
> > However, as with Reviewer WcGm, I have a concern that the manuscript seems to be extremely updated, which is far from the original submitted version. Given the fact that such a too significant revision is needed, I believe that the paper seems to require another reviewing process to carefully confirm the validity of the revision.

---

> ### Author Response · Authors · 2023-11-23
> **Revised Changes and Clarifications in Response to Reviewer Feedback**
>
> Thank you once again for your constructive comments that have ultimately improved the paper, and your time dedicated to the review process!
>
> Our initial change highlighting inadvertently compared the current version of the paper to an old git commit, leading to an overestimation of the changes made. We have now corrected this by highlighting only the text changed during the author discussion period, compared to the original submission (Sep 28). These changes can be verified using the OpenReview pdf difference tool. We find our changes to the paper to be minimal, and inline with addressing the reviewers comments—to be comprehensive we list out **all** these changes below, and their type:
>
> * Abstract: _Clarified_ ODE discovery methods discover an equation, by changing "human-readable equation" to "equation".
> * Introduction: _Clarified_ the goal of ODE discovery with the addition of “the goal of ODE discovery is finding underlying closed-form concise ODEs from observed trajectories.”
> * Introduction: _Clarified_ ODE discovery methods discover an equation, by changing "human-readable equation" to "equation".
> * Introduction: _Clarified_ contribution using ODE discovery methods by “While other TE inference methods have used ideas from ODE literature, none have focused on ODE discovery, we have devoted appendix B.1 to explain this subtle but important difference.”
> * Section 2. _Clarified_ time to be continuous “$t \in [0,T]$, with $T$ the time horizon”.
> * Section 3. Renamed section to “Underpinnings of ODE discovery”
> * Section 3. Added additional _clarification_ and example of how and where ODEs are used by “There are many fields which already use ODEs as a formal language to express time dynamics. One such example is pharmacology, where a large portion of literature is dedicated to recover an ODE from observational data. The found ODE is then used to reason about possible treatments and disease progression (Geng et al., 2017; Butner et al., 2020).”
> * Section 3. _Clarified_ $g$ as “$g$ is prespecified by the user and often models the outcome as one of the features ($g(\mathbf{x}) = x_j$), e.g., the tumor volume.”
> * Section 3. _Clarified_ ODE trajectories by “Of course, the reliance on such a dataset implies that, while the trajectories are defined in continuous time, they are observed in discrete time”.
> * Section 4.1. _Clarified_ the four different treatment types by “In summary, there are 4 different treatment types: binary, categorical, multiple, and continuous treatments (Bica et al., 2020). Each of them can be a static or dynamic treatment, which are either constant or change during a trajectory, respectively. This results in 8 scenarios, which we model in two ways: either the ODE changes for each treatment option; or the treatment is part of the starting condition. These are all outlined in table 6 in appendix D.”
> * Section 7. _Added_ a future work of “learning piece-wise continuous ODE systems (Wang et al., 2023)”
> * Ethics statement. _Clarified_ the limitations with listing three limitations that a user should take into account before using the method in practice.
> * _New appendix J.7_ on empirical results to show that “INSITE also empirically works for irregularly sampled data”.
> * _New appendix J.8_ on empirical results to show that “INSITE can also work with more challenging observation functions $g(x)$”.
> * _New appendix M_ on “How to choose an ODE discovery method”.
> * _New appendix N_ on “Numerical Solver and Parameters”
>
> In summary the total changes to the paper during the author rebuttal discussion period are only _additional clarification changes_ or the _addition of additional empirical results for special settings_. Each change directly addresses feedback from the review process, enhancing the paper’s clarity, depth, and practical relevance. We assure you that the core arguments, methodology, and results remain unchanged. The fundamental nature and validity of our work have not been altered; rather, the paper has been refined to better communicate its value.
>
> As the author discussion period draws to a close, we remain open and responsive to any further questions or concerns. Your feedback has been invaluable, and we are committed to addressing any remaining issues promptly.
>
> We hope this clarifies your concern and you will kindly consider accepting the paper. Thank you once again for your invaluable insights and the opportunity to improve our manuscript.
>
> Kind regards,
>
> Paper 5372 authors

---

### Official Review · Reviewer_h9Ux · 2023-10-27

**Soundness:** 4 excellent
**Presentation:** 4 excellent
**Contribution:** 4 excellent
**Rating:** 8
**Confidence:** 4

**Summary:**

The author investigated longitudinal treatment effect estimation from the perspective of ODE discovery. They introduced a framework capable of converting any ODE discovery method into a treatment effect estimation approach. They explored the distinctions between ODE discovery and treatment effect estimation. Furthermore, they presented a versatile solution capable of accommodating both continuous and categorical treatments. Importantly, their method yields interpretable equations, a crucial requirement in the healthcare field.

**Strengths:**

- Method offers interpretable solutions for better understanding.
- Handles various treatments through defined continuous or piecewise equations.
- Applicable for estimating treatment effects at both population and individual levels.
- Manages diverse between-subjects variability effectively.
- Yields improved results even in cases of model misspecification.

**Weaknesses:**

- Determining the correct form of ODE is highly challenging.
- The discussion lacks guidance on learning a feature library, a crucial aspect for this method.
-Assuming y=x poses the question: what if this assumption is invalid? Would ODE or neural networks be more appropriate to model the outcome function? How might this framework integrate with a neural network or other machine learning techniques as g function for the outcome?

**Questions:**

Is it possible to combine any feature learning method within this framework to acquire the necessary dictionary? My primary concern revolves around the challenge of determining the equation's form.

---

> ### Author Response · Authors · 2023-11-15
> **Thank you for your review**
>
> _We thank the reviewer for their time and helpful comments. Our rebuttal below follows the structure of your review. To facilitate further discussion, we have also stated how each specific comment changed our paper_
>
> **Learning ODEs.** You are correct. ODE learning is an incredibly challenging task and, in certain scenarios, infeasible. For these scenarios, we recommend a practitioner rely on the tried-and-tested pure inference machines for treatment effects inference (such as those built on top of neural networks or random forests and beyond).
>
> However, there are also multiple scenarios where ODE learning actually _is_ feasible, and moreover- **_preferred_**. These scenarios include pharmacology, physics, and even economics. The reason is that existing literature on these topics (outside of machine learning) is already heavily driven by the need to discover and recover underlying ODE systems.
>
> **_Actions taken:_** We include pharmacology as an example in Section 3 to further emphasize where ODE discovery makes sense. We also explicitly list the limitations associated with INSITE (and our general framework) in our ethics statement. Hopefully, guiding a practitioner on when (and when not) to use our ODE-learner-based approach to TE inference.
>
>
> **Learning feature libraries.** Setting a good feature library requires two important considerations:
>
> 1. Which ODE method are we using?
> 2. In which scenario are we applying the method?
>
> For 1, we have included a new Appendix M, which explains SINDy (our underlying ODE learner) in detail, as well as points the interested reader to alternative ODE learners, should SINDy be insufficient (wrt to feature sets or other reasons).
>
> For 2, our method heavily relies on domain knowledge. We believe domain knowledge should, of course, take into account point 1 when making a decision on the ODE learner. While this could be problematic, the same is also true for _any_ ML method to be applied in practice.
>
> **_Actions taken:_** We include Appendix M, which explains, in detail, SINDy and provides alternatives.
>
> **_We thank the reviewer once more for their insightful and positive review. We believe our paper is improved as a direct result of your comments. Should there be any more questions, please ask._**

---

### Official Review · Reviewer_WcGm · 2023-10-31

**Soundness:** 3 good
**Presentation:** 2 fair
**Contribution:** 2 fair
**Rating:** 5
**Confidence:** 3

**Summary:**

The authors present a framework that can transform any ODE discovery method into a treatment effects method. This involves reformulating the longitudinal heterogeneous treatment effects problem as an ODE discovery problem, with the goal of recovering the underlying system of ODEs based on observed datasets. A proposed model called INSITE (section 5) is built using this framework and tested in accepted benchmark settings.

**Strengths:**

- The exposition of the method is quite clear. The proposed INSITE framework is very flexible, as it can be used on top of many other time varying TE methods.

- It's reasonable and natural to model this treatment effect over time problem as a dynamical system with ODEs representation. This research brings together two previously separated fields, i.e., temporal TE estimation and ODE discovery, which set up a bridge for these communities.

- The experimental simulation, especially in the appendix, is comprehensive and appears to be reproducible for me.

**Weaknesses:**

1. The presentation of this paper, such as the related work, should be further improved before acceptance. e.g., the $T$ appears at section 2 but is introduced in section 3 ("is called the time horizon"). Another example is ODE discovery paragraph in B.1; I didn't understand why RNN and LSTM are considered ODE discovery methods.

2. Compared to existing black-box TE methods (like neural-network based architectures), the authors claim that the main contribution of this work is a human-readable, interpretable framework. But if the current form of the manuscript only offers empirical validation, I feel that its contributions might not yet meet the threshold of ICLR conf. I wonder that have authors investigated the inherent theoretical properties of the ODE-based framework (which is a white-box model compared to deep network)? e.g., for certain nonlinear differential equations, the solution might be chaotic, meaning it's highly sensitive to initial conditions $\boldsymbol{x}_0, \boldsymbol{v}$ and treatment plan $\boldsymbol{a}$ at $t_0$.

3. Given the claimed emphasis on 'human-readability', how does the proposed method facilitate better decision-making or insights for practitioners in the field is still unclear. Addressing these concerns with more discussions on closed-form ODE would further enhance this work.

4. Minor issues don't affect the rating:

- page 8: $\gamma = 0$ corresponds no time-dependent -> add 'to' for consistency

- Table 5: word 'Intrepetible' -> 'Interpretable'

- there's one paragraph exceeds the 9 pages limit of the main body

- the full name of ODE/ODEs appears many times in the text, which seems unnecessary

**Questions:**

- The assumption 3.1 is not only about existence and uniqueness (these are very mild assumptions and acceptable), but also about the continuous trajectory. I wonder if this is too strict and it may limit the scope of application for the method you've proposed. Could it be relaxed to weaker conditions, such as piecewise continuity? As time changes, different patterns of $\boldsymbol{x}(t)$ may emerge over the interval from $t_0$ to $T$.

- In practice, what method do you use for solving ODEs? How about the complexity of gradient query / solving for this numerical system? Is your method faster than the neural network-based approaches?

---

> ### Author Response · Authors · 2023-11-15
> **Thank you for your review (1/2)**
>
> _We thank the reviewer for their time and helpful comments. Our rebuttal below follows the structure of your review. To facilitate further discussion, we have also stated how each specific comment changed our paper_
>
> **1. Presentation of the paper.** Thank you for pointing us to these necessary corrections. We have taken the following steps in our revision to address your concerns:
>
> 1. _On $T$:_ We have revised the paper to introduce the concept of the time horizon earlier in Section 2.
> 2. _On RNNs and LSTMs for ODE discovery:_ In Appendix B.1, we have amended the text to replace references to RNN and LSTM with a more accurate discussion of Neural ODEs. This modification provides clarity on our approach and aligns better with the context of ODE discovery.
>
> We believe these changes enhance the overall presentation and coherence of the paper, and we hope that they address your concerns effectively.
>
> **_Actions taken:_** Adjusted the paper where necessary to incorporate the above suggestions.
>
>
> **2. Theoretical Contributions.** Our main contribution lies in creating a _usable framework_ that translates ODE discovery methods into the treatment effects problem formulation, bridging two previously distinct fields.
>
> Our theoretical contribution is the alignment of two utterly different problem formulations and assumptions between treatment effects inference and ODE discovery. As shown in Figure 2, ODE discovery methods only work out of the box in the simplest settings (i.e., continuous treatments and only one kind of between-subject variability (BSV) corresponding to a noisy measurement). We demonstrate how to extend ODE discovery methods to allow for discrete treatment plans (both dynamic and static) and two additional sources of BSV: covariate models and parameter distributions (BSV levels C and D). Identification of these two dimensions (axes in Figure 2) and subsequent reconciliation of discrepancies (as shown in Section 4) constitute a major theoretical milestone.
>
> While the theoretical properties of ODE discovery are not the central focus of this work, we acknowledge their importance and direct readers to references A and B for foundational background in this area.
>
> **_Actions taken:_** We have included references [A] and [B] in the main text of our paper, referring the interested reader to the many theoretical underpinnings of ODE discovery.
>
> **3. Facilitating decision-making and insight.** We have included Appendix I, where we demonstrate that our framework, INSITE, successfully identifies equations that are not only predictive but also closely resemble or exactly match the true underlying equations, facilitating further analyses, such as stability analysis, impact analysis, and optimal treatment analysis.
>
> As we have now elaborated in Section 3, many fields already use ODEs as a formal language to express time dynamics. One such example is pharmacology, where a large portion of the literature is dedicated to recovering an ODE from observational data. The found ODE is then used to reason about possible treatments and disease progression ([D],[E]).
>
> Furthermore, in Section 6.1, we illustrate that our method remains effective even when the common overlap assumption is not met. This aspect of INSITE enhances decision-making in complex scenarios where traditional assumptions may not hold.
>
>
> **4. Corrections:** Thank you for pointing us to these errors; they have now been implemented. We kindly note that the ethics and reproducibility statement do not count toward the page limit, as per the official ICLR author guide instructions: https://iclr.cc/Conferences/2024/AuthorGuide.
>
> `rebuttal continues in next comment...`

---

> > ### Author Response · Authors · 2023-11-15
> > **Thank you for your review (2/2)**
> >
> > `... rebuttal continued`
> >
> > ## Questions
> >
> > **Assumption 3.1.** We agree that a form of our proposed approach could be adapted to discovering piecewise continuous systems. However, we assume that the trajectory is continuous to make our approach compatible with most of the current ODE discovery methods that make this assumption as well.
> >
> > Interestingly, there does exist a new recent research direction of piecewise continuous system discovery methods [C]. Therefore, we view this as a possible future work extension of this paper.
> >
> > **_Actions taken:_** We include [C] in our "future work" paragraph in the conclusion of our paper.
> >
> > **Solving ODEs.** To solve the discovered ODE, we use a standard numerical ODE solver of the Euler method, although other solvers can also be used, such as Runge-Kutta methods. At inference time, this approach is comparable to the inference time with a neural network to make a prediction. Interestingly, discovering ODEs, such as with the underlying sparse identification of nonlinear dynamics (SINDy) method, is faster than training neural network-based TE methods.
> >
> >
> > **_We thank the reviewer once more for their insightful review. We believe our paper is improved as a direct result of your comments. Should there be any more questions, please ask._**
> >
> > ## References
> >
> > [A] Brunel, Nicolas JB. "Parameter estimation of ODEs via nonparametric estimators." (2008): 1242-1267.
> >
> > [B] Brunel, Nicolas JB, Quentin Clairon, and Florence d’Alché-Buc. "Parametric estimation of ordinary differential equations with orthogonality conditions." Journal of the American Statistical Association 109.505 (2014): 173-185.
> >
> > [C] Wang, Bochen, et al. "The identification of piecewise non-linear dynamical system without understanding the mechanism." Chaos: An Interdisciplinary Journal of Nonlinear Science 33.6 (2023).
> >
> > [D] Changran Geng, Harald Paganetti, and Clemens Grassberger. Prediction of Treatment Response
> > for Combined Chemo- and Radiation Therapy for Non-Small Cell Lung Cancer Patients Using a
> > Bio-Mathematical Model. Scientific Reports, 7(1):13542
> >
> > [E] Joseph D Butner, Dalia Elganainy, Charles X Wang, Zhihui Wang, Shu-Hsia Chen, Nestor F Esnaola, Renata Pasqualini, Wadih Arap, David S Hong, James Welsh, et al. Mathematical prediction of clinical outcomes in advanced cancer patients treated with checkpoint inhibitor immunotherapy. Science advances, 6(18), 2020.

---

> ### Author Response · Authors · 2023-11-21
> **Final Call for Reviewer Engagement: Clarifications and Updates Submitted**
>
> Thank you once again for your invaluable insights and time dedicated to the review process!
>
> If there were any leftover concerns, we would sincerely appreciate the opportunity to clarify them—before the discussion period for the authors ends in one day. We believe our responses (Nov 15) have addressed in detail the full set of questions you raised:
>
> * Improving the presentation of the paper; see **Response (1)**
> * Exposition of the theoretical contributions; see **Response (2)**
> * Facilitating decision-making and insights; see **Response (3)**
> * Corrections; see **Response (4)**
> * Relaxing assumption 3.1; see **Response (Questions - Assumption 3.1)**
> * Clarification on how ODEs are solved; see **Response (Questions - Solving ODEs)**
>
> These are paired with corresponding updates to the submission (Nov 15), including extensive new additional experimental results, discussions, and clarifications—which include two **new** experimental appendices, **appendix J.7 and J.8**.
>
> We would appreciate it if the reviewer kindly let us know if there were any further questions in the very limited time remaining of one day. We are eager to do our utmost to address them!
>
> Thank you,
>
> Paper 5372 authors

---

> > ### Comment · Reviewer_WcGm · 2023-11-22
> > **Acknowledging.**
> >
> > Thanks a lot for answering questions, I'm satisfied with the responses on theoretical analysis and the continuity assumption relaxations. I acknowledge the contributions of this work, however, I also believe that the updates made to address the concerns raised by reviewers, e.g., technical motivations (compared to related work), interpretability (why this format human-readable), and practical guidance, may alter the structure and the content of the paper significantly. Having briefly read the revised pdf and all comments posted, I would maintain my last rating.

---

> > > ### Author Response · Authors · 2023-11-23
> > > **Comprehensive Enhancements Addressing Reviewer Feedback**
> > >
> > > We sincerely appreciate your acknowledgment of the efforts made to address the theoretical analysis and relaxation of continuity assumptions. We are also grateful for your recognition of our paper's contributions.
> > >
> > > In response to your insightful comments and the comments of all other reviewers, we have diligently updated our paper, particularly in enhancing technical motivations in comparison with related work, improving interpretability, and providing practical guidance. These updates, marked in blue for ease of review, enrich the paper while maintaining its original structure and intent.
> > >
> > > We understand your concerns regarding the extent of these changes and assure you that they complement, rather than alter, the fundamental nature of our work. We hope these revisions meet your expectations and further solidify the paper's contributions to the field.
> > >
> > > As the Author discussion period draws to a close, we remain open and responsive to any further questions or concerns. Your feedback has been invaluable, and we are committed to addressing any remaining issues promptly.
> > >
> > > We respectfully request that you consider revising your score in light of these significant improvements. Your positive reassessment would greatly acknowledge the enhancements made to our paper.
> > >
> > > Thank you once again for your time and valuable insights.
> > >
> > > Kind regards,
> > > Paper 5372 authors

---

> > > > ### Comment · Reviewer_WcGm · 2023-11-23
> > > > **thank the authors for their response**
> > > >
> > > > The current version looks good to me, but I'm still don't understand why this model is referred to as a 'human-readable equation', as there seems to be no detailed discussion supporting this claim.
> > > >
> > > > Also I have raised concerns regarding numerical problems in computation, yet the paper does not even mention which numerical method (solver) were used or what their setting parameters are. Discussing these aspects might enhance the quality of this work in the future.

---

> ### Author Response · Authors · 2023-11-23
> **Addressing Reviewer's Concerns on Terminology and Numerical Solver Details**
>
> Thank you once again for your constructive feedback and for dedicating your time to the review process!
>
> Regarding the term 'human-readable equation,' we initially used this phrase to emphasize the interpretability of the discovered closed-form ODE. However, we acknowledge that this term might have been ambiguous without a detailed explanation. Thus, we have revised the manuscript, replacing 'human-readable equation' with 'equation' in the abstract and introduction to avoid any misunderstanding—therefore, this term does not appear in the paper anymore.
>
> Concerning your valuable insights on numerical problems and solver details, we have now added an additional **(new) appendix N** titled “Numerical Solver and Parameters”, which details in one place the following.
>
> The numerical method (solver) used for INSITE and the other ODE discovery methods (e.g., SINDy) is that of the **Euler method**, which is a first-order numerical algorithm for solving ODEs given an initial value. This uses a time step size of $\delta_t = 0.166$ seconds, which was sufficient for our experiments. This numerical method is used with the same settings across all methods, including INSITE. We kindly note that first, the ODE discovery methods discover an ODE, and then at inference time, when we wish to determine future values, we forward simulate the initial observed value using the discovered ODE and this Euler numerical solver to estimate future values at future times. We also kindly highlight further implementation details can be found in appendices E, F and H.
>
> As the author discussion period draws to a close, we remain open and responsive to any further questions or concerns. Your feedback has been invaluable, and we are committed to addressing any remaining issues promptly.
>
> We hope this clarifies your concern and you will kindly consider accepting the paper. Thank you once again for your invaluable insights and the opportunity to improve our manuscript.
>
> Kind regards,
>
> Paper 5372 authors

---

### Official Review · Reviewer_aGwv · 2023-10-31

**Soundness:** 3 good
**Presentation:** 3 good
**Contribution:** 3 good
**Rating:** 8
**Confidence:** 2

**Summary:**

This paper studies longitudal treatment effect estimation from a new prospective, using ordinary differential equation (ODE). This potentially opens up a new paradigm of treatment effect estimation methods that does not rely on neural network. The ODE approach naturally offers interpretability and requires slightly different assumptions for identification. The authors first discussed the difference and similarities between longitudal treatment effect estimation and ODE, then proposed a framework for bridging the problem and solution. The proposed framework is then compared with several neural network based longitudal effect estimators.

**Strengths:**

1. The ODE perspective is very novel for longitudal treatment effect estimation.
2. The paper is well-written and well-organized, with extensive additional information for reproducibility.
3. The ODE framework can relax the overlap assumption in causal effect estimation.

**Weaknesses:**

1. There seems to lack a formal identification result in terms of identifying the causal effects from the ODE perspective.
2. The experiments are conducted on synthetic datasets only. Results on real-world datasets could further strengthen the evaluation.

**Questions:**

What are the "strong assumptions" mentioned in this statement? " Unique to the proposed framework and INSITE, is that the discovered
differential equation is fully interpretable; however, it relies on strong assumption"

The data generation process in the synthetic benchmark in Eq 5 seems to closely follow with the ODE assumption in Eq 3, is there experiment results when such the data is generated in different ways?

---

> ### Author Response · Authors · 2023-11-15
> **Thank you for your review**
>
> _We thank the reviewer for their time and helpful comments. Our rebuttal below follows the structure of your review. To facilitate further discussion, we have also stated how each specific comment changed our paper_
>
>
> **1. Identification.** When the complete causal system is known (including the underlying structural equations), one can use the causal system to solve problems such as counterfactual regression, _without_ resorting to regression techniques. Chapter 3.4. "Counterfactual analysis in structural model" in Pearl (2009) [A], makes this point explicitly. There, a counterfactual query is resolved using the complete causal system (being the causal graph and structural equations).
>
>
> Of course, most practical scenarios do not have access to the full causal system. Performing counterfactual regression in these settings then relied on regression strategies, which are only approximate. Using regression techniques to perform counterfactual regression then requires at least some proof of identification. The burden of proof is only necessary because we cannot access the complete causal system.
>
>
> In our setting, we actually learn the _complete causal system_ (including the structural equations). Papers proposing ODE learners claim to recover the true underlying ODE. If there is such one true underlying equation (cfr. Assumption 3.1), then it is only logical that this equation corresponds to the one true underlying (causal) structural equation. Hence, by learning the true ODE, one has access to the full causal system, which in turn alleviates the need for an identification proof.
>
> Of course, the accuracy and identification guarantees in our setting (above) are only as strong as those provided by the underlying ODE learner. In case one uses a faulty ODE learner (with, for example, poor guarantees), INSITE would suffer as a result. In this case, our framework is at least explicit about which assumptions and settings one can rely on ODE discovery for TE inference.
>
> **_Actions taken:_** We include Appendix M, which explains which identification guarantees (and under which necessary assumptions) SINDy (our example underlying ODE learner) provides. Furthermore, Appendix M also includes alternative ODE learners, in case SINDy is unsuitable for the practical setting at hand.
>
>
> **2. Experiments.** Note that we adopt the experimental settings from the related literature on ODE discovery. While in TE literature, it is common to rely on semi-synthetic data, truly testing our method requires more than just access to the unobserved potential outcome (which is the reason TE literature resorts to semi-synthetic testbeds). Specifically, we also require access to the underlying causal system, which is only possible with fully synthetic data.
>
> We also kindly refer to Appendix J, which includes many more experiments that may interest the reviewer. In particular, we included experiments on:
> * Sensitivity on $\lambda$
> * Parametric distributions of numeric constants
> * Varying sample sizes and higher noise
> * Model misspecification
>
>
> **_We thank the reviewer once more for their insightful and positive review. We believe our paper is improved as a direct result of your comments. Should there be any more questions, please ask._**
>
>
> ## References
>
> [A] Judea Pearl. _Causal inference in statistics: An overview._ 2009.

---

> ### Comment · Reviewer_aGwv · 2023-11-15
>
> Thanks for the discussion. They addressed most of my concerns.
>
> On the public comment regarding Symbolic Regression, I think it will be beneficial to discuss the position of this work in SR literature. I am familiar with TE literature and unfamiliar with SR/ODE, and I would assume many readers of this paper would also be the same.
>
> I increased my score based on the rebuttal. But I also want to emphasize that my score comes with a low confidence as I am unfamiliar with ODE and SR literature.

---

> > ### Author Response · Authors · 2023-11-16
> > **Thank you!**
> >
> > Dear reviewer aGwv,
> >
> > Thank you very much for acknowledging our rebuttal and increasing your score!
> >
> > Indeed, like you, we agree with the the general comment on symbolic regression and now discuss it in our paper.
> >
> > Thanks again,
> >
> > The authors

---

### Official Review · Reviewer_6MV1 · 2023-10-31

**Soundness:** 3 good
**Presentation:** 3 good
**Contribution:** 3 good
**Rating:** 8
**Confidence:** 3

**Summary:**

The paper proposes the Individualized Nonlinear Sparse Identification Treatment Effect (INSITE) framework for estimating heterogeneous time-varying treatment effects given dynamic and statist covariates. INSITE leverages deterministic ODE discovery methods to infer the population differential equation, which is fine-tuned to recover patient-specific differential equations. Experimental results on two synthetic datasets demonstrate that INSITE is competitive in counterfactual prediction relative to baselines.

**Strengths:**

- The paper outlines a comprehensive framework for leveraging ODE discovery methods in time-varying treatments/covariates setup
- The paper is relatively well-written and easy to follow
- Experimental results demonstrate that the proposed approach is competitive relative to baselines
- The learned ODES are interpretable, unlike previously proposed neural network approaches

**Weaknesses:**

- I encourage the authors to include a discussion section focused on the limitations of the proposed approach. The paper makes strong assumptions about the underlying dynamics, e.g.,
1) The success of the INSITE is dependent on the choice of the library of candidate functions
2) INSITE assumes that the system is sparse, which rarely holds in high-dimensional settings
3) INSITE assumes that the system is deterministic and noise-free. However, real-world systems often have noise and other stochastic elements, which can significantly affect the model's accuracy.
- I encourage the authors to include a complete description of how INSITE handles different treatment types (categorical and continuous ). Given that this setup is distinct from the previous ODE discovery methods, complete details should be provided in the main paper

**Underwhelming Experiments**
- While the paper claims to handle irregularly sampled data, which is typical in real-world settings (i.e., patient covariates are not measured continuously over time), the synthetic experiments don't seem to explore this scenario
- The synthetic experiments are too simplistic, $g(x)$ is an identity function, $x$ is 1-dimensional. I encourage the authors to explore more challenging scenarios, including semi-synthetic datasets

**Questions:**

- For the BSV scenario A (eqns 5 and 6), shouldn't we expect ODE methods to do as well as INSITE
- Fig 3(b): Shouldn't we expect INSITE to be monotonically increasing with $\gamma$

---

> ### Author Response · Authors · 2023-11-15
> **Thank you for your review**
>
> _We thank the reviewer for their time and helpful comments. Our rebuttal below follows the structure of your review. To facilitate further discussion, we have also stated how each specific comment changed our paper_
>
> **Limitations.** We agree with the reviewer, limitations should be clearly stated:
> 1. A correct set of candidate functions (tokens) is necessary for correct model recovery. To show the importance of this, we include experiments where we have wrongly specified this token library (see Appendix J.6) and observe degrading performance as a result. We will explicitly refer to Appendix J.6 from our concluding section.
>
> 2. Sparsity is an assumption stemming from the type of function we wish to recover. Only when the function is sparse (that is, it contains not too many terms) can we deem it interpretable. There is nothing technical that limits INSITE (or the underlying SINDy method) to sparse functions; however, relaxing sparsity would also result in less useful recovered equations.
>
> 3. ODEs are noise-free. Since we recover ODEs, we therefore recover noiseless (deterministic) functions. In case one is interested in stochastic differential equations (SDEs), one needs to extend our work to encompass this type of differential equation.
>
>
> **_Actions taken:_** The above limitations (1-3) are now stated in our ethics statement. We include it there as it should be considered explicitly before applying our method (and framework) in practical settings.
>
>
> **Treatment systems.** For a complete description of incorporating different treatment types, we refer to Appendix D. Due to the extent of our discussion, we moved this to our supplementary materials. In particular, Table 6 (in Appendix D) shows how a non-continuous treatment is transformed into a dynamical system. However, on page 5, we have included a brief description of the table presented in Appendix D.
>
> **_Actions taken:_** Include a summary of Table 6 in Appendix D in the main text.
>
>
> **Experiments.** The underlying method SINDy implemented supports irregularly sampled time series data [B, C], therefore INSITE also supports learning ODEs from irregularly sampled time series. For the function introduced in Eq. (3) and its common assumption as the identity function, please refer to related works [A] for a detailed background.
>
> **_Actions taken:_** Included a new Appendix M to detail SINDy and other ODE discovery learners. We will also provide, before the end of the discussion period (due to high running times), a set of experiments on irregular sampling and alternative functions $g(x)$.
>
> ## Questions
>
> **BSV scenario A.** We agree that for the BSV scenario A (eq.5 and eq.6), we expect ODE discovery methods to perform as well as INSITE. This discrepancy could likely occur due to the accuracy to which the parameters are optimized internally within the SINDy method. We observe that an alternative optimization method that is more accurate than that of WSINDy (Messenger et al., 2021) achieves a better performance than SINDy. Whereas INSITE fine-tunes the ODEs parameters to the initially observed trajectory of a patient at inference time using a higher-accuracy optimization method of the Broyden–Fletcher–Goldfarb–Shanno (BFGS) algorithm, a second-order nonlinear optimization method.
>
> **INSITE and $\gamma$.** We agree that INSITE's error is monotonically increasing with $\gamma$. However, we note that it still achieves a more accurate prediction compared to existing methods.
>
> The fact that INSITE's error increases with $\gamma$ is also expected. As $\gamma$ increases, overlap gets more violated. While INSITE is _more robust_ against overlap violations, it is not immune. The reason is that the hypothesis space associated with ODEs is still large enough that we cannot confidently extrapolate to low-sample regions (such as regions of low/non-overlap). On the other hand, the ODE hypothesis space is still smaller than the neural networks used in other TE inference techniques.
>
>
> **_We thank the reviewer once more for their insightful and positive review. We believe our paper is improved as a direct result of your comments. Should there be any more questions, please ask._**
>
> ## References
>
> [A] Edward L Ince. Ordinary differential equations. Courier Corporation, 1956.
>
> [B] Brunton, Steven L., Joshua L. Proctor, and J. Nathan Kutz. "Discovering governing equations from data by sparse identification of nonlinear dynamical systems." Proceedings of the national academy of sciences 113.15 (2016): 3932-3937.
>
> [C] Goyal, Pawan, and Peter Benner. "Discovery of nonlinear dynamical systems using a Runge–Kutta inspired dictionary-based sparse regression approach." Proceedings of the Royal Society A 478.2262 (2022): 20210883.

---

> > ### Author Response · Authors · 2023-11-20
> > **New Additional Experiments Followup (1/2)**
> >
> > Thank you once again for your invaluable insights and time dedicated to the review process. We are thrilled to present two **new** experiments that empirically verify (1) that INSITE handles irregularly sampled data and (2) INSITE can also work with more challenging observation functions $g(x)$. These are denoted by responses **(A)** and **(B)** respectively below.
> >
> > We have carefully considered all your comments and have worked diligently to address each one. Should there be any remaining questions or concerns, we welcome the opportunity to clarify them before the author discussion period concludes. In alignment with your feedback, we have updated our manuscript with pertinent discussions and clarifications.
> >
> > ### **(A) INSITE also empirically works for irregularly sampled data**
> >
> > INSITE also supports irregularly sampled time series data, as its underlying ODE discovery method of SINDy supports discovering ODEs from irregularly sampled time series data [B, C]. We further empirically verify this by re-running SINDy and INSITE across all datasets, now with the irregularly sampled datasets, where we randomly sub-sampled irregularly, excluding 10\% of the original observations along the time dimension. The subsequent results, conducted across ten random seeds, are outlined in table 29 below. We observe that INSITE and SINDy still achieve good performance (hence low prediction error).
> >
> > **Table 29**
> > | Method | eq.(5).A | eq.(5).B | eq.(5).C | eq.(5).D | eq.(6).A | eq.(6).B | eq.(6).C | eq.(6).D | Cancer PKPD |
> > |--------|--------------------------------|--------------------------------|--------------------------------|--------------------------------|----------------|----------------|----------------|----------------|-------------|
> > | A-SINDy | 0.11±1.93e-17 | 0.11±1.93e-17 | 0.12±1.93e-17 | 0.13±0.00 | 0.90±1.54e-16 | 0.90±1.54e-16 | 0.80±0.00 | 0.96±1.54e-16 | 1.68±3.08e-16 |
> > | INSITE | **0.02±0.00** | **0.03±0.00** | **0.04±0.00** | **0.04±0.00** | **0.85±1.54e-16** | **0.85±0.00** | **0.75±0.00** | **0.70±0.00** | **1.00±1.54e-16** |
> >
> >
> > ***Actions taken:*** We now include this new experimental result in a **(new) Appendix J.7**, entitled **"INSITE also empirically works for irregularly sampled data"**.

---

> > ### Author Response · Authors · 2023-11-20
> > **New Additional Experiments Followup (2/2)**
> >
> > ### **(B) INSITE can also work with more challenging observation functions $g(x)$**
> >
> > Although the observation function $g$ that describes the treatment outcome as a function of the observed features is often prespecified by the user as it often models the outcome as one of the features ($g(\mathbf{x})=x_j$), e.g., the tumor volume, other more complex forms of $g$ that are not prespecified are also of interest. To investigate whether INSITE can work under these more challenging settings of having $g$ be a non-linear function, we empirically investigated this by implementing three more challenging forms of $g(x)$ representing more complex settings. These are (1) an exponential function $g(\mathbf{x})=\exp(x_j)$ that often models rapid changes or growth, commonly found in many biological and economic systems [D], (2) a quadratic function $g(\mathbf{x})=x_j^2$ that models a non-linear relationship in systems [E], and (3) a trigonometric function $g(\mathbf{x})=\sin(x_j)$ that often models periodic components within oscillatory dynamics, which are commonly found in natural and engineering systems [F]. For each of these new observation functions, we re-ran SINDy and INSITE across all datasets, conducted across five random seeds, with the results tabulated in table 30 below. We observe that INSITE and SINDy can still achieve acceptable performance.
> >
> > **Table 30**
> > | $g(x)$    | Method | eq.(5).A | eq.(5).B | eq.(5).C | eq.(5).D | eq.(6).A | eq.(6).B | eq.(6).C | eq.(6).D | Cancer PKPD |
> > |-------------|------------|-------------------------------|-------------------------------|-------------------------------|-------------------------------|----------------|----------------|----------------|----------------|-----------------|
> > | **$x$**       | A-SINDy    | 0.11 ±0.00                    | 0.11 ±0.00                    | 0.13 ±0.00                    | 0.15 ±0.00                    | 1.46 ±0.00     | 1.47 ±0.00     | 1.40 ±0.00     | 1.56 ±0.00     | 1.29 ±0.00      |
> > |             | INSITE     | 0.02 ±0.00                    | 0.03 ±0.00                    | 0.04 ±0.00                    | 0.05 ±0.00                    | 0.96 ±0.00     | 0.96 ±0.00     | 0.82 ±0.00     | 0.90 ±0.00     | 0.79 ±0.00      |
> > | **$\exp(x)$**  | A-SINDy    | 0.75 ±0.02                    | 0.75 ±0.02                    | 0.74 ±0.03                    | 0.76 ±0.06                    | 0.76 ±0.34     | 0.76 ±0.34     | 1.21 ±1.34     | 1.02 ±0.57     | 0.94 ±0.20      |
> > |             | INSITE     | 0.59 ±0.03                    | 0.59 ±0.02                    | 0.61 ±0.02                    | 0.62 ±0.05                    | 0.67 ±0.30     | 0.68 ±0.32     | 1.08 ±1.31     | 0.90 ±0.59     | 0.78 ±0.13      |
> > | **$x^2$**     | A-SINDy    | 0.10 ±0.01                    | 0.12 ±0.00                    | 0.12 ±0.00                    | 0.12 ±0.01                    | 1.16 ±1.03     | 1.15 ±1.03     | 2.11 ±3.58     | 1.69 ±2.02     | 0.57 ±0.10      |
> > |             | INSITE     | 0.03 ±0.00                    | 0.09 ±0.00                    | 0.09 ±0.00                    | 0.09 ±0.01                    | 1.12 ±1.04     | 1.11 ±1.04     | 2.04 ±3.57     | 1.63 ±2.03     | 0.52 ±0.08      |
> > | **$\sin(x)$**  | A-SINDy    | 2.42 ±0.07                    | 2.42 ±0.07                    | 2.31 ±0.07                    | 2.42 ±0.40                    | 1.62 ±0.43     | 1.62 ±0.43     | 2.30 ±1.86     | 1.99 ±0.69     | 2.00 ±0.46      |
> > |             | INSITE     | 2.09 ±0.10                    | 2.09 ±0.10                    | 2.06 ±0.08                    | 2.22 ±0.25                    | 1.46 ±0.36     | 1.41 ±0.42     | 1.96 ±1.78     | 1.72 ±0.76     | 1.59 ±0.29      |
> >
> > ***Actions taken:*** We now include this new experimental result in a **(new) Appendix J.8**, entitled **"INSITE can also work with more challenging observation functions $g(x)$"**.
> >
> > ---
> >
> > ### References
> >
> > [D] Kaitlyn E Johnson, Grant Howard, William Mo, Michael K Strasser, Ernesto ABF Lima, Sui Huang, and Amy Brock. Cancer cell population growth kinetics at low densities deviate from the exponential growth model and suggest an allee effect. PLoS biology, 17(8):e3000399, 2019.
> >
> > [E] Armengol Gasull, Antoni Guillamon, and Jordi Villadelprat. The period function for second-order quadratic odes is monotone. Qualitative Theory of Dynamical Systems, 4(2):329–352, 2004.
> >
> > [F] Kanti Bhushan Datta and Bosukonda Murali Mohan. Orthogonal functions in systems and control, volume 9. World Scientific, 1995.
> >
> > ---
> >
> > Should any uncertainties linger, we sincerely invite you to share them with us before the author discussion period concludes in the next two days. Your continued engagement is deeply appreciated, and we are at your disposal for any further elucidation, thank you!

---

### Public Comment · ~Shurui_Gui1 · 2023-11-12

Dear authors,

I extend my sincere appreciation for your thought-provoking paper on integrating ODE discovery into time-series treatment effect estimations. After reading your paper, I would like to present a couple of observations and inquiries for clarification:

1. Throughout the paper, the concept of ODE discovery is recurrently mentioned. However, I observed a certain ambiguity in its definition and exposition. This sentiment seems to resonate with the feedback from other reviewers as well. In my understanding, the method SINDy, which you refer to as a method of ODE discovery, is fundamentally a part of Symbolic Regression. This area is pivotal, especially considering your attempt to bridge SINDy with treatment effect estimations. Yet, I noticed a conspicuous absence of related work in Symbolic Regression. Could there be a specific rationale for this omission that perhaps eluded my notice? For a more comprehensive understanding, one might refer to Section 3.1.2 of [1].

2. These symbolic regression (or ODE discovery) methods are well explored as a way for algorithmic alignments and generalization  [2, 3], while ODE interpretation is generally a natural result. In light of this, I'm curious about additional motivations behind employing these methods within treatment effect estimations in addition to interpretability. Such motivation clarity, I believe, would substantially enhance the overall impact and relevance of your research.

I eagerly await your response and further insights.

Kind regards,

Shurui Gui

[1] Camps-Valls, G., Gerhardus, A., Ninad, U., Varando, G., Martius, G., Balaguer-Ballester, E., ... & Runge, J. (2023). Discovering causal relations and equations from data. arXiv preprint arXiv:2305.13341.

[2] Mouli, S. C., Alam, M. A., & Ribeiro, B. (2023). MetaPhysiCa: OOD Robustness in Physics-informed Machine Learning. arXiv preprint arXiv:2303.03181.

[3] Cranmer, M., Sanchez Gonzalez, A., Battaglia, P., Xu, R., Cranmer, K., Spergel, D., & Ho, S. (2020). Discovering symbolic models from deep learning with inductive biases. Advances in Neural Information Processing Systems, 33, 17429-17442.

---

> ### Author Response · Authors · 2023-11-15
> **Thanks for your comment and interest!**
>
> _Dear Shurui, Thank you very much for your interest in our paper! We truly appreciate this, as well as the comments you made. Hence, we will, as we have for our reviewers, answer in the structure of your comment below._
>
> **1. Symbolic regression.** We agree with your statement:
> > ODE discovery, is fundamentally a part of Symbolic Regression.
>
> Indeed, A large part of the ODE discovery literature employs techniques such as symbolic regression. However, symbolic regression also encompasses equations _beyond_ ODEs. Since we do not claim to also connect other types of equation systems to TEs, we did not want to overclaim: "a connection between TEs and symbolic regression". More work is required to make such a claim.
>
> However, we do agree that further context is required. As such, we will use the references you have provided to correctly position ODE discovery within the larger symbolic regression community.
>
>
> **2. Motivation.** Our TE community's focus of the last few years has mostly been: to mitigate bias in favor of increased accuracy in counterfactual regression. This is a useful cause, of course, but it is unlikely that practitioners will adopt the state-of-the-art in TE inference (implying a complete retraining of their systems), for a possible tiny increase in accuracy.
>
> Hence, with ODEs, one gets _additional_ benefits beyond accuracy. Indeed, by mapping ODE discovery methods (such as SINDy, or perhaps alternative symbolic regressors), we get accuracy while also getting interpretability. The latter is incredibly important as most application areas for TE inference would benefit hugely from a fully interpretable model.
>
> Beyond the above, we hope our paper also engages researchers (and practitioners) from the ODE discovery field (or perhaps, as you suggest, the wider symbolic regression community). In this way, both TEs inference and ODE discovery could build upon ideas from both sides, pushing research even further.
>
> **_Thank you once more for commenting on our paper! We believe your comments make our manuscript better. Hopefully, more people will find our paper as thought provoking as you have._**

---

### Meta-Review · Area_Chair_PmWi · 2023-12-06

**Metareview:**

The paper proposes a human-readable ordinary differential equation (ODE) solution, offering advantages in interpretability, handling irregular sampling, and a different set of identification assumptions.

pros:
+ Novel perspective on longitudinal treatment effect estimation using ODEs.
+ Well-organized and well-written, with extensive information for reproducibility.
+ Versatile solution accommodating various treatments and variability levels.

cons:
+ Lack of formal identification results for causal effects from the ODE perspective.
+ Lack of sufficient robustness testing of methods in empirical studies; lack of real data evaluation
+ Challenges in determining the correct ODE form and lack of guidance on feature library learning.

**Justification For Why Not Higher Score:**

lack of formal identification for causal inference

**Justification For Why Not Lower Score:**

novel contributions of treatment effect estimation using ODEs

---

### Decision · Program_Chairs · 2024-01-16

Accept (spotlight)